# Learning Energy-Based Generative Models via Potential Flow: A Variational Principle Approach to Probability Density Homotopy Matching

**Junn Yong Loo**                                                     *loo.junnyong@monash.edu*
*Monash University Malaysia*

**Fang Yu Leong**                                                     *leong.fangyu@monash.edu*
*Monash University Malaysia*

**Michelle Adeline**                                          *made0008@student.monash.edu*
*Monash University Malaysia*

**Julia Kaiwen Lau**                                                     *julia.lau@monash.edu*
*Monash University Malaysia*

**Hwa Hui Tew**                                                          *hwa.tew@monash.edu*
*Monash University Malaysia*

**Arghya Pal**                                                        *arghya.pal@monash.edu*
*Monash University Malaysia*

**Vishnu Monn Baskaran**                                             *vishnu.monn@monash.edu*
*Monash University Malaysia*

**Chee-Ming Ting**                                                 *ting.cheeming@monash.edu*
*Monash University Malaysia*

**Raphaël C.-W. Phan**                                             *raphael.phan@monash.edu*
*Monash University Malaysia*

**Reviewed on OpenReview:** *https://openreview.net/forum?id=vc7poEYOFK*

## Abstract

Energy-based models (EBMs) are a powerful class of probabilistic generative models due to their flexibility and interpretability. However, relationships between potential flows and explicit EBMs remain underexplored, while contrastive divergence training via implicit Markov chain Monte Carlo (MCMC) sampling is often unstable and expensive in high-dimensional settings. In this paper, we propose Variational Potential (VAPO) Flow Bayes, a new energy-based generative framework that eliminates the need for implicit MCMC sampling and does not rely on auxiliary networks or cooperative training. VAPO learns an energy-parameterized potential flow by constructing a flow-driven density homotopy that is matched to the data distribution through a variational loss minimizing the Kullback-Leibler divergence between the flow-driven and marginal homotopies. This principled formulation enables robust and efficient generative modeling while preserving the interpretability of EBMs. Experimental results on image generation, interpolation, out-of-distribution detection, and compositional generation confirm the effectiveness of VAPO, showing that our method performs competitively with existing approaches in terms of sample quality and versatility across diverse generative modeling tasks. The code is available at `https://github.com/ljun0004/VAPO`.

# 1 Introduction

Energy-based models (EBMs) have emerged as a flexible and expressive class of probabilistic generative models (Nijkamp et al., 2019; Du & Mordatch, 2019; Grathwohl et al., 2020b; Gao et al., 2020; Du et al., 2021; Gao et al., 2021; Grathwohl et al., 2020a; Yang et al., 2023; Zhu et al., 2024). By assigning a potential energy that correlates with the unnormalized data likelihood (Song & Kingma, 2021), EBMs offer a structured energy landscape for probability density estimation, providing several notable advantages. First, EBMs are interpretable, as the underlying energy function can be visualized in terms of energy surfaces. Second, they are highly expressive and do not impose strong architectural constraints (Bond-Taylor et al., 2022), enabling them to capture complex data distributions. Third, EBMs exhibit inherent robustness to Out-of-Distribution (OOD) inputs, given that regions with low likelihood are naturally penalized (Du & Mordatch, 2019; Grathwohl et al., 2020a). Building on their origins in Boltzmann machines (Hinton, 2002), EBMs also share conceptual ties with statistical physics, allowing practitioners to adapt physical insights and tools for model design and analysis (Feinauer & Lucibello, 2021). They have demonstrated promising performance in various applications beyond image modeling, including text generation (Deng et al., 2020), robot learning (Du et al., 2020), point cloud synthesis (Xie et al., 2021a), trajectory prediction (Pang et al., 2021; Wang et al., 2023), molecular design (Liu et al., 2021), and anomaly detection (Yoon et al., 2023).

Despite these advantages, training deep EBMs often relies on implicit Markov Chain Monte Carlo (MCMC) sampling for contrastive divergence. In high-dimensional settings, MCMC suffers from poor mode mixing and slow mixing (Du & Mordatch, 2019; Nijkamp et al., 2019; Gao et al., 2020; Grathwohl et al., 2020a; Nijkamp et al., 2022; Bond-Taylor et al., 2022), yielding biased estimates that may optimize unintended objectives (Grathwohl et al., 2020b). Truncated chains, in particular, can lead models to learn an implicit sampler rather than a true density, which prevents valid steady-state convergence and inflates computational overhead. As a result, the generated samples can deviate significantly from the target distribution (Grathwohl et al., 2020b). To mitigate these issues, some works propose auxiliary or cooperative strategies that learn complementary models to either avoid MCMC via variational inference (Xiao et al., 2021a) or combine short-run MCMC refinements with learned generator distributions (Xie et al., 2020; Grathwohl et al., 2021; Hill et al., 2022). Nevertheless, these approaches could complicate model architectures and training procedures.

In parallel, flow-based models have advanced generative modeling by leveraging continuous normalizing flows and optimal transport techniques to surpass diffusion models in sample quality and efficiency (Kim et al., 2021; Song et al., 2021). Notable examples include Flow Matching (Lipman et al., 2023), which models diffeomorphic mappings between noise and data; Rectified Flow (Liu et al., 2023b), which optimizes sampling paths; Stochastic Interpolants (Albergo & Vanden-Eijnden, 2023; Rezende & Mohamed, 2015; Chen et al., 2018), which incorporate stochastic processes into flows for complex data geometries; Schrödinger Bridge Matching (Shi et al., 2023), which integrates entropy-regularized optimal transport with diffusion; and Poisson Flow Generative Model (PFGM) (Xu et al., 2022), which introduces an augmented space governed by the Poisson equation. However, these methods do not directly parameterize probability density and lack the theoretical advantages of EBMs, such as generating conservative vector fields aligned with log-likelihood gradients (Salimans & Ho, 2021).

Recent approaches, such as Action Matching (Neklyudov et al., 2023), explicitly model the energy (action) to generate data-recovery vector fields, thus providing a structured approach to learning conservative dynamics. Meanwhile, Diffusion Recovery Likelihood (DRL) (Gao et al., 2021) and Denoising Diffusion Adversarial EBMs (DDAEBM) (Geng et al., 2024) refine conditional EBMs by improving sampling efficiency and training stability through diffusion-based probability paths. However, a direct connection between energy-parameterized flow models and explicit (marginal) EBMs remains unexplored, limiting the application of flow-based techniques for learning EBMs. Furthermore, existing generative models have yet to adopt variational formulations, such as the Deep Ritz method, to align the evolution of density paths.

To address the computational challenges of existing energy-based methods, we propose Variational Potential (VAPO) Flow Bayes, a novel energy-based generative framework grounded in variational principles that eliminates the need for auxiliary models and implicit MCMC sampling. VAPO employs the Deep Ritz method to learn an energy-parameterized potential flow, ensuring alignment between the flow-driven density homotopy and the data-recovery likelihood homotopy. To address the intractability of homotopy matching,

we formulate a variational loss function that minimizes the Kullback-Leibler (KL) divergence between these density homotopies. Additionally, we validate the learned potential energy as an effective parameterization of the stationary Boltzmann energy. Through empirical validations, we benchmark VAPO against state-of-the-art generative models, showcasing its competitive performance in Fréchet Inception Distance (FID) for image generation and excellent OOD detection with high Area Under the Receiver Operating Characteristic Curve (AUROC) scores across multiple datasets.

## 2    Background and Related Works

In this section, we provide an overview of EBMs, particle flow, and the Deep Ritz method, collectively forming the cornerstone of the proposed VAPO framework.

### 2.1    Energy-based Models (EBMs)

Denote $\bar{x} \in \Omega \subseteq \mathbb{R}^n$ as the training data, EBMs approximate the data likelihood $p_{\text{data}}(\bar{x})$ via defining a Boltzmann distribution

$$p_B(x) = \frac{e^{\Phi_B(x)}}{Z} \tag{1}$$

where $\Phi_B$ is the Boltzmann energy parameterized via neural networks and $Z = \int_\Omega e^{\Phi_B(x)} \, dx$ is the normalizing constant. Given that this partition function is analytically intractable for high-dimensional data, EBMs perform the Maximum Likelihood Estimation (MLE) by minimizing the negative log-likelihood loss $\mathcal{L}_{\text{MLE}}(\theta) = -\mathbb{E}_{p_{\text{data}}(\bar{x})}[\log p_B(\bar{x})] = \mathbb{E}_{p_{\text{data}}(\bar{x})}\left[\Phi_B(\bar{x})\right] - \mathbb{E}_{p_{\text{data}}(\bar{x})}\left[\log Z\right]$. The gradient of this MLE loss with respect to model parameters $\theta$ is approximated via the contrastive divergence (Hinton, 2002) loss $\nabla_\theta \mathcal{L}_{\text{MLE}} = \mathbb{E}_{p_{\text{data}}(\bar{x})}\left[\nabla_\theta \Phi_B(\bar{x})\right] - \mathbb{E}_{p_B(x)}\left[\nabla_\theta \Phi_B(x)\right]$. Nonetheless, EBMs are computationally intensive due to the implicit MCMC generating procedure required for generating negative samples $x \sim p_B(x)$ implicitly during training.

### 2.2    Particle Flow

Particle flow, introduced by Daum & Huang (2007), is a class of nonlinear Bayesian filtering (sequential inference) methods designed to approximate the posterior distribution $p(x_t \mid \bar{x}_{\leq t})$ of the sampling process given observations. While closely related to normalizing flows (Rezende & Mohamed, 2015) and neural ordinary differential equations (ODEs) (Chen et al., 2018), these frameworks do not explicitly accommodate a Bayes update. Instead, particle flow achieves Bayes update $p(x_t \mid \bar{x}_{\leq t}) \propto p(x_t \mid \bar{x}_{<t}) \, p(\bar{x}_t \mid x_t, \bar{x}_{<t})$ by transporting the prior samples $x_t \sim p(x_t \mid \bar{x}_{<t})$ through an ODE $\frac{dx}{dt} = v(x, t)$ parameterized by a velocity field $v(x, t)$, over pseudo-time $t \in [0, 1]$. The velocity field is designed such that the sample density follows a log-homotopy that induces the Bayes update. Despite its effectiveness in time-series inference (Pal et al., 2021b; Chen et al., 2019b; Yang et al., 2014) and its robustness against the curse of dimensionality (Surace et al., 2019), particle flow, particularly potential flow where the velocity field $v(x, t) = \Phi(x, t)$ is the gradient of potential energy, remains largely unexplored in energy-based generative modeling.

### 2.3    Deep Ritz Method

The Deep Ritz method is a deep learning-based variational numerical approach, originally proposed by E & Yu (2018), for solving scalar elliptic partial differential equations (PDEs) in high dimensions. Consider the following Poisson's equation, fundamental to many physical models:

$$\begin{aligned} \Delta_x u(x) &= \Gamma(x), & x \in \Omega \\ u(x) &= 0, & x \in \partial\Omega \end{aligned} \tag{2}$$

where $\Delta_x$ is the Laplace operator, and $\partial\Omega$ denotes the boundary of $\Omega$. For a Sobolev function $u \in \mathcal{H}_0^1(\Omega)$ (definition in Proposition 2) and square-integrable $\Gamma \in L^2(\Omega)$, the variational principle ensures that a weak

solution of the Euler-Lagrange boundary value equation (2) is equivalent to the variational problem of minimizing the Dirichlet energy (Müller & Zeinhofer, 2019), as follows:

$$u = \arg\min_v \int_\Omega \left( \frac{1}{2} \|\nabla_x v(x)\|^2 - \Gamma(x)\, v(x) \right) dx + \eta \int_{\partial\Omega} v(x)^2 \ dx \tag{3}$$

where $\nabla_x$ denotes the Del operator (gradient). In particular, the Deep Ritz method parameterizes the trial function $v$ using neural networks and performs the optimization (3) via stochastic gradient descent. To enforce the Dirichlet boundary condition, the second component of the Dirichlet energy (3), weighted by a positive constant $\eta$, must be evaluated on the boundary $\partial\Omega$. This necessitates acquiring additional boundary samples $x \in \partial\Omega$ during neural network training, thereby introducing extra computational overhead. The Deep Ritz method is predominantly applied for finite element analysis (Liu et al., 2023a) due to its versatility and effectiveness in handling high-dimensional PDE systems. In (Olmez et al., 2020), the Deep Ritz method is employed to solve the density-weighted Poisson equation arising from the feedback particle filter (Yang et al., 2013). However, its application in generative modeling remains unexplored.

## 3   Variational Potential (VAPO) Flow Bayes

In this section, we introduce VAPO, a novel energy-based generative modeling framework inspired by particle flow and the Deep Ritz method. VAPO encompasses four key elements: constructing a Bayesian marginal homotopy between the Gaussian prior and data likelihood (Section 3.1), designing a potential flow that aligns the flow-driven homotopy with the marginal homotopy (Section 3.2), formulating a variational loss function using the Deep Ritz method (Section 3.4), and establishing connections between homotopy matching, diffusion, and EBMs (Section 3.3).

### 3.1   Interpolating Between Prior and Data Likelihood: Log-Homotopy Bayesian Transport

Let $\bar{x} \in \Omega$ denote the training data, with likelihood $p_{\text{data}}(\bar{x})$, and let $x \in \Omega$ represent the generative samples. First, we define a Gaussian prior $q(x) = \mathcal{N}(0, \omega^2 I)$ and a Gaussian conditional data likelihood $p(\bar{x} \mid x) = \mathcal{N}(\bar{x}; x, \nu^2 I)$, both with isotropic covariances. This data likelihood satisfies the state space model $x = \bar{x} + \nu\,\epsilon$, where $\epsilon$ is the standard Gaussian noise. The standard deviation $\nu$ is usually set to be small so that $x$ closely resembles $\bar{x}$. The aim of flow-based generative modeling is to learn a density homotopy (path) interpolating between the prior and the data likelihood for generative modeling. On that account, consider the following conditional (data-conditioned) probability density log-homotopy $\rho : \Omega^2 \times [0, 1] \to \mathbb{R}$:

$$\rho(x \mid \bar{x}, t) = \frac{e^{h(x \mid \bar{x}, t)}}{\int_\Omega e^{h(x \mid \bar{x}, t)}\ dx} \tag{4}$$

where $h : \Omega^2 \times [0, 1] \to \mathbb{R}$ is a log-linear function:

$$h(x \mid \bar{x}, t) = \alpha(t)\, \log q(x) + \beta(t)\, \log p(\bar{x} \mid x) \tag{5}$$

where $\alpha : [0, 1] \to [0, 1]$ and $\beta : [0, 1] \to [0, 1]$ are both monotonically increasing functions parameterized by time $t$. The following proposition shows that this log-homotopy transformation results in a Gaussian perturbation kernel.

**Proposition 1.** *Consider a Gaussian prior $q(x) = \mathcal{N}(x; 0, \omega^2 I)$ and a conditional data likelihood $p(\bar{x} \mid x) = \mathcal{N}(\bar{x}; x, \nu^2 I)$. The log-homotopy transport (4) corresponds to a Gaussian perturbation kernel $\rho(x \mid \bar{x}, t) = \mathcal{N}(x; \mu(t)\,\bar{x}, \sigma(t)^2 I)$, characterized by the time-varying mean and standard deviation:*

$$\mu(t) = \text{sigmoid}\left( \log\left( \frac{\beta(t)}{\alpha(t)} \frac{\omega^2}{\nu^2} \right) \right), \quad \sigma(t) = \sqrt{\frac{\nu^2}{\beta(t)}\, \mu(t)} \tag{6}$$

*where* $\text{sigmoid}(z) = \frac{1}{1 + e^{-z}}$ *denotes the logistic (sigmoid) function.*

*Proof.* Refer to Appendix C.1. ☐

Hence, the density homotopy equation 4 represents a tempered Bayesian transport mapping from the Gaussian prior $q(x)$ to the posterior kernel

$$\rho(x \mid \bar{x}, 1) = \frac{e^{h(x|\bar{x},1)}}{\int_\Omega e^{h(x|\bar{x},1)} \, dx} = \frac{p(\bar{x} \mid x) \, q(x)}{\int_\Omega p(\bar{x} \mid x) \, q(x) \, dx} = p(x \mid \bar{x}) \tag{7}$$

which is the maximum a posteriori estimation centered on discrete data samples. To approximate the intractable data likelihood, we can then consider the following marginal probability density homotopy:

$$\bar{\rho}(x,t) = \int_\Omega p_{\text{data}}(\bar{x}) \, \rho(x \mid \bar{x}, t) \, d\bar{x}, \tag{8}$$

where it remains that $p(x,0) = q(x)$, and we have $\bar{\rho}(x,1) = \int_\Omega p_{\text{data}}(\bar{x}) \, p(x \mid \bar{x}) \, d\bar{x} = p(x)$. Therefore, this marginal homotopy defines a data-recovery path interpolation between the Gaussian prior $q(x)$ and the approximate data likelihood $p(x)$. In particular, $p(x)$ represents a Bayesian approximation of the true data likelihood, by convolving the discrete data likelihood $p_{\text{data}}(\bar{x})$ with the posterior distribution $p(x \mid \bar{x})$. Nevertheless, the marginalization in (8) is intractable, thereby precluding a closed-form solution for the marginal homotopy. To overcome this challenge, we propose a potential flow-driven density homotopy, whose time evolution is aligned with this data-recovery marginal homotopy.

### 3.2 Modeling Potential Flow in a Data-Recovery Homotopy Landscape

Our goal is to model a potential flow whose density evolution aligns with the marginal homotopy, thereby directing samples toward the data likelihood. We begin by deriving the time evolution of the marginal homotopy in the following proposition.

**Proposition 2.** *Consider the conditional homotopy $\rho(x \mid \bar{x}, t)$ in (4) with Gaussian conditional data likelihood $p(\bar{x} \mid x) = \mathcal{N}(\bar{x}; x, \nu^2 I)$. Then, the time evolution (derivative) of the marginal homotopy $\bar{\rho}(x,t)$ is given by the following partial differential equation (PDE):*

$$\frac{\partial \bar{\rho}(x,t)}{\partial t} = -\frac{1}{2} \, \mathbb{E}_{p_{data}(\bar{x})} \left[ \rho(x \mid \bar{x}, t) \left( \gamma(x, \bar{x}, t) - \bar{\gamma}(x, \bar{x}, t) \right) \right] \tag{9}$$

*where $\gamma$ denotes the innovation term*

$$\gamma(x, \bar{x}, t) = \frac{\dot{\alpha}(t)}{\omega^2} \, \|x\|^2 + \frac{\dot{\beta}(t)}{\nu^2} \, \|x - \bar{x}\|^2 \tag{10}$$

*Here, $\dot{\alpha}(t)$ and $\dot{\beta}(t)$ denote the time-derivatives, and $\bar{\gamma}(x, \bar{x}, t) = \mathbb{E}_{\rho(x|\bar{x},t)}[\gamma(x, \bar{x}, t)]$ denotes the expectation.*

*Proof.* Refer to Appendix C.2. $\qquad\square$

A potential flow involves subjecting the prior samples to an energy-generated velocity field, where their trajectories $(x(t))$ satisfy the following ODE:

$$\frac{dx(t)}{dt} = \nabla_x \Phi(x,t) \tag{11}$$

where $\Phi : \Omega \times [0,1] \to \mathbb{R}$ is a scalar potential energy, and $\nabla_x$ denotes the Del operator (gradient) with respect to the data samples $x(t)$. The vector field $\nabla_x \Phi \in \Omega$ represents the divergence (irrotational) component in the Helmholtz decomposition. By incorporating this potential flow, the flow-driven density homotopy $\rho_\Phi(x,t)$ evolves via the continuity equation (Gardiner, 2009):

$$\frac{\partial \rho_\Phi(x,t)}{\partial t} = -\nabla_x \cdot \left( \rho_\Phi(x,t) \, \nabla_x \Phi(x,t) \right) \tag{12}$$

which corresponds to the transport equation for modeling fluid advection. Our aim is to model the potential energy such that the evolution of the prior density under the potential flow emulates the evolution of the

marginal homotopy. In other words, we seek to achieve homotopy matching, $\rho_\Phi \equiv \bar{\rho}$, by aligning their respective time evolutions as described in (9) and (12). This leads to the following PDE, which takes the form of a density-weighted Poisson equation:

$$\nabla_x \cdot \left( \rho_\Phi(x, t) \, \nabla_x \Phi(x, t) \right) = \frac{1}{2} \, \mathbb{E}_{p_{\text{data}}(\bar{x})} \left[ \rho(x \mid \bar{x}, t) \left( \gamma(x, \bar{x}, t) - \bar{\gamma}(x, \bar{x}, t) \right) \right] \tag{13}$$

However, this Poisson equation remains intractable due to the lack of a closed-form expression for $\rho_\Phi$. To overcome this limitation, we substitute the intractable $\rho_\Phi$ with the target marginal homotopy $\bar{\rho}$, enabling direct sampling and a variational principle approach. In the following proposition, we demonstrate that the revised Poisson's equation minimizes the KL divergence between the flow-driven and conditional homotopies, yielding statistically optimal homotopy matching.

**Proposition 3.** *Consider a potential flow of the form (11) and given that $\Phi \in \mathcal{H}_0^1(\Omega, p)$, where $\mathcal{H}_0^n$ denotes the (Sobolev) space of n-times differentiable functions that are compactly supported, and square-integrable with respect to marginal homotopy $\bar{\rho}(x, t)$. Solving for the potential energy $\Phi(x)$ that satisfies the following density-weighted Poisson's equation:*

$$\nabla_x \cdot \left( \bar{\rho}(x, t) \, \nabla_x \Phi(x, t) \right) = \frac{1}{2} \, \mathbb{E}_{p_{data}(\bar{x})} \left[ \rho(x \mid \bar{x}, t) \left( \gamma(x, \bar{x}, t) - \bar{\gamma}(x, \bar{x}, t) \right) \right] \tag{14}$$

*is then equivalent to minimizing the KL divergence $\mathcal{D}_{\text{KL}} \left[ \rho_\Phi(x, t) \| \bar{\rho}(x, t) \right]$ between the flow-driven homotopy and the conditional homotopy.*

*Proof.* Refer to Appendix C.3. $\qquad\square$

Therefore, solving this density-weighted Poisson's equation corresponds to performing a homotopy matching $\rho_\Phi \equiv \bar{\rho}$. In the following section, we demonstrate that this homotopy matching gives rise to a Boltzmann energy expressed in terms of the potential energy $\Phi$ when the marginal homotopy $\bar{\rho}$ reaches its stationary equilibrium, thereby establishing a connection between our proposed potential flow framework and EBMs.

### 3.3 Connections to Diffusion Process and Energy-Based Modeling

In this section, we clarify the relationship between diffusion models and flow matching within the homotopy matching framework. Building on this insight, we establish a link between our proposed potential flow framework and energy-based modeling.

First, we present results from diffusion models. It has been outlined in Song et al. (2021) that the conditional density homotopy, represented by the Gaussian perturbation kernel $\rho(x \mid \bar{x}, t) = \mathcal{N}\left(x; \mu(t) \, \bar{x}, \sigma(t)^2 I\right)$, characterizes a diffusion process governed by the following stochastic differential equation (SDE):

$$dx(t) = -f(t) \, x(t) \, dt + g(t) \, dW(t) \tag{15}$$

where $W(t) \in \mathbb{R}^n$ denote the standard Wiener process. Note that the time parameterization with respect to $t$ here is the reverse of the conventional parameterization used in diffusion models, where the diffusion process transitions from $x(1) \sim p_{\text{data}}(\bar{x})$ to $x(0) \sim q(x) = \mathcal{N}(0, \omega^2 I)$ as defined in Section 3.1. In addition, the time-varying drift $f : [0, 1] \to \mathbb{R}$ and diffusion $g : [0, 1] \to \mathbb{R}$ coefficients are shown by Karras et al. (2022) to be given by

$$f(t) = -\frac{\dot{\mu}(t)}{\mu(t)}, \quad g(t) = -\sqrt{2 \, \sigma(t) \left( \dot{\sigma}(t) + f(t) \, \sigma(t) \right)} \tag{16}$$

where $\dot{\mu}(t)$ and $\dot{\sigma}(t)$ denote the time-derivatives. It has also been shown in Song et al. (2021) that the following deterministic probability flow ODE:

$$\frac{dx(t)}{dt} = -f(t) \, x(t) + \frac{1}{2} \, g(t)^2 \, \nabla_x \log \bar{\rho}(x, t) \tag{17}$$

results in the same marginal probability homotopy $\bar{\rho}(x, t)$ as the forward-time diffusion SDE (16). Subsequently, we highlight the link between the diffusion process and the vector field modeled in flow matching.

**Proposition 4.** *The conditional vector field in flow matching (Lipman et al., 2023), given by*

$$\frac{dx(t)}{dt} = v(x \mid \bar{x}, t) = \dot{\mu}(t)\,\bar{x} + \dot{\sigma}(t)\,\epsilon \tag{18}$$

*with standard Gaussian noise $\epsilon \sim \mathcal{N}(0, I)$, satisfies the conditional probability flow ODE governing the diffusion process conditioned on boundary condition $x(1) \sim p_{data}(\bar{x})$. It follows that the marginal vector field, given by the law of iterated expectation (tower property) $\mathbb{E}[U|X=x] = \mathbb{E}[\mathbb{E}[U \mid X=x, Y] \mid X=x]$:*

$$\frac{dx(t)}{dt} = v(x, t) = \mathbb{E}_{p_{data}(\bar{x}|x)}\big[v(x \mid \bar{x}, t) \mid x\big] = \int_\Omega v(x \mid \bar{x}, t)\,\frac{\rho(x \mid \bar{x}, t)\,p_{data}(\bar{x})}{\bar{\rho}(x, t)}\,d\bar{x} \tag{19}$$

*also satisfies the marginal probability flow ODE (17).*

*Proof.* Refer to Appendix C.4. □

Building on this result, we establish a connection between the proposed potential flow framework and EBMs. The following proposition demonstrates that homotopy matching, e.g., $\rho_\Phi \equiv \bar{\rho}$ leads to an energy-parameterized Boltzmann equilibrium.

**Proposition 5.** *Given that the flow-driven homotopy $\rho_\Phi(x, t)$ matches the data-recovery marginal homotopy $\bar{\rho}(x, t)$, they exhibit the same Fokker–Planck dynamics. As the time-varying marginal density $\bar{\rho}(x, t)$ converges to its stationary equilibrium $\bar{\rho}_\infty(x)$, i.e., when $\frac{\partial \bar{\rho}(x,t)}{\partial t} \to 0$, the Fokker-Planck dynamics reach the Boltzmann distribution (1), where the Boltzmann energy $\Phi_B(x)$ is defined as follows:*

$$\Phi_B(x) = \frac{4\,\Phi_\infty(x) + f_\infty\,\|x\|^2}{g_\infty^2} \tag{20}$$

*where $\Phi_\infty(x)$, $f_\infty$, and $g_\infty$ denote the steady-state potential energy, drift, and diffusion coefficients, respectively, associated with the stationary equilibrium.*

*Proof.* Refer to Appendix C.5. □

On that note, we uncover the connection between the proposed VAPO framework and EBMs, demonstrating the validity of the potential energy as a parameterization of a Boltzmann energy. This holds provided that $\bar{\rho}$ converges to its stationary equilibrium and $\rho_\Phi$ learns to match these convergent dynamics. In the following section, we introduce a variational principle approach to solving the density-weighted Poisson equation (14), thereby addressing the intractable homotopy matching problem.

### 3.4 Variational Potential Energy Loss Formulation: Deep Ritz Method

Solving the density-weighted Poisson's equation (14) is particularly challenging in high-dimensional settings. Numerical approximation struggles to scale with higher dimensionality, as selecting suitable basis functions, such as in the Galerkin approximation, becomes increasingly complex (Yang et al., 2016). Similarly, a diffusion map-based algorithm demands an exponentially growing number of particles to ensure error convergence (Taghvaei et al., 2020). To address these challenges, we propose a variational loss function using the Deep Ritz method. This approach casts Poisson's equation as a variational problem compatible with stochastic gradient descent. Consequently, the proposed approach solves Eq. (14), effectively aligning the flow-driven homotopy with the marginal homotopy. Directly solving Poisson's equation (14) is challenging. Therefore, we first consider the following weak formulation:

$$\int_\Omega \frac{1}{2}\,\mathbb{E}_{p_{\text{data}}(\bar{x})}\Big[\rho(x \mid \bar{x}, t)\,\big(\gamma(x, \bar{x}, t) - \bar{\gamma}(x, \bar{x}, t)\big)\Big]\,\Psi\,dx = \int_\Omega \nabla_x \cdot \Big(\bar{\rho}(x, t)\,\nabla_x \Phi(x, t)\big)\Big)\,\Psi\,dx \tag{21}$$

This PDE must hold for all differentiable trial functions $\Psi$. In the following proposition, we introduce a variational loss function that is equivalent to solving this weak formulation of the density-weighted Poisson's equation.

**Proposition 6.** *The variational problem of minimizing the following loss functional:*

$$\mathcal{L}(\Phi, t) = \text{Cov}_{\rho(x|\bar{x},t)\, p_{data}(\bar{x})}\left[\Phi(x,t), \gamma(x,\bar{x},t)\right] + \mathbb{E}_{\bar{\rho}(x,t)}\left[\left\|\nabla_x \Phi(x,t)\right\|^2\right] \tag{22}$$

*with respect to the potential energy $\Phi$, is equivalent to solving the weak formulation (21) of the density-weighted Poisson's equation (14). Here, $\|\cdot\|$ denotes the Euclidean norm, and $\text{Cov}$ denotes the covariance. Furthermore, the variational problem (22) admits a unique solution $\Phi \in \mathcal{H}_0^1(\Omega; \rho)$ if the marginal homotopy $p$ satisfy the Poincaré inequality:*

$$\mathbb{E}_{\bar{\rho}(x,t)}\left[\left\|\nabla_x \Phi(x,t)\right\|^2\right] \geq \eta\, \mathbb{E}_{\bar{\rho}(x,t)}\left[\left\|\Phi(x,t)\right\|^2\right] \tag{23}$$

*for some positive scalar constant $\eta > 0$ (spectral gap).*

*Proof.* Refer to Appendix C.6. ◻

*Remark* 1. The integration by parts in (66) and (87) require that the marginal density $\bar{\rho}(x)$ vanishes on the boundary $\partial\Omega$ of some open, bounded domain $\Omega \subset \mathbb{R}^n$, so that the boundary integral $\int_{\partial\Omega} \bar{\rho}(x)\,(\nabla_x \Phi \cdot \hat{n})\, dx = 0$ holds. In standard implementations, although the training data $\bar{x}$ are typically normalized to lie within $[-1,1]^n$, we may define the perturbed samples as $x \in \Omega \subset \mathbb{R}^n$, where $\Omega$ is chosen to contain the support of the data distribution. Accordingly, the open bounded domain $\Omega$ can be defined sufficiently large so that the conditional homotopy $\rho(x \mid \bar{x}, t)$ approaches zero at the boundary $\partial\Omega$. Since $\rho(x \mid \bar{x}, t)$ is a Gaussian perturbation kernel, it decays exponentially and is effectively negligible near the boundary, thereby satisfying the required condition. As a result, the marginal distribution $\bar{\rho}(x, t)$ also vanishes at $\partial\Omega$, ensuring the validity of the integration by parts required to formulate both Proposition 3 Proposition 6.

Overall, Propositions 3 and 6 recast the intractable problem of minimizing the KL divergence between the flow-driven homotopy and the marginal homotopy as an equivalent variational problem of solving the loss function (22). By optimizing the potential energy with respect to this loss and transporting the prior samples through the ODE (11), the prior particles evolve along a trajectory that aligns with the marginal homotopy. In particular, the covariance loss here plays an important role by ensuring that the normalized innovation (residual sum of squares) is inversely proportional to the potential energy. As a result, the energy-generated velocity field $\nabla_x \Phi$ consistently points in the direction of greatest potential ascent, thereby driving the flow of prior particles towards high likelihood regions. Given that homotopy matching is performed over the entire time horizon, we apply stochastic integration to the loss function over time, where $t \sim \mathcal{U}(0, t_{\text{end}})$ is drawn from the uniform distribution.

### 3.5 Training Implementation

In our implementation, we adopt the Optimal Transport Flow Matching (OT-FM) framework (Lipman et al., 2023) for training, where it corresponds to the SDE parameterization $f(t) = -\frac{1}{t}$, $g(t) = \sqrt{\frac{2\,(1-t)}{t}}$ as outline in (Kingma & Gao, 2023), or equivalently $\alpha(t) = \frac{\omega^2}{1-t}$, $\beta(t) = \frac{\nu^2 t}{(1-t)^2}$ for the log-homotopy transformation derived in (5). To establish a Boltzmann equilibrium, we further require $\frac{\partial \bar{\rho}(x,t)}{\partial t} \to 0$ so that the time-varying marginal density $\bar{\rho}(x, t)$ converges to the stationary Boltzmann distribution. However, the Gaussian perturbation kernel $\rho(x \mid \bar{x}, t) = \mathcal{N}\left(\mu(t)\,\bar{x}, \sigma(t)^2 I\right)$ employed by the flow-based probability paths is defined only over a finite time interval $t \in [0, t_{\max}]$. Additionally, the marginal homotopy $\bar{\rho}(x, t)$ is not guaranteed to reach equilibrium within this prescribed time window.

To resolve these limitations of the flow-based probability paths, we explicitly enforce stationarity in our training implementation, by imposing a steady-state equilibrium $p_\infty(x) = \bar{\rho}(x, t \geq t_{\max})$ beyond some cutoff time $t_{\max} < t_{\text{end}}$ close to the terminal time. This steady-state equilibrium $p_\infty(x) \equiv p_B(x)$ thus corresponds to the stationary Boltzmann distribution, parameterized by the energy function derived in (20) with $f_\infty = f(t_{\max})$ and $g_\infty = g(t_{\max})$. Given that a steady-state equilibrium is enforced via $p_\infty(x) = \bar{\rho}(x, t \geq t_{\max}) \approx p_{\text{data}}(\bar{x})$, the stationary Boltzmann distribution approximates the true data likelihood by design.

---
**Algorithm 1** VAPO Training

---
**input:** Initial model parameters $\theta$, mean and standard deviation scheduling functions $\mu(t)$, $\sigma(t)$, cutoff time $t_{\max}$, terminal time $t_{\mathrm{end}}$, decay exponent $\kappa$, spectral gap constant $\eta$, and batch size $B$.
**repeat**
    Sample observed data $\bar{x}_i \sim p_{\mathrm{data}}(\bar{x})$, $t_i \sim \mathcal{U}(0, t_{\mathrm{end}})$, and $\epsilon_i \sim \mathcal{N}(0, I)$
    Set $t_i = \min(t_i, t_{\max})$ and sample $x_i \sim \rho(x \mid \bar{x}, t_i)$ via reparameterization $x_i = \mu(t_i)\,\bar{x}_i + \sigma(t_i)\,\epsilon_i$
    Compute gradient $\nabla_x \Phi_\theta(x_i, t_i)$ w.r.t. $x_i$ via backpropagation
    Calculate VAPO loss $\frac{1}{B} \sum_{i=1}^{B} \mathcal{L}(\Phi_\theta, t_i)$, backpropagate and update model parameters $\theta$
**until** convergence

---

Finally, our VAPO loss function is implemented as follows:

$$\mathcal{L}^{\mathrm{VAPO}}(\Phi) = \int_0^{t_{\mathrm{end}}} \mathcal{L}(\Phi, t)\, dt = \mathbb{E}_{\mathcal{U}(0, t_{\mathrm{end}})}\left[\mathcal{L}(\Phi, t)\right] \tag{24}$$

where

$$\begin{aligned}
\mathcal{L}(\Phi, t) = \ &\mathrm{Cov}_{\rho(x\mid\bar{x},t)\, p_{\mathrm{data}}(\bar{x})}\left[\Phi(x,t), w(t)\,\gamma(x,\bar{x},t)\right] - \frac{\nabla_x \Phi(x,t) \cdot v(x \mid \bar{x}, t)}{\left\|\nabla_x \Phi(x,t)\right\| \left\|v(x \mid \bar{x}, t)\right\|} \\
&+ \mathbb{E}_{\rho(x\mid\bar{x},t)\, p_{\mathrm{data}}(\bar{x})}\left[\left\|\nabla_{(x,t)}\Phi(x,t)\right\|^2 + \eta \left\|\Phi(x,t)\right\|^2\right]
\end{aligned} \tag{25}$$

Here, we incorporate an additional cosine distance between the potential gradient $\nabla_x \Phi$ and the conditional vector field in (18) to the loss function. While this cosine distance does not influence the learning of the potential energy's magnitude (magnitude learning is entirely supervised by the covariance loss), it enforces directional alignment between the gradient and the vector field. To further enforce convergence toward a steady-state potential $\Phi_\infty(x)$, we additionally encourage quasi-static dynamics by minimizing the Euclidean norm of the time derivative $\left|\frac{\partial \Phi}{\partial t}\right|^2$ alongside the gradient norm during training. Also, a weighting $w(t) = (1-t)^\kappa$ with decay exponent $\kappa > 1$ is applied to the innovation term to balance the covariance loss across time to stabilize training.

Considering that the marginal homotopy may not satisfy the Poincaré inequality (23), we include the right-hand side of this inequality in the loss function to enforce the uniqueness of the minimizer. To empirically validate the existence of a positive Poincaré constant $\eta$, Figure 11 plots the ratio between the mean gradient norm $\mathbb{E}\left[\|\nabla_x \Phi\|^2\right]$ and the mean energy norm $\mathbb{E}\left[\|\Phi\|^2\right]$ over training iterations on CIFAR-10, without applying the additional Poincaré regularization loss. It shows that the ratio is bounded below by $\eta = 6.81 \times 10^{-5}$, thereby confirming the existence of a positive Poincaré constant during training.

Nonetheless, our experiments indicate that the existence and magnitude of such an unenforced Poincaré constant vary across different neural architectures. For completeness, we incorporate the Poincaré regularization with a small $\eta$ for both the WideResNet and U-Net models, which we fine-tune during training for optimal results. The cutoff time $t_{\max}$, terminal time $t_{\mathrm{end}}$, decay exponent $\kappa$, and spectral gap constant $\eta$ are hyperparameters to be determined during training. Algorithm 1 summarizes the training procedure of our proposed VAPO framework.

## 4  Experiments

In this section, we validate the energy-based generative modeling capabilities of VAPO across several key tasks. Section 4.1 explores 2D density estimation. Section 4.2 presents the unconditional generation and spherical interpolation results on CIFAR-10 and CelebA. Section 4.4 evaluates mode coverage and model generalization through energy histograms of train and test data and the nearest neighbors of generated samples. Section 4.5 examines unsupervised OOD detection performance on various datasets. Section 4.6 verifies the convergence of long-run ODE samples to a Boltzmann equilibrium. Additional results on ablation study and computational efficiency are provided in Appendix A. Additional discussions of the results are also provided in Appendix B. Finally, implementation details, including architecture, training, numerical solvers, datasets, and FID evaluation, are provided in Appendix D.

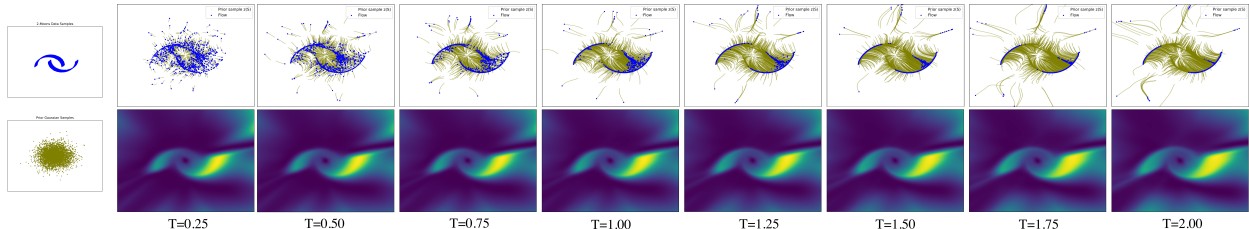

Figure 1: 2D potential flow. Top: Sample trajectories from the Gaussian prior noise distribution (black) to the target 2-Moons distribution (blue), driven by the potential energy $\Phi(x,t)$ and sampled using an ODE solver. Bottom: Time evolution of the learned potential energy landscape $\Phi(x,t)$.

## 4.1 Density Estimation on 2D Data

To verify the convergence properties of the potential energy and to assess the validity of the Boltzmann energy (20), we conduct density estimation on 2D synthetic datasets. Specifically, we learn a potential flow that transforms an unimodal Gaussian prior distribution into a 2-Moons target distribution. Figure 1 shows the sample trajectories driven by the potential flow $dx(t) = \nabla_x \Phi(x,t)\,dt$, obtained via the deterministic Euler solver. Figure 2 presents the sample trajectories and density estimation of the Boltzmann distribution $p_B \propto e^{\Phi_B(x)}$, obtained via the Stochastic Gradient Langevin Dynamics (SGLD). Notably, both the potential energy $\Phi(x)$ and the Boltzmann energy $\Phi_B(x)$ exhibit stable convergence toward their steady-state equilibrium. Furthermore, the results indicate that the estimated Boltzmann density closely aligns with the ground-truth 2-Moons distribution. These results highlight the effectiveness of our variational principle approach in learning the Boltzmann stationary distribution through homotopy matching against the stationary-enforced marginal $p_\infty(x)$.

Nonetheless, a standard formulation of the forward-time SDE (15), or equivalently, the marginal probability flow ODE (17), is valid only in the case of a unimodal Gaussian prior, e.g., $q(x) = \rho(x \mid \bar{x}, t = 0) = \mathcal{N}(0, \omega^2 I)$, as discussed in Kingma & Gao (2023). This assumption underpins the consistency of the Fokker–Planck dynamics with the continuous-time diffusion framework, ensuring the validity of the stationary Boltzmann energy in the limit. We acknowledge this limitation of our current framework. As a direction for future work, we propose extending the forward-time SDE or ODE formulation of continuous-time diffusion to be admissible for more general prior distributions, such as mixtures of Gaussians or learned priors, to accommodate multi-modal data while maintaining consistency with our proposed energy-based framework.

## 4.2 Unconditional Image Generation

For image generation, we consider three VAPO model variants: an autonomous (independent of time) energy model $\Phi(x)$ parameterized by Zagoruyko & Komodakis (2016), and a time-varying energy model $\Phi(x,t)$ parameterized by U-Net (Ronneberger et al., 2015). Figure 3 shows the uncurated and unconditional

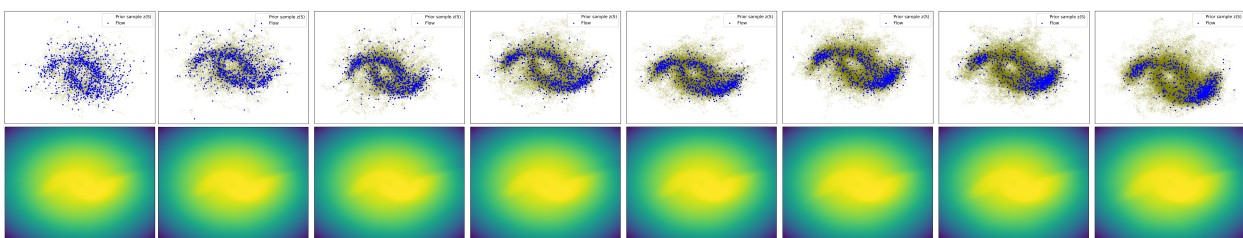

Figure 2: 2D Boltzmann density estimation. Top: Sample trajectories from the Gaussian prior noise distribution (black) to the target 2-Moons distribution (blue), driven by the Boltzmann energy and sampled via SGLD. Middle: Visualization of the log-density estimation (up to an additive constant) $\log p_B(x) = \Phi_B(x)$ parameterized by Boltzmann energy.

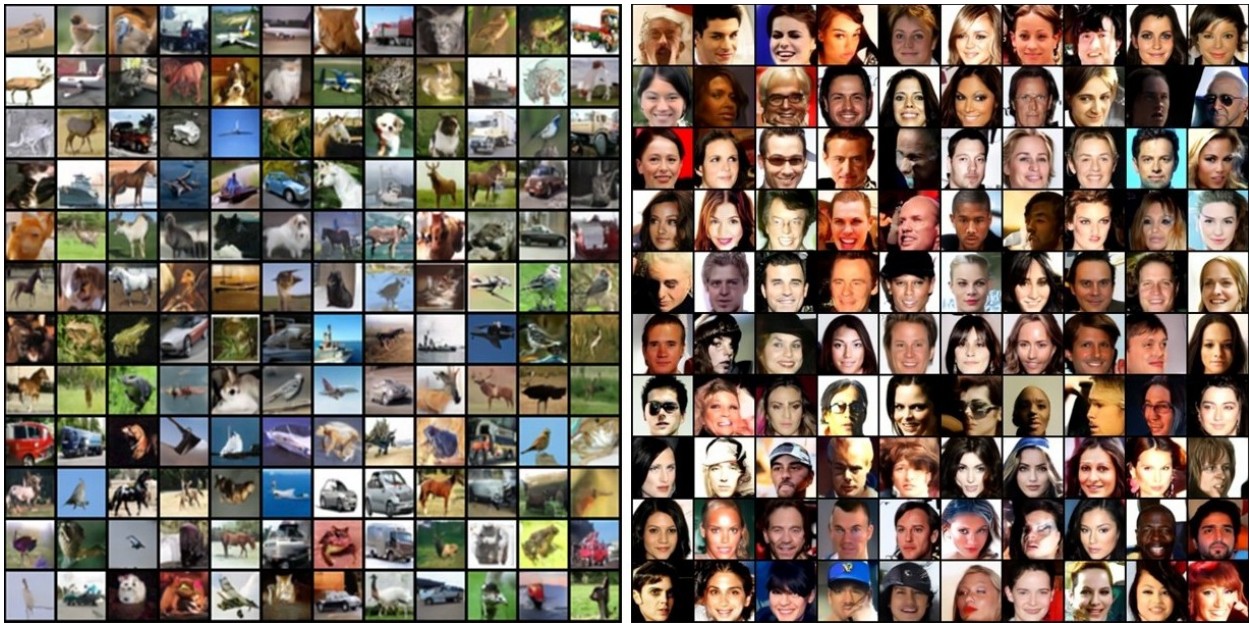

Figure 3: Uncurated and unconditional samples generated for CIFAR-10 (left) and CelebA (right).

| Male and Young | Male and Smile | Young and Smile |
| :---: | :---: | :---: |

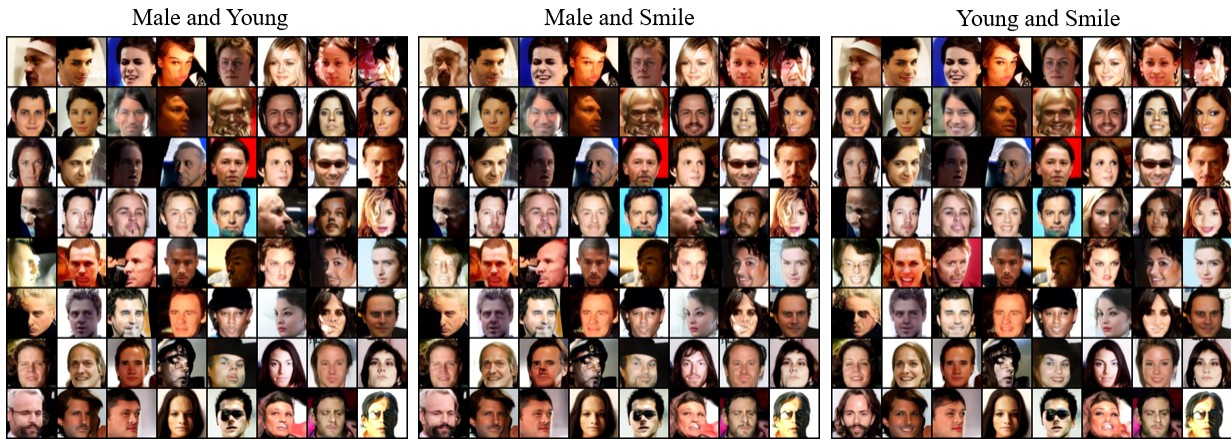

Figure 4: Compositional and conditional CelebA samples generated based on three attribute pairs.

image samples generated using the time-varying energy model on CIFAR-10 $32 \times 32$ and CelebA $64 \times 64$. The generated samples are of decent quality and resemble the original datasets, despite not having the highest fidelity as achieved by state-of-the-art models. Table 1 summarizes the quantitative evaluations of our framework in terms of FID (Heusel et al., 2017) scores on the CIFAR-10. In particular, the VAPO models achieved FID scores competitive to existing generative models. Figures 7 and 8 show additional uncurated samples of unconditional image generation on CIFAR-10 and CelebA, respectively.

## 4.3 Image Interpolation and Compositional Generation

To achieve smooth and semantically coherent image interpolation, we perform spherical interpolation between two Gaussian noises and subsequently apply ODE sampling to the interpolated noises. Figures 9 and 10 show additional interpolation results on CIFAR-10 and CelebA, respectively. For compositional sample generation, we first train a class-conditioned energy model $\Phi(x, c)$, and then sample by averaging the conditional energies across selected classes. Figure 4 presents compositional generation results conditioned on composite CelebA

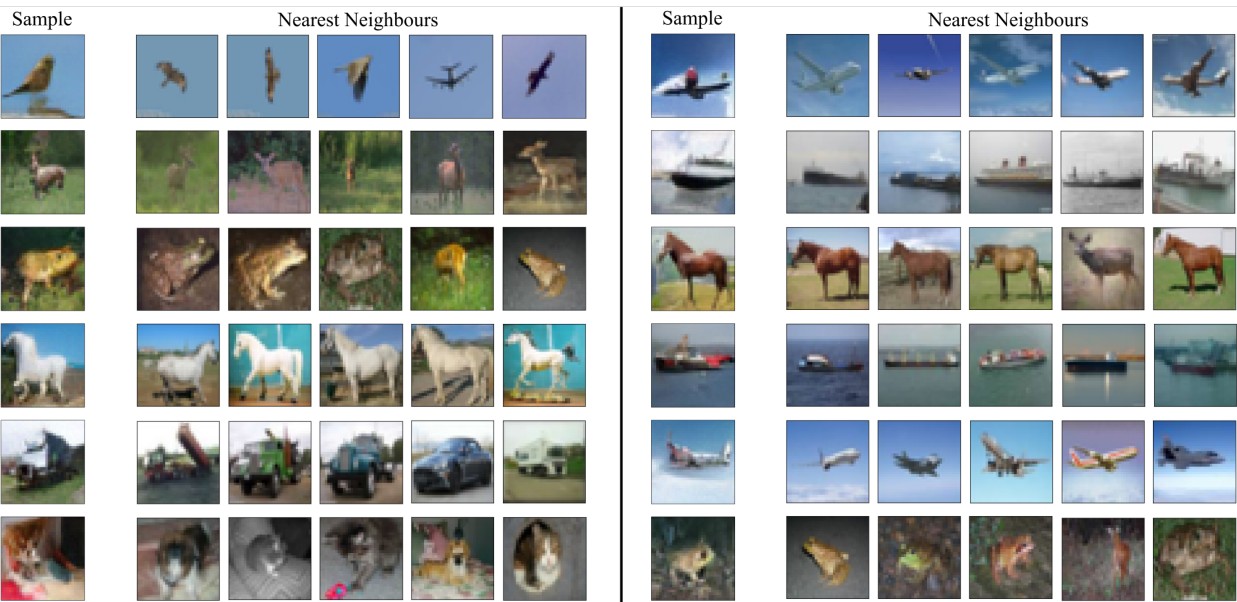

Figure 5: Generated CIFAR-10 samples and their five nearest neighbors in train set based on pixel distance.

attributes, specifically (*Male, Young*), (*Male, Smile*), and (*Young, Smile*). However, certain compositional samples show limited variation across attribute pairs, suggesting that incorporating composition weights could improve attribute-specific conditioning.

## 4.4 Model Generalization and Mode Evaluation

To evaluate the generalization capability of the VAPO model, Figure 5 presents the nearest neighbors of the generated samples in the CIFAR-10 training set. The results show that nearest neighbors differ significantly from the generated samples, suggesting that our model does not overfit the training data and generalizes well to the underlying data distribution. To validate the mode coverage and over-fitting ability, Figure 6 presents a histogram of the CIFAR-10 training and test datasets across the potential energy estimated by VAPO. The histogram shows that the learned energy model assigns similar energy values to images from both sets. This indicates that the VAPO model generalizes well to unseen test data while maintaining broad mode coverage of the training distribution.

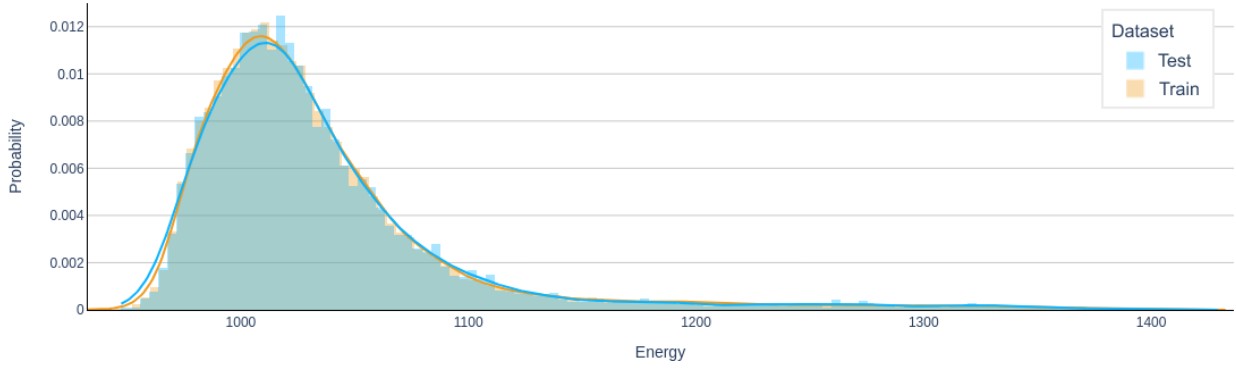

Figure 6: Histogram of the CIFAR-10 training and test datasets across model-parameterized potential energy.

Table 1: FID scores on unconditional CIFAR-10 image generation.

| Energy-based Models | FID ↓ | Other Likelihood-based Models | FID ↓ |
|---|---|---|---|
| EBM-IG (Du & Mordatch, 2019) | 38.2 | ResidualFlow (Chen et al., 2019a) | 47.4 |
| EBM-FCE (Gao et al., 2020) | 37.3 | Glow (Kingma & Dhariwal, 2018) | 46.0 |
| CoopVAEBM (Xie et al., 2021b) | 36.2 | DC-VAE (Parmar et al., 2021) | 17.9 |
| CoopNets (Xie et al., 2020) | 33.6 | **GAN-based Models** | |
| Divergence Triangle (Han et al., 2019) | 30.1 | SN-GAN (Miyato et al., 2018) | 21.7 |
| VERA (Grathwohl et al., 2021) | 27.5 | SNGAN-DDLS Che et al. (2020) | 15.4 |
| EBM-CD (Du et al., 2021) | 25.1 | BigGAN (Brock et al., 2019) | 14.8 |
| GEBM (Arbel et al., 2021) | 19.3 | **Score-based and Diffusion Models** | |
| HAT-EBM (Hill et al., 2022) | 19.3 | NCSN-v2 (Song & Ermon, 2020) | 10.9 |
| CF-EBM (Zhao et al., 2020) | 16.7 | DDPM Distil (Luhman et al., 2021) | 9.36 |
| CoopFlow (Xie et al., 2022) | 15.8 | DDPM (Ho et al., 2020) | 3.17 |
| VAEBM (Xiao et al., 2021a) | 12.2 | NCSN++ (Song et al., 2021) | 2.20 |
| DRL (Gao et al., 2021) | 9.58 | **Flow-based Models** | |
| CLEL (Lee et al., 2022) | 8.61 | Action Matching (Neklyudov et al., 2023) | 10.0 |
| DDAEBM (Geng et al., 2024) | 4.82 | Flow Matching (Lipman et al., 2023) | 6.35 |
| CDRL (Zhu et al., 2024) | 3.68 | Rectified Flow (Liu et al., 2023b) | 4.85 |
| VAPO (Autonomous) | **14.5** | DSBM (Shi et al., 2023) | 4.51 |
| VAPO (Time-varying) | **6.72** | PFGM (Xu et al., 2022) | 2.35 |

Table 2: AUROC scores ↑ for OOD detection on several datasets.

| Models | CIFAR-10 Interpolation | CIFAR-100 | CelebA | SVHN |
|---|---|---|---|---|
| PixelCNN (Salimans et al., 2017) | 0.71 | 0.63 | - | 0.32 |
| GLOW (Kingma & Dhariwal, 2018) | 0.51 | 0.55 | 0.57 | 0.24 |
| NVAE (Vahdat & Kautz, 2020) | 0.64 | 0.56 | 0.68 | 0.42 |
| EBM-IG (Du & Mordatch, 2019) | 0.70 | 0.50 | 0.70 | 0.63 |
| VAEBM (Xiao et al., 2021a) | 0.70 | 0.62 | 0.77 | 0.83 |
| CLEL (Lee et al., 2022) | 0.72 | 0.72 | 0.77 | 0.98 |
| DRL (Gao et al., 2021) | - | 0.44 | 0.64 | 0.88 |
| CDRL (Zhu et al., 2024) | 0.75 | 0.78 | 0.84 | 0.82 |
| VAPO (Ours) | 0.78 | 0.67 | 0.84 | 0.61 |

## 4.5 Out-of-Distribution Detection

Given that the potential flow corresponds to a stationary Boltzmann distribution, the Boltzmann energy $\Phi_B$ from (20) can be used to distinguish between in-distribution and OOD samples based on their assigned energy values. Specifically, the potential energy model trained on the CIFAR-10 training set assigns energy values to both in-distribution samples (CIFAR-10 test set) and OOD samples from various other image datasets. We evaluate OOD detection performance using the AUROC metric, where a higher score reflects better model's efficacy in accurately assigning lower energy values to OOD samples. Table 2 compares the AUROC scores of VAPO with those of various likelihood-based and EBMs. The results show that our model performs exceptionally well on interpolated CIFAR-10 and CelebA $32 \times 32$, while performing moderately on CIFAR-100 and SVHN.

## 4.6 Long-Run Steady-State Equilibrium

Figure 12 illustrates long-run ODE sampling over an extended time horizon $t \in [0, 20]$ using the autonomous energy model parameterized by WideResNet. Additionally, Figure 13 illustrates long-run ODE sampling using the time-varying energy model parameterized by U-Net. The results indicate a similar deterioration in image quality over extended time periods, albeit to a greater extent compared to the autonomous model.

Figure 14 plots the mean gradient norm $\mathbb{E}\left[\|\nabla_x \Phi\|^2\right]$ and the mean energy norm $\mathbb{E}\left[\|\Phi\|^2\right]$, neither of which exhibit convergence. These results are consistent with those observed in EBMs trained using non-convergent short-run MCMC (Agoritsas et al., 2023; Nijkamp et al., 2020). This issue arises from the inherent difficulty neural network models face in learning complex energy landscapes in high-dimensional spaces. Regions that remain unseen during training can correspond to poorly modeled areas of the energy landscape, often resulting in the emergence of sharp local minima. Consequently, ODE-based sampling may become trapped in these local minima, leading to mode collapse and poor mixing, which manifest as visual artifacts such as excessive saturation and loss of background details.

To resolve these issues, we replace the deterministic ODE solver with the conventional SGLD sampler for image generation, enabling sampling from the Boltzmann energy via $x_{t+1} = x_t + \Delta_t \nabla_x \Phi_B(x_t) + \sqrt{2\Delta_t}\,\epsilon$ where $\epsilon \sim \mathcal{N}(0, \lambda^2 I)$ denotes isotropic Gaussian noise with temperature scale $\lambda$ (standard deviation), and $\Delta_t$ is the step size. The injected stochasticity from the diffusive noise in SGLD facilitates escape from local minima and enhances mixing efficiency during sampling. As shown in Figures 15 and 17, SGLD mitigates mode collapse and the long-run image samples converge well to the stationary equilibrium. Furthermore, Figures 16 and 18 demonstrate that the gradient norm converges to zero, while the energy norm asymptotically stabilizes, indicating steady-state thermalization. These SDE-based sampling results confirm that equilibrium convergence is achievable with a stochastic sampler. Nonetheless, the temperature scale $\lambda$ must be carefully tuned to balance convergence speed and sample quality. Moreover, our experiments show that ODE-based sampling consistently yields better FID scores, potentially due to the deterministic nature of the proposed potential flow and the straightness of the linearly interpolated OT-FM trajectories, which contribute to sharper and more consistent sample generation.

## 5    Conclusion

We propose VAPO, a novel energy-based potential flow framework designed to reduce the computational cost and instability typically associated with EBM training. Empirical results demonstrate that VAPO outperforms several existing EBMs in unconditional image generation and achieves competitive performance in OOD detection, highlighting its versatility across diverse generative modeling tasks. Despite these promising results, future work will aim to refine the training strategy to improve scalability to higher-resolution images and other data modalities, while addressing the limitations outlined in this work. Additionally, exploring generative models that inherently incorporate Neumann boundary conditions into the design of their blurring perturbation kernels (Rissanen et al., 2023; Hoogeboom & Salimans, 2023; Daras et al., 2023) presents a promising direction for improving energy landscape modeling and enhancing sample diversity without incurring the computational burden of long-run MCMC sampling.

**Broader Impact Statement**

Generative models represent a rapidly growing field of study with overarching implications in science and society. Our work proposes a new generative model designed for efficient data generation and OOD detection, with potential applications in fields such as medical imaging, entertainment, and content creation. However, as with any powerful technology, generative models come with substantial risks, including the potential misuse in creating deepfakes or misleading content that could undermine social security and trust. Given this dual-use nature, it is essential to implement safeguards, such as classifier-based guidance, to prevent the generation of biased or harmful content. Moreover, generative models are vulnerable to backdoor adversarial attacks and can inadvertently amplify biases present in the training data, reinforcing social inequalities. Although our work uses standard datasets, it is important to address how such biases are handled. We are actively exploring methods to identify and mitigate biases during both the training and generation phases. This includes employing fairness-aware training algorithms and evaluating the model's output for biased patterns. Furthermore, while this work demonstrates the potential benefits of generative models, the ethical concerns surrounding their deployment must be considered. Addressing these issues will require ongoing collaboration to develop frameworks for responsible use, including transparency, model interpretability, and robust safeguards against malicious applications. By proactively engaging with these ethical concerns, the broader community can contribute to the responsible advancement of generative modeling technologies.

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
