# A Additional Results

In this section, we present additional experiments that further validate the effectiveness and efficiency of the proposed VAPO framework. We first conduct an ablation study to evaluate the contribution of key loss components and architectural choices, demonstrating their impact on both FID performance and convergence to the Boltzmann equilibrium. Then, we compare the computational efficiency of VAPO against recent EBM baselines, highlighting its advantages in training and inference time while maintaining competitive generative performance across model variants.

## A.1 Ablation Study

To isolate and quantify the impact of individual components in the proposed VAPO loss function. Table 3 presents an ablation study conducted on a smaller (VAPO-Base) model and a reduced training batch size to accelerate training. Notably, the FID scores increase without (B) the covariance loss and (C) the cosine distance gradient alignment, indicating that these loss components are essential to the VAPO training. We note that since (B) learns only the normalized gradient and not the energy magnitude, it requires careful tuning of denormalization during ODE sampling.

Subsequently, (D) replaces the cosine distance with inner product, and (E) replaces the entire VAPO loss with the flow matching loss of Lipman et al. (2023). Although these loss configurations yield better FID performances, the sampling results and norm plots in Figures 19 - 22 show that neither of these configurations achieves Boltzmann equilibrium under SGLD sampling. Removing the cosine distance in (D) eliminates the scale invariance of cosine similarity, leading to large variations in gradient magnitudes that disrupt the stability of energy required for steady-state convergence. Similarly, the flow matching loss in (E) does not inherently enforce Boltzmann stationarity. Applying the same weighting $w(t)$ to the flow matching loss would halt the learning of the gradient field cutoff time $t_{\max}$.

For these reasons, we do not adopt the loss configurations (D) and (E) in our loss framework, despite the lower FID scores. In contrast, the variational nature of the covariance loss allows it to enforce that the energy values $\Phi(x, t)$ remain stationary near $t = 1$ without impeding learning. This covariance loss is fundamental as it corresponds to the Fokker-Planck dynamics underlying a probability path. More importantly, its adaptability to weighting ensures a proper establishment of the stationary Boltzmann distribution. Finally, by (F) incorporating a larger VAPO-Base model and (G) increasing the training batch size, we obtain improved FID scores with (A) the proposed VAPO loss function.

## A.2 Computational Efficiency

Table 4 compares the training-time computational efficiency of VAPO against the recent EBM baselines. Here, we included an additional smaller model (VAPO-Base) with fewer parameters but a higher FID score. The training time and GPU memory footprint are measured on a single A100 GPU of 80G memory. Italicized values represent estimates for the VAEBM model based on experiments conducted using smaller batch size, as the model cannot be trained on a single GPU using the prescribed batch sizes. Additionally, Table 5 compares the inference-time computational efficiency of VAPO against the recent EBM baselines. Overall, these results suggest that our method provides improved computational efficiency in both training and

Table 3: Ablation study across different training losses and model configurations.

| Loss Configuration | Model Variant | Parameter Count | Batch Size | FID ↓ |
|---|---|---|---|---|
| (A) VAPO loss | Base | 38.3M | 64 | 9.45 |
| (B) VAPO loss without covariance | Base | 38.3M | 64 | 13.1 |
| (C) VAPO loss without cosine distance | Base | 38.3M | 64 | 11.3 |
| (D) cosine distance → inner product | Base | 38.3M | 64 | 9.21 |
| (E) VAPO loss → flow matching loss | Base | 38.3M | 64 | 8.90 |
| (F) VAPO loss | Large | 55.7M | 64 | 7.66 |
| (G) VAPO loss | Large | 55.7M | 128 | 6.72 |

Table 4: Comparison of training-time computational efficiency against EBM baselines.

| Method | Sampling/Perturbation Approach | Parameter Count | Memory Usage (GB) | Training Iterations | Training Time (hrs) | FID ↓ |
|---|---|---|---|---|---|---|
| VAEBM Xiao et al. (2021a) | SGLD + Variational Inference + Replay Buffer | 135.9M | *129* | 25K | *414* | 12.2 |
| DRL (Gao et al. (2021)) | SGLD + Diffusion | 38.6M | 56 | 240K | 172 | 9.58 |
| CLEL (Lee et al. (2022)) | SGLD + Replay Buffer | 30.7M | 10 | 100K | 133 | 8.61 |
| CDRL (Zhu et al. (2024)) | SGLD + Amortized Inference + Diffusion | 34.8M | 69 | 400K | 144 | 4.31 |
| VAPO-Base (Ours) | Stationary-Enforced OT-FM | 38.3M | 35 | 300K | 48 | 9.33 |
| VAPO-Large (Ours) | Stationary-Enforced OT-FM | 55.7M | 60 | 300K | 112 | 6.72 |

Table 5: Comparison of inference-time computational efficiency against EBM baselines.

| Method | Numerical Approach | Parameter Count | Sampling Steps | Inference Time (s) | Training Latency (ms) | FID ↓ |
|---|---|---|---|---|---|---|
| VAEBM (Xiao et al. (2021a)) | SGLD | 135.1M | 16 | 21.3 | 13.31 | 12.2 |
| CoopFlow (Xie et al. (2022)) | SGLD | 45.9M | 30 | 2.5 | 0.833 | 15.8 |
| DRL (Gao et al. (2021)) | SGLD | 34.8M | 180 | 23.9 | 1.328 | 9.58 |
| CDRL (Zhu et al. (2024)) | SGLD | 38.6M | 90 | 12.2 | 1.356 | 4.31 |
| VAPO (Ours) | ODE Solver | 55.7M | 74 | 14.6 | 1.968 | 6.72 |

inference compared to most strong EBM baselines while maintaining competitive FID scores, particularly for the base model, which has a parameter count similar to the baselines. Nonetheless, there remains a gap in FID performance relative to the state-of-the-art generative models.

# B  Additional Discussions

In this section, we discuss the strengths and limitations of our proposed VAPO framework in the broader context of energy-based generative modeling. We first highlight the advantages of VAPO over conventional diffusion and flow-based approaches, emphasizing its interpretability, theoretical alignment with Boltzmann energy, and improved sampling efficiency. Next, we examine the critical role of MCMC in achieving Boltzmann-convergent training and sampling. While our method avoids the high cost of long-run MCMC by leveraging structured probability paths, we show that such paths may fail to explore low-density regions in high-dimensional spaces. This limitation can result in mode collapse and degraded sample quality when using deterministic ODE-based sampling. Finally, we analyze the remaining performance gap between VAPO and state-of-the-art EBMs, attributing it to architectural simplicity, trade-offs between convergence and generative sharpness, and the added complexity of modeling marginal rather than conditional distributions.

## B.1  Advantages over Diffusion and Flow-based Generative Models

Our proposed energy-parameterized potential flow offers several advantages over conventional diffusion and flow matching models, where vector fields are directly parameterized by neural networks rather than derived as the gradients of scalar-valued energy functions. Specifically, these benefits include:

**Interpretable Energy Landscape via Marginal Density Modeling** By explicitly parameterizing the vector field as the gradient of a scalar potential energy, our method provides a natural energy-based representation of data dynamics. Such a formulation supports key energy-based modeling tasks, including explicit (marginal) density estimation, composable generation, and OOD detection capabilities not inherently provided by conventional diffusion or flow matching approaches. As shown in Table 2, VAPO demonstrates strong OOD detection performance due to our proposed energy-based formulation.

**Theoretical Connection to Boltzmann Energy** Our Boltzmann energy formulation in Proposition 5 rigorously connects the deterministic potential flow to a stationary Boltzmann distribution characterized by the Boltzmann energy $\Phi_B$. This theoretical foundation firmly situates our approach within the energy-based modeling framework, offering theoretical coherence that is lacking in existing diffusion and flow matching models. As a result, it allows our approach to combine the efficiency of flow-based probability paths with the interpretability and rigor of the Boltzmann energy representation. As shown in Figure 2, the potential flow converges effectively to the stationary Boltzmann distribution.

**Optimality of Conservative Vector Field** Our approach learns a purely gradient-based vector field $\nabla_x \Phi$, in contrast to diffusion and flow matching methods, which may incorporate divergence-free components, as noted in Neklyudov et al. (2023). By enforcing a conservative energy function through Helmholtz decomposition, our method reduces the dynamical cost associated with these divergence-free components, enabling more efficient particle transport and improving training efficiency, as demonstrated by the comparative benchmark in Table 4.

**Efficient Deterministic ODE Sampling** By eliminating the reliance on implicit MCMC sampling, our approach reduces computational overhead and avoids common convergence issues encountered in traditional EBMs. Furthermore, our potential flow formulation enables deterministic ODE-based sampling that is generally more stable and efficient for generating high-quality samples with fewer steps than stochastic sampling methods, as demonstrated by the comparative results in Table 5.

## B.2    Incorporating Langevin Dynamics for Boltzmann-Convergent Sampling

Conventional EBM training often relies on convergent (long-run) MCMC sampling to thoroughly explore the data space and assign appropriate energy values across the landscape, particularly in low-density regions. This process helps smooth out sharp local minima and mitigates overfitting to high-density areas. In contrast, our framework employs a conditional homotopy $\rho(x \mid \bar{x}, t)$ (perturbation kernel) to guide training samples along a structured dynamic probability path. As noted by Nijkamp et al. (2019), such probability paths resemble short-run MCMC behavior, which limits their capacity to explore low-density regions in high-dimensional spaces. Consequently, many of these regions remain unseen during training and are thus poorly modeled in the resulting energy landscape. ODE-based samplers further exacerbate this issue due to their lack of stochasticity, making them more susceptible to becoming trapped in sharp local minima and suffering from poor mixing. Compared to stochastic samplers like Stochastic Gradient Langevin Dynamics (SGLD), deterministic sampling is inherently more sensitive to gaps in energy coverage. This limitation is not unique to our approach - it is a general challenge for diffusion-based and flow-based models that rely on time-dependent Gaussian perturbations to construct their training trajectories.

To achieve proper Boltzmann convergence and reduce mode collapse under deterministic ODE sampling, it is necessary to improve data space coverage beyond what is provided by structured Gaussian perturbations. This could be addressed by incorporating long-run MCMC during training, although doing so incurs the substantial computational overhead characteristic of traditional EBMs. As a potential direction for future work, we propose integrating long-run MCMC sampling within the reverse-diffusion conditional path, following techniques developed by Gao et al. (2021),Zhu et al. (2024), and Geng et al. (2024). However, these methods primarily model the conditional distribution $p(x_{t-1} \mid x_t)$ rather than the marginal data distribution $p(x)$, which remains the focus of our current work. As a result, these conditional EBMs do not establish a direct connection to the Boltzmann distribution $p(x) \propto e^{\Phi(x)}$, which forms the theoretical foundation of marginal energy-based modeling. Adapting such conditional modeling techniques to the marginal EBM setting would require substantial methodological developments. To offset the training cost of long-run MCMC, future work may also explore hybrid strategies involving pre-sampled replay buffers or distillation from a pre-trained convergent EBM.

## B.3    Addressing Performance Gaps: Modeling Trade-offs and Theoretical Challenges

While our proposed VAPO framework achieves competitive FID performance, there remains a noticeable gap compared to state-of-the-art EBMs. In the following discussion, we analyze the underlying factors contributing to this discrepancy.

**Difference in Model Architecture** DDAEBM (Geng et al., 2024) utilizes a multi-model architecture comprising three distinct components: (1) a generator model parameterized by the modified U-Net architecture of Xiao et al. (2021b), (2) an energy model parameterized by the NCSN++ architecture from Song et al. (2021), and (3) a CNN-based encoder. This multi-component architecture enables specialized modules to collaboratively refine each other's behavior through adversarial training, thereby enhancing generative performance. In contrast, our VAPO model adopts a simpler framework design, employing only a single EBM parameterized by the U-Net architecture described in Dhariwal & Nichol (2021). Although this single-

component architecture has a comparable parameter count to that of the NCSN++ used in DDAEBM, it may lack the collaborative optimization and mutual refinement advantages that arise from multi-model training setups. Consequently, our VAPO's FID scores align more closely with those of single-energy or joint-energy EBMs (Salimans & Ho, 2021; Gao et al., 2021; Lee et al., 2022). We hypothesize that incorporating additional auxiliary model components with adversarial training strategies, as utilized by DDAEBM, could enhance the representational capacity and further sharpen the energy landscape of our VAPO model, potentially leading to improved FID performance. Investigating this hybrid approach, which balances computational efficiency with adversarial training strategy, will be reserved for our future work.

**Boltzmann stationarity and FID Trade-off** In our previous ablation study, we observed a notable trade-off between Boltzmann stationarity and FID performance. This observation aligns with insights from Agoritsas et al. (2023) and Nijkamp et al. (2020), which show that non-convergent EBMs outperform convergent EBMs trained with long-run MCMC sampling in image generation. Models such as DDAEBM fall within this class of non-convergent EBMs. We hypothesize that this is due to the inherent tension between accurate equilibrium modeling and sharp generative quality. Specifically, imposing strict Boltzmann convergence tends to smooth out the energy landscape, inadvertently flattening local minima that correspond to meaningful data modes. Although such smoothing enhances theoretical interpretability by faithfully approximating the true equilibrium of the Boltzmann distribution, it compromises the sharpness and detail of generated samples, leading to higher FID scores. To mitigate this limitation, we propose exploring advanced sampling techniques such as the Metropolis-Adjusted Langevin Algorithm (MALA) used in Pal et al. (2021a) or other adaptive short-run samplers in future work. The gradient-informed proposal and acceptance steps of MALA could potentially enhance local mode exploration efficiency without fully abandoning the Boltzmann equilibrium.

**Conditional vs Marginal EBMs** Another critical contributing factor is the distinction between conditional EBMs as modeled by DDAEBM, and marginal EBMs considered by VAPO. In particular, Geng et al. (2024) articulate that modeling conditional distributions simplifies the learning task, as these conditional distributions are inherently less multi-modal compared to complex marginal distributions. DDAEBM leverages this insight by decomposing the generation process into discrete diffusion steps, each focusing on simpler, conditional distributions that are easier to model effectively. In contrast, our VAPO explicitly models a global marginal distribution, thus inherently facing greater complexity due to increased modality. Consequently, achieving comparable generative performance poses additional challenges. To address this limitation, future work could explore hierarchical or multi-stage conditional modeling strategies to simplify explicit marginal modeling. Nevertheless, conditional EBMs inherently lack a direct relationship to the marginal Boltzmann distribution, which is fundamental to the theoretical underpinnings of our VAPO framework. Therefore, adapting conditional modeling techniques to fully marginal EBMs would require substantial development.

## C  Proofs and Derivations

### C.1  Proof of Proposition 1

*Proof.* Based on the definitions of $q(x)$ and $p(\bar{x} \mid x)$, we can expand their logarithms (ignoring additive constants) as follows:

$$\log q(x) = -\frac{1}{2\,\omega^2}\frac{1}{2\,\omega^2}\,\|x\|^2 + (\text{terms independent of } x) \tag{26}$$

$$\log p(\bar{x} \mid x) = -\frac{1}{2\,\nu^2}\,\|\bar{x} - x\|^2 + (\text{terms independent of } x) \tag{27}$$

Substituting these into (5), we obtain:

$$h(x \mid \bar{x}, t) = -\frac{\alpha(t)}{2\,\omega^2}\,\|x\|^2 - \frac{\beta(t)}{2\,\nu^2}\,\|\bar{x} - x\|^2 \tag{28}$$

Expanding the squared term:

$$\|\bar{x} - x\|^2 = \|x\|^2 - 2\,x^T\bar{x} + \|\bar{x}\|^2 \tag{29}$$

and substituting it back into $h(x \mid \bar{x}, t)$:

$$h(x \mid \bar{x}, t) = -\left(\frac{\alpha(t)}{\omega^2} + \frac{\beta(t)}{\nu^2}\right)\|x\|^2 + \frac{\beta(t)}{\nu^2}\,x^T\bar{x} \tag{30}$$

Recognizing the quadratic form in terms of $x$, we identify that $\rho(x \mid \bar{x}, t)$ is a Gaussian density:

$$\rho(x \mid \bar{x}, t) = \mathcal{N}\big(x; \mu(t)\bar{x}, \sigma(t)^2 I\big) \tag{31}$$

whose mean $\mu(t)$ and variance $\sigma(t)^2$ can be obtained by completing the square.

Define

$$A := \frac{\alpha(t)}{\omega^2} + \frac{\beta(t)}{\nu^2}, \qquad B := \frac{\beta(t)}{\nu^2} \tag{32}$$

Then the exponent becomes

$$h(x \mid \bar{x}, t) = -\frac{1}{2}\Big[A\,\|x\|^2 - 2B\,x^T\bar{x}\Big] \tag{33}$$

and we wish to express this quadratic form as follows

$$A\,\|x - \mu(t)\,\bar{x}\|^2 + (\text{terms independent of } x) \tag{34}$$

Expanding $A\,\|x - \mu(t)\,\bar{x}\|^2$, we obtain

$$A\,\|x - \mu(t)\,\bar{x}\|^2 = A\,\|x\|^2 - 2A\,\mu(t)\,x^T\bar{x} + A\,\mu(t)^2\,\|\bar{x}\|^2 \tag{35}$$

To match the linear term, the mean of the Gaussian is thus given by

$$\mu(t) = \frac{B}{A} = \frac{\beta(t)/\nu^2}{\alpha(t)/\omega^2 + \beta(t)/\nu^2} = \text{sigmoid}\left(\log\left(\frac{\beta(t)}{\alpha(t)}\frac{\omega^2}{\nu^2}\right)\right) \tag{36}$$

where $\text{sigmoid}(z) = \frac{1}{1+e^{-z}}$ denotes the standard logistic (sigmoid) function.

By comparing with the standard Gaussian exponent

$$-\frac{1}{2\sigma^2}\,\|x - \mu(t)\,\bar{x}\|^2, \tag{37}$$

we deduce that the variance is given by

$$\sigma(t)^2 = \frac{1}{A} = \frac{1}{\alpha(t)/\omega^2 + \beta(t)/\nu^2}. \tag{38}$$

Using the expression obtained for $\mu(t)$, the standard deviation can also be written as

$$\sigma(t) = \sqrt{\frac{\nu^2}{\beta(t)} \mu(t)}. \tag{39}$$

$\square$

## C.2 Proof of Proposition 2

*Proof.* Differentiating the conditional homotopy $\rho(x \mid \bar{x}, t)$ in (4) with respect to $t$, we have:

$$
\begin{aligned}
\frac{\partial \rho(x \mid \bar{x}, t)}{\partial t} &= \frac{1}{\int_\Omega e^{h(x|\bar{x},t)} dx} \frac{\partial [e^{h(x|\bar{x},t)}]}{\partial t} - \frac{e^{h(x|\bar{x},t)}}{[\int_\Omega e^{h(x|\bar{x},t)} dx]^2} \frac{\partial [\int_\Omega e^{h(x|\bar{x},t)} dx]}{\partial t} \\
&= \frac{1}{\int_\Omega e^{h(x|\bar{x},t)} dx} \frac{\partial [e^{h(x|\bar{x},t)}]}{\partial f} \frac{\partial h(x \mid \bar{x}, t)}{\partial t} - \frac{e^{h(x|\bar{x},t)}}{[\int_\Omega e^{h(x|\bar{x},t)} dx]^2} \int_\Omega \frac{\partial [e^{h(x|\bar{x},t)}]}{\partial f} \frac{\partial h(x \mid \bar{x}, t)}{\partial t} dx \\
&= \frac{e^{h(x|\bar{x},t)}}{\int_\Omega e^{h(x|\bar{x},t)} dx} \frac{\partial h(x \mid \bar{x}, t)}{\partial t} - \frac{e^{h(x|\bar{x},t)}}{\int_\Omega e^{h(x|\bar{x},t)} dx} \int_\Omega \frac{e^{h(x|\bar{x},t)}}{\int_\Omega e^{h(x|\bar{x},t)} dx} \frac{\partial h(x \mid \bar{x}, t)}{\partial t} dx \\
&= \rho(x \mid \bar{x}, t) \left( \frac{\partial h(x \mid \bar{x}, t)}{\partial t} - \int_\Omega \rho(x \mid \bar{x}, t) \frac{\partial h(x \mid \bar{x}, t)}{\partial t} dx \right) \\
&= -\frac{1}{2} \rho(x \mid \bar{x}, t) \left( \frac{d\alpha(t)}{dt} \frac{x^T x}{\omega^2} + \frac{d\beta(t)}{dt} \frac{(x - \bar{x})^T (x - \bar{x})}{\nu^2} \right. \\
&\qquad\qquad\qquad \left. - \int_\Omega \rho(x \mid \bar{x}, t) \frac{d\alpha(t)}{dt} \frac{x^T x}{\omega^2} + \frac{d\beta(t)}{dt} \frac{(x - \bar{x})^T (x - \bar{x})}{\nu^2} dx \right)
\end{aligned} \tag{40}
$$

where we have applied the quotient rule in the first equation and the chain rule in the second equation.
Subsequently, define

$$\gamma(x, \bar{x}, t) = \frac{d\alpha(t)}{dt} \frac{\|x\|^2}{\omega^2} + \frac{d\beta(t)}{dt} \frac{\|x - \bar{x}\|^2}{\nu^2} \tag{41}$$

and using the fact that:

$$\frac{\partial \bar{\rho}(x, t)}{\partial t} = \frac{\partial \int_\Omega \rho(x \mid \bar{x}, t) \, p_{\text{data}}(\bar{x}) \, d\bar{x}}{\partial t} = \int_\Omega \frac{\partial \rho(x \mid \bar{x}, t)}{\partial t} \, p_{\text{data}}(\bar{x}) \, d\bar{x} \tag{42}$$

we can substitute (40) into (42) to obtain:

$$\frac{\partial \bar{\rho}(x, t)}{\partial t} = -\int_\Omega p_{\text{data}}(\bar{x}) \, \rho(x \mid \bar{x}, t) \left( \gamma(x, \bar{x}, t) - \int_\Omega \rho(x \mid \bar{x}, t) \, \gamma(x, \bar{x}, t) \, dx \right) d\bar{x} \tag{43}$$

Given that both $\rho(x \mid \bar{x}, t)$ and $p_{\text{data}}(\bar{x})$ are normalized (proper) density functions, writing (43) in terms of expectations yields the PDE in (9).

$\square$

## C.3 Proof of Proposition 3

Here, we used the Einstein tensor notation interchangeably with the conventional notation for vector dot product and matrix-vector multiplication in PDE. Also, we omit the time index $t$ of $\Phi(x, t)$ in this section for brevity.

*Proof.* Applying forward Euler to the particle flow ODE (11) using step size $\Delta_t$, we obtain:

$$x_{t+\Delta_t} = \alpha(x_t) = x_t + \Delta_t\, u(x_t) \tag{44}$$

where

$$u(x_t) = \nabla_x \Phi(x_t) \tag{45}$$

where $x_t$ denotes the discretization of $x(t)$. Hereafter, we abbreviate $x_t$, $\alpha(x_t)$, $\nu(x_t)$ as $x$, $\alpha$, $\nu$, respectively.

Assuming that the $\alpha : \Omega \to \Omega$ is a diffeomorphism (bijective function with differentiable inverse), the push-forward operator $\alpha_\# : \mathbb{R} \to \mathbb{R}$ defines the density transformation $\rho^\Phi(\alpha, t + \Delta_t) := \alpha_\# \rho^\Phi(x, t)$. Associated with this change of variable formula is the following density transformation:

$$\rho^\Phi\big(\alpha, t + \Delta_t\big) = \frac{1}{|D_x\alpha|}\, \rho^\Phi(x, t) \tag{46}$$

where $|D_x\alpha|$ denotes the Jacobian determinant of $\alpha$, where the the Jacobian is taken with respect to $x$.

From (9) and (43), we obtain:

$$\frac{\partial \log \bar\rho(x, t)}{\partial t} = \frac{1}{\bar\rho(x, t)} \frac{\partial \bar\rho(x, t)}{\partial t} = -\frac{1}{\bar\rho(x, t)} \frac{1}{2}\, \mathbb{E}_{p_{\text{data}}(\bar x)} \Big[\rho(x \mid \bar x, t)\big(\gamma(x, \bar x, t) - \bar\gamma(x, \bar x, t)\big)\Big] \tag{47}$$

Applying the forward Euler method to (47), we obtain:

$$\log \bar\rho(x, t + \Delta_t) \geq \log \bar\rho(x, t) - \frac{\Delta_t}{2} \frac{1}{\bar\rho(x, t)}\, \mathbb{E}_{p_{\text{data}}(\bar x)} \Big[\rho(x \mid \bar x, t)\big(\gamma(x, \bar x, t) - \bar\gamma(x, \bar x, t)\big)\Big] \tag{48}$$

Applying the change-of-variables formula (46) and then substituting (48) into the KL divergence $\mathcal{D}_{\text{KL}}\big[\rho(x, t+\Delta_t)\|\bar\rho(x, t+\Delta_t)\big]$ at time $t + \Delta_t$, we obtain:

$$\mathcal{D}_{\text{KL}}\big[\rho^\Phi(x, t + \Delta_t)\|\bar\rho(x, t + \Delta_t)\big] = \int_\Omega \rho^\Phi(x, t)\, \log\left(\frac{\rho^\Phi\big(\alpha, t + \Delta_t\big)}{\bar\rho\big(\alpha, t + \Delta_t\big)}\right) dx$$

$$= \int_\Omega \rho^\Phi(x, t)\left(\log \rho^\Phi(x, t) - \log|D_x\alpha| - \log \bar\rho(\alpha, t)\right. \tag{49}$$

$$\left. + \frac{\Delta_t}{2} \frac{1}{\bar\rho(\alpha, t)}\, \mathbb{E}_{p_{\text{data}}(\bar x)} \Big[\rho(\alpha \mid \bar x, t)\big(\gamma(\alpha, \bar x, t) - \bar\gamma(\alpha, \bar x, t)\big)\Big] + C\right) dx$$

Consider minimizing the KL divergence (49) with respect to $\alpha$ as follows:

$$\min_\alpha\, \mathcal{D}_{\text{KL}}(\alpha) = \min_\alpha\, \underbrace{\frac{\Delta_t}{2} \int_\Omega \rho^\Phi(x, t) \frac{1}{\bar\rho(\alpha, t)}\, \mathbb{E}_{p_{\text{data}}(\bar x)} \Big[\rho(\alpha \mid \bar x, t)\big(\gamma(\alpha, \bar x, t) - \bar\gamma(\alpha, \bar x, t)\big)\Big] dx}_{\mathcal{D}_1^{\text{KL}}(\alpha)}$$

$$\underbrace{- \int_\Omega \rho^\Phi(x, t)\, \log \bar\rho(\alpha, t)\, dx}_{\mathcal{D}_2^{\text{KL}}(\alpha)} \quad \underbrace{- \int_\Omega \rho^\Phi(x, t)\, \log|D_x\alpha|\, dx}_{\mathcal{D}_3^{\text{KL}}(\alpha)} \tag{50}$$

where we have neglected the constant terms that do not depend on $\alpha$.

To solve the optimization (50), we consider the following optimality condition in the first variation of $\mathcal{D}_{\text{KL}}$:

$$\mathcal{I}(\alpha, \nu) = \frac{d}{d\epsilon}\, \mathcal{D}_{\text{KL}}\big(\alpha + \epsilon\, \nu\big)\bigg|_{\epsilon=0} = 0 \tag{51}$$

This condition must hold for all trial functions $\nu$.

Subsequently, taking the variational derivative of the first functional $\mathcal{D}_1^{\mathrm{KL}}$ in (50), we obtain:

$$
\begin{aligned}
\mathcal{I}^1(\alpha, \nu) &= \frac{d}{d\epsilon} \mathcal{D}_1^{\mathrm{KL}}(\alpha + \epsilon\nu) \Big|_{\epsilon=0} \\
&= \frac{\Delta_t}{2} \int_\Omega \rho^\Phi(x, t) \frac{d}{d\epsilon} \left\{ \frac{1}{\bar{\rho}(\alpha + \epsilon\nu, t)} \, \mathbb{E}_{p_{\mathrm{data}}(\bar{x})} \left[ \rho(\alpha + \epsilon\nu \mid \bar{x}, t) \left( \gamma(\alpha + \epsilon\nu, \bar{x}, t) - \bar{\gamma}(\alpha + \epsilon\nu, \bar{x}, t) \right) \right] \right\} \Big|_{\epsilon=0} dx \\
&= \frac{\Delta_t}{2} \int_\Omega \rho^\Phi(x, t) \frac{\partial}{\partial \alpha} \left\{ \frac{1}{\bar{\rho}(\alpha, t)} \, \mathbb{E}_{p_{\mathrm{data}}(\bar{x})} \left[ \rho(\alpha \mid \bar{x}, t) \left( \gamma(\alpha, \bar{x}, t) - \bar{\gamma}(\alpha, \bar{x}, t) \right) \right] \right\} \nu \, dx \\
&= \frac{\Delta_t}{2} \int_\Omega \rho^\Phi(x, t) \, D_x \left\{ \frac{1}{\bar{\rho}(\alpha, t)} \, \mathbb{E}_{p_{\mathrm{data}}(\bar{x})} \left[ \rho(\alpha \mid \bar{x}, t) \left( \gamma(\alpha, \bar{x}, t) - \bar{\gamma}(\alpha, \bar{x}, t) \right) \right] \right\} (D_x \alpha)^{-1} \nu \, dx
\end{aligned}
\tag{52}
$$

where the last equation is due to chain rule $\frac{\partial f}{\partial \alpha} = D_x f \, (D_x \alpha)^{-1}$.

Applying the Taylor series expansion to the derivative $\frac{\partial g}{\partial x_i}(\alpha)$ with respect to $x_i$ yields:

$$
\frac{\partial g(\alpha)}{\partial x_i} = \frac{\partial g(x + \Delta_t u)}{\partial x_i} = \frac{\partial g(x)}{\partial x_i} + \Delta_t \sum_j \frac{\partial^2 g(x)}{\partial x_i \, \partial x_j} u_j + O(\Delta_t^2)
\tag{53}
$$

In addition, the inverse of Jacobian $D_x \alpha^{-1}$ can be expanded via the Neuman series to obtain:

$$
D_x \alpha^{-1} = \left( \mathrm{I} + \Delta_t D_x u \right)^{-1} = \mathrm{I} - \Delta_t D_x u + O(\Delta_t^2)
\tag{54}
$$

Using the Taylor series and Neuman series expansions in (53) and (54), we can write (52) in tensor notation, as follows:

$$
\mathcal{I}^1(\alpha, \nu) = \frac{\Delta_t}{2} \int_\Omega \rho^\Phi(x, t) \sum_i \frac{\partial}{\partial x_i} \left\{ \frac{1}{\bar{\rho}(x, t)} \, \mathbb{E}_{p_{\mathrm{data}}(\bar{x})} \left[ \rho(x \mid \bar{x}, t) \left( \gamma(x, \bar{x}, t) - \bar{\gamma}(x, \bar{x}, t) \right) \right] \right\} \nu_i \, dx + O(\Delta_t^2)
\tag{55}
$$

Taking the variational derivative of the second functional $\mathcal{D}_2^{\mathrm{KL}}$ in (50) yields:

$$
\begin{aligned}
\mathcal{I}^2(\alpha, \nu) &= \frac{d}{d\epsilon} \mathcal{D}_2^{\mathrm{KL}}(\alpha + \epsilon\nu) \Big|_{\epsilon=0} = \int_\Omega \rho^\Phi(x, t) \frac{d}{d\epsilon} \log \bar{\rho}(\alpha + \epsilon\nu) \Big|_{\epsilon=0} dx \\
&= \int_\Omega \rho^\Phi(x, t) \frac{1}{\bar{\rho}(\alpha, t)} \nabla_x \bar{\rho}(\alpha, t) \cdot \nu \, dx = \int_\Omega \rho^\Phi(x, t) \nabla_x \log \bar{\rho}(\alpha, t) \cdot \nu \, dx
\end{aligned}
\tag{56}
$$

where we have used the derivative identity $d \log g = \frac{1}{g} \, dg$ to obtain the second equation.

Using the Taylor series expansion (53), we can write (56) in tensor notation, as follows:

$$
\begin{aligned}
\mathcal{I}^2(\alpha, \nu) &= -\int_\Omega \rho^\Phi(x, t) \sum_i \left( \frac{\partial \log \bar{\rho}(x, t)}{\partial x_i} - \Delta_t \sum_j \frac{\partial^2 \log \bar{\rho}(x, t)}{\partial x_i \, \partial x_j} u_j \right) \nu_i \, dx + O(\Delta_t^2) \\
&= -\int_\Omega \rho^\Phi(x, t) \sum_i \left( \frac{\partial \log \bar{\rho}(x, t)}{\partial x_i} - \Delta_t \sum_j \frac{\partial^2 \log \bar{\rho}(x, t)}{\partial x_i \, \partial x_j} u_j \right) \nu_i \, dx + O(\Delta_t^2)
\end{aligned}
\tag{57}
$$

Similarly, taking the variational derivative of the $\mathcal{D}_3^{\text{KL}}$ term in (50), we obtain:

$$\mathcal{I}^3(\alpha, \nu) = \frac{d}{d\epsilon} \mathcal{D}_3^{\text{KL}}(\alpha + \epsilon\nu) \Big|_{\epsilon=0} = \int_\Omega \rho^\Phi(x,t) \frac{d}{d\epsilon} \log |D(\alpha + \epsilon\nu)| \Big|_{\epsilon=0} dx$$

$$= \int_\Omega \rho^\Phi(x,t) \frac{1}{|D_x\alpha|} \frac{d}{d\epsilon} |D(\alpha + \epsilon\nu)| \Big|_{\epsilon=0} dx = \int_\Omega \rho^\Phi(x,t) \operatorname{tr}\left(D_x\alpha^{-1} D\nu\right) dx \tag{58}$$

where we have used the following Jacobi's formula:

$$\frac{d}{d\epsilon} |D(\alpha + \epsilon\nu)| \Big|_{\epsilon=0} = |D_x\alpha| \operatorname{tr}\left(D_x\alpha^{-1} D\nu\right) \tag{59}$$

to obtain the last equation in (58).

Substituting in (54) and using the Taylor series expansion (53), (56) can be written in tensor notation as follows:

$$\mathcal{I}^3(\alpha, \nu) = \int_\Omega \sum_i \left( \rho^\Phi(x,t) \frac{\partial\nu_i}{\partial x_i} - \Delta_t \sum_j \rho^\Phi(x,t) \frac{\partial u_j}{\partial x_i} \frac{\partial\nu_i}{\partial x_j} \right) dx + O(\Delta_t^2)$$

$$= \int_\Omega \sum_i \left( \frac{\partial\rho^\Phi(x,t)}{\partial x_i} \nu_i - \Delta_t \sum_j \frac{\partial}{\partial x_j} \left\{ \rho^\Phi(x,t) \frac{\partial u_j}{\partial x_i} \right\} \nu_i \right) dx + O(\Delta_t^2) \tag{60}$$

$$= \int_\Omega \sum_i \left( \frac{\partial\rho^\Phi(x,t)}{\partial x_i} - \Delta_t \sum_j \frac{\partial}{\partial x_j} \left\{ \rho^\Phi(x,t) \frac{\partial u_j}{\partial x_i} \right\} \right) \nu_i \, dx + O(\Delta_t^2)$$

where we have used integration by parts to obtain the second equation.

Taking the limit $\lim \Delta_t \to 0$, the terms $O(\Delta_t^2)$ that approach zero exponentially vanish. Subtracting (55) by (57) and (60), then equating to zero, we obtain the first-order optimality condition (51) as follows:

$$\int_\Omega \bar\rho(x,t) \sum_i \sum_j - \frac{\partial}{\partial x_i} \left\{ \frac{1}{\bar\rho(x,t)} \frac{\partial}{\partial x_j} \left\{ \bar\rho(x,t) \, u_j \right\} \right\}$$
$$+ \frac{1}{2} \frac{\partial}{\partial x_i} \left\{ \frac{1}{\bar\rho(x,t)} \mathbb{E}_{p_{\text{data}}(\bar x)} \left[ \rho(x \mid \bar x, t) \left( \gamma(x, \bar x, t) - \bar\gamma(x, \bar x, t) \right) \right] \right\} \nu_i \, dx = 0 \tag{61}$$

where we have assumed that $\rho^\Phi(x,t) \equiv \bar\rho(x,t)$ holds and have used the following identities:

$$\frac{\partial \log \bar\rho(x,t)}{\partial x_i} = \frac{1}{\bar\rho(x,t)} \frac{\partial\bar\rho(x,t)}{\partial x_i}$$
$$\frac{\partial^2 \log \bar\rho(x,t)}{\partial x_i \partial x_j} = \frac{\partial}{\partial x_i} \left( \frac{1}{\bar\rho(x,t)} \frac{\partial\bar\rho(x,t)}{\partial x_j} \right) \tag{62}$$

Given that $\nu_i$ can take any value, equation (61) holds (in the weak sense) only if the terms within the round bracket vanish. Integrating this term with respect to the $x_i$, we obtain:

$$\sum_j \frac{\partial}{\partial x_j} \left\{ \bar\rho(x,t) \, u_j \right\} = \frac{1}{2} \mathbb{E}_{p_{\text{data}}(\bar x)} \left[ \rho(x \mid \bar x, t) \left( \gamma(x, \bar x, t) - \bar\gamma(x, \bar x, t) \right) \right] + \bar\rho(x,t) \, C \tag{63}$$

which can also be written in vector notation as follows:

$$\nabla_x \cdot \left( \bar\rho(x,t) \, u \right) = \frac{1}{2} \mathbb{E}_{p_{\text{data}}(\bar x)} \left[ \rho(x \mid \bar x, t) \left( \gamma(x, \bar x, t) - \bar\gamma(x, \bar x, t) \right) \right] + \bar\rho(x,t) \, C \tag{64}$$

To find the scalar constant $C$, we integrate both sides of (64) to obtain:

$$\int_\Omega \nabla_x \cdot \big(\bar{\rho}(x,t)\,u\big)\,dx = \frac{1}{2} \int_\Omega \mathbb{E}_{p_{\text{data}}(\bar{x})} \Big[\rho(x \mid \bar{x}, t)\,\big(\gamma(x,\bar{x},t) - \bar{\gamma}(x,\bar{x},t)\big)\Big]\,dx \;+\; \int_\Omega \bar{\rho}(x,t)\,C\,dx$$
$$= \frac{1}{2} \int_\Omega \mathbb{E}_{p_{\text{data}}(\bar{x})} \Big[\rho(x \mid \bar{x}, t)\,\big(\gamma(x,\bar{x},t) - \bar{\gamma}(x,\bar{x},t)\big)\Big]\,dx \;+\; C \tag{65}$$

Applying the divergence theorem to the left-hand side of (65), we obtain:

$$\int_\Omega \nabla_x \cdot \big(\bar{\rho}(x,t)\,u\big)\,dx = \int_{\partial\Omega} \bar{\rho}(x,t)\,u \cdot \hat{n}\,dx \tag{66}$$

where $\hat{n}$ is the outward unit normal vector to the boundary $\partial\Omega$ of $\Omega$.

Given that $\bar{\rho}(x,t)$ is a normalized (proper) density with compact support (vanishes on the boundary), the term (66) becomes zero, leading to $C = 0$. Substituting this result along with $u(x) = \nabla_x \Phi(x)$ into (64), we arrive at the following PDE:

$$\nabla_x \cdot \big(\bar{\rho}(x,t)\,\nabla_x \Phi(x)\big) = \frac{1}{2}\,\mathbb{E}_{p_{\text{data}}(\bar{x})} \Big[\rho(x \mid \bar{x}, t)\,\big(\gamma(x,\bar{x},t) - \bar{\gamma}(x,\bar{x},t)\big)\Big] \tag{67}$$

Therefore, assuming that the base case $\rho_0(x) \equiv \bar{\rho}_0(x)$ holds and that a solution to (67) exists at every $t$, the proposition follows by the principle of induction.

$\square$

## C.4  Proof of Proposition 4

To show that the conditional and marginal homotopies satisfy the reverse diffusion process, we first express the forward-time SDE and ODE of Song et al. (2021):

$$dx(\tau) = f(\tau)\,x(\tau)\,d\tau + g(\tau)\,dW(\tau)$$
$$\frac{dx(\tau)}{d\tau} = f(\tau)\,x(\tau) - \frac{1}{2}\,g(\tau)^2\,\nabla_x \log p(x,\tau) \tag{68}$$

in terms of reverse time $t = 1 - \tau$, via applying the change of variable $dt = -d\tau$ as follows:

$$dx(t) = -f(t)\,x(t)\,dt + g(t)\,dW(t)$$
$$\frac{dx(t)}{dt} = -f(t)\,x(t) + \frac{1}{2}\,g(t)^2\,\nabla_x \log \bar{\rho}(x,t) \tag{69}$$

which gives (15) and (17).

Substituting the marginal score $\nabla_x \log \bar{\rho}(x,t)$ with the conditional score:

$$\nabla_x \log \rho(x \mid \bar{x}, t) = \frac{1}{\rho(x \mid \bar{x}, t)}\,\nabla_x \rho(x \mid \bar{x}, t) = -\frac{\epsilon}{\sigma(t)} \tag{70}$$

and applying reparameterization $x(t) = \mu(t)\,\bar{x} + \sigma(t)\,\epsilon$ and (16), we can write the conditional ODE as follows:

$$\begin{aligned}
\frac{dx(t)}{dt} &= v(x \mid \bar{x}, t) \\
&= -f(t)\,x(t) + \frac{1}{2}\,g(t)^2\,\nabla_x \log \rho(x \mid \bar{x}, t) \\
&= -f(t)\,x(t) + \frac{1}{2}\,g(t)^2\,\frac{\epsilon}{\sigma(t)} \\
&= -f(t)\,x(t) + \sigma(t)\,\big(\dot{\sigma}(t) + f(t)\,\sigma(t)\big)\,\frac{\epsilon}{\sigma(t)} \\
&= \frac{\dot{\mu}(t)}{\mu(t)}\,\big(x(t) - \sigma(t)\,\epsilon\big) + \dot{\sigma}(t)\,\epsilon \\
&= \dot{\mu}(t)\,\bar{x} + \dot{\sigma}(t)\,\epsilon
\end{aligned} \tag{71}$$

and thus corresponds to the conditional vector field defined in flow matching (Lipman et al., 2023). Marginalizing (71) with respect to

$$p_{\text{data}}(\bar{x} \mid x) = \frac{\rho(x \mid \bar{x}, t) \, p_{\text{data}}(\bar{x})}{\bar{\rho}(x, t)} \tag{72}$$

and substituting (19) and applying (70), we obtain

$$
\begin{aligned}
v(x,t) &= \int_{\Omega} \left( -f(t) \, x(t) + \frac{1}{2} \, g(t)^2 \, \nabla_x \log \rho(x \mid \bar{x}, t) \right) \frac{\rho(x \mid \bar{x}, t) \, p_{\text{data}}(\bar{x})}{\bar{\rho}(x, t)} \, d\bar{x} \\
&= -f(t) \, x(t) + \frac{1}{2} \, g(t)^2 \int_{\Omega} \nabla_x \log \rho(x \mid \bar{x}, t) \frac{\rho(x \mid \bar{x}, t) \, p_{\text{data}}(\bar{x})}{\bar{\rho}(x, t)} \, d\bar{x} \\
&= -f(t) \, x(t) + \frac{1}{2} \, g(t)^2 \int_{\Omega} \frac{1}{\rho(x \mid \bar{x}, t)} \frac{\rho(x \mid \bar{x}, t) \, p_{\text{data}}(\bar{x})}{\bar{\rho}(x, t)} \nabla_x \rho(x \mid \bar{x}, t) \, d\bar{x} \\
&= -f(t) \, x(t) + \frac{1}{2} \, g(t)^2 \frac{1}{\bar{\rho}(x, t)} \nabla_x \bar{\rho}(x, t) \\
&= -f(t) \, x(t) + \frac{1}{2} \, g(t)^2 \frac{1}{\bar{\rho}(x, t)} \nabla_x \log \bar{\rho}(x, t)
\end{aligned}
\tag{73}
$$

and thus corresponds to the marginal probability flow ODE 17.

### C.5 Proof of Proposition 5

*Proof.* Based on the result of Proposition 4 and using (12), we can express the homotopy matching problem

$$\frac{\partial \rho_{\Phi}(x, t)}{dt} = \frac{\partial \bar{\rho}(x, t)}{dt} \tag{74}$$

equivalently as

$$\nabla_x \cdot \left( \rho_{\Phi} \, \nabla_x \Phi(x, t) \right) = \nabla_x \cdot \left( \rho_{\Phi} \left( -f(t) \, x(t) + \frac{1}{2} \, g(t)^2 \, \nabla_x \log \bar{\rho}(x, t) \right) \right) \tag{75}$$

Given that this matching holds identically, we have

$$\nabla_x \Phi(x, t) = -f(t) \, x(t) + \frac{1}{2} \, g(t)^2 \, \nabla_x \log \bar{\rho}(x, t) \tag{76}$$

Furthermore, given that both the forward-time ODE and SDE of Song et al. (2021) exhibit the same marginal probability density $\bar{\rho}(x, t)$, it is shown that they satisfy the following reverse-time SDE:

$$dx(\tau) = \left( f(\tau) \, x(\tau) - g(\tau)^2 \, \nabla_x \log \bar{\rho}(x, t) \right) d\tau + g(\tau) \, dW(\tau) \tag{77}$$

which reverses the diffusion process as outlined by Anderson (1982) and Song et al. (2021). Applying the change of variable $dt = -d\tau$, this reverse-time SDE can similarly be written in terms of $t = 1 - \tau$ as

$$dx(t) = -\left( f(t) \, x(t) - g(t)^2 \, \nabla_x \log \bar{\rho}(x, t) \right) dt + g(t) \, dW(t) \tag{78}$$

where $dW(t)$ does not change sign, since the Wiener process is invariant under time reversal.

Subsequently, the Fokker-Plank dynamic that governs the time evolution of the marginal density homotopy $\bar{\rho}(x, t)$ is given by

$$\frac{\partial \bar{\rho}(x, t)}{\partial t} = -\nabla_x \cdot \left( \bar{\rho}(x, t) \left( -f(t) \, x(t) + g(t)^2 \, \nabla_x \log \bar{\rho}(x, t) \right) \right) + \frac{1}{2} \, g(t)^2 \, \Delta_x \, \bar{\rho}(x, t) \tag{79}$$

where $\Delta_x = \nabla_x \cdot \nabla_x$ denotes the Laplacian. By substituting (76) into this Fokker-Plank equation, we then have

$$\frac{\partial \bar{\rho}(x,t)}{\partial t} = - \nabla_x \cdot \left( \bar{\rho}(x,t) \left( 2 \nabla_x \Phi(x,t) + f(t) \, x(t) \right) \right) + \frac{1}{2} \, g(t)^2 \, \Delta_x \, \bar{\rho}(x,t) \tag{80}$$

At equilibrium $\frac{\partial \bar{\rho}(x,t)}{\partial t} = 0$, the Fokker-Planck equation admits a unique normalized steady-state solution, given by the Boltzmann distribution:

$$p_B(x) \propto \exp \left( \frac{2}{g_\infty^2} \left( 2 \, \Phi_\infty(x) + \frac{f_\infty}{2} \, x(t)^T x(t) \right) \right) \tag{81}$$

when the potential energy function, the drift coefficient, and the diffusion coefficient reach their time-independent steady states $\Phi_\infty(x)$, $f_\infty$ and $g_\infty$ at equilibrium. The Boltzmann distribution can then be written in terms of a coherent Boltzmann energy $\Phi_B$ considered in EBMs, as follows:

$$p_B(x) = \frac{e^{\Phi_B}}{Z} \tag{82}$$

where

$$\Phi_B(x) = \frac{4 \, \Phi_\infty(x) + f_\infty \, \|x\|^2}{g_\infty^2} \tag{83}$$

$\square$

and $Z = \int_\Omega e^{\Phi_B(x)} \, dx$ is the normalizing constant.

### C.6   Proof of Proposition 6

*Proof.* The variational loss function in (22) can be written as follows:

$$\mathcal{L}(\Phi,t) = \frac{1}{2} \, \mathbb{E}_{\rho(x|\bar{x},t) \, p_{\text{data}}(\bar{x})} \left[ \Phi(x) \left( \gamma(x,\bar{x},t) - \bar{\gamma}(x,\bar{x},t) \right) \right] + \frac{1}{2} \, \mathbb{E}_{\bar{\rho}(x,t)} \left[ \left\| \nabla_x \Phi(x) \right\|^2 \right] \tag{84}$$

where we have assumed, without loss of generality, that a normalized energy $\bar{E}_\theta(x,t) = 0$. For an unnormalized solution $\Phi(x)$, we can always obtain a normalization by subtracting its mean.

The optimal solution $\Phi$ of the functional (84) is given by the first-order optimality condition:

$$\mathcal{I}(\Phi, \Psi) = \frac{d}{d\epsilon} \mathcal{L}(\Phi(x) + \epsilon \Psi(x), t) \bigg|_{\epsilon=0} = 0 \tag{85}$$

which must hold for all trial function $\Psi$.

Taking the variational derivative of the particle flow objective (85) with respect to $\epsilon$, we have:

$$\begin{aligned}
\mathcal{I}(\Phi, \Psi) &= \frac{d}{d\epsilon} \mathcal{L}(\Phi + \epsilon \Psi) \bigg|_{\epsilon=0} \\
&= \frac{1}{2} \int_{\Omega \times \Omega} p_{\text{data}}(\bar{x}) \, \rho(x \mid \bar{x}, t) \left( \gamma(x,\bar{x},t) - \bar{\gamma}(x,\bar{x},t) \right) \frac{d}{d\epsilon} (\Phi + \epsilon \Psi) \, d\bar{x} \, dx \\
&\quad + \frac{1}{2} \int_\Omega \bar{\rho}(x,t) \frac{d}{d\epsilon} \left\| \nabla_x (\Phi + \epsilon \Psi) \right\|^2 dx \\
&= \frac{1}{2} \int_{\Omega \times \Omega} p_{\text{data}}(\bar{x}) \, \rho(x \mid \bar{x}, t) \left( \gamma(x,\bar{x},t) - \bar{\gamma}(x,\bar{x},t) \right) \Psi \, d\bar{x} \, dx \; + \int_\Omega \bar{\rho}(x,t) \, \nabla_x \Phi \cdot \nabla_x \Psi \, dx
\end{aligned} \tag{86}$$

Given that $\Phi \in \mathcal{H}_0^1(\Omega; \rho)$, its value vanishes on the boundary $\partial\Omega$. Therefore, the second summand of the last expression in (86) can be written, via multivariate integration by parts, as

$$\int_\Omega \bar{\rho}(x, t) \nabla_x \Phi \cdot \nabla_x \Psi = -\int_\Omega \nabla_x \cdot \left(\bar{\rho}(x, t) \nabla_x \Phi\right) \Psi \, dx \tag{87}$$

By substituting (87) into (86), we get

$$\mathcal{I}(\Phi, \Psi) = \int_\Omega \left(\frac{1}{2} \int_\Omega p_{\text{data}}(\bar{x}) \, \rho(x \mid \bar{x}, t) \left(\gamma(x, \bar{x}, t) - \bar{\gamma}(x, \bar{x}, t)\right) d\bar{x} - \int_\Omega \nabla_x \cdot \left(\bar{\rho}(x, t) \nabla_x \Phi\right)\right) \Psi \, dx \tag{88}$$

and equating it to zero, we obtain the weak formulation (21) of the density-weighted Poisson's equation.

Given that the Poincaré inequality (23) holds, Theorem 2.2 of Laugesen et al. (2015) presents a rigorous proof of existence and uniqueness for the solution of the weak formulation (21), based on the Hilbert-space form of the Riesz representation theorem. □

# D    Experimental Details

## D.1    Model architecture

Our network architectures for the autonomous and time-varying VAPO models are based on the WideResNet (Zagoruyko & Komodakis, 2016) and the U-Net (Ronneberger et al., 2015), respectively. For WideResNet, we include a spectral regularization loss during model training to penalize the spectral norm of the convolutional layer. Also, we apply weight normalization with data-dependent initialization (Salimans & Kingma, 2016) on the convolutional layers to further regularize the model's output. Our WideResNet architecture adopts the model hyperparameters reported by Xiao et al. (2021a). For U-Net, we remove the final scale-by-sigma operation (Kim et al., 2021; Song et al., 2021) and replace it with the Euclidean norm $\frac{1}{2}\|x - f_\theta(x)\|^2$ computed between the input $x(t)$ and the output of the U-Net $f_\theta(x)$. Our U-Net architecture adopts the hyperparameters used by Lipman et al. (2023). In both the WideResNet and U-Net models, we replace LeakyReLU activations with Gaussian Error Linear Unit (GELU) activations (Hendrycks & Gimpel, 2017), which we find improves training stability and convergence.

## D.2    Training

We use the Lamb optimizer (You et al., 2020) and a learning rate of $10^{-3}$ for all the experiments. We find that Lamb performs better than Adam over large learning rates. We use a batch size of 128 and 64 for training CIFAR-10 and CelebA, respectively. For all experiments, we set a cutoff time of $t_{\max} = 1 - 10^{-5}$, a terminal time of $t_{\mathrm{end}} = 1$, a decay exponent of $\kappa = 1.5$, and a spectral gap constant of $\eta = 10^{-4}$ during training. Here, the mean and standard deviation scheduling functions $\mu(t) = t$ and $\sigma(t) = 1 - t$ follow those defined by the OT-FM path. All models are trained on a single NVIDIA A100 (80GB) GPU until the FID scores, computed on 5k samples, no longer show improvement. We observe that the models converge within 300k training iterations.

## D.3    Numerical Solver

In our experiments, the default ODE solver is the black-box solver from the SciPy library using the RK45 method (Dormand & Prince, 1980), following the approach of Xu et al. (2022). We allow additional ODE iterations to further refine the samples in regions of high likelihood, which we observe improves the quality of the generated images. This is achieved by extending the time horizon; our experiments indicate that setting the terminal time to $t_{\mathrm{end}} = 1.575$ yields the best ODE sampling results.

## D.4    Datasets

We conduct our experiments using the CIFAR-10 (Krizhevsky, 2009) and CelebA (Liu et al., 2015) datasets. CIFAR-10 consists of $50,000$ training images and $10,000$ test images at a resolution of $32 \times 32$. The CelebA dataset contains $202,599$ face images, with $162,770$ used for training and $19,962$ for testing. Each image is first cropped to $178 \times 178$ before being resized to $64 \times 64$. During resizing, we enable anti-aliasing by setting the antialias parameter to True. Additionally, we apply random horizontal flipping as a data augmentation technique.

## D.5    Quantitative Evaluation

We employ the FID and inception scores as quantitative evaluation metrics for assessing the quality of generated samples. For CIFAR-10, the FID is computed between $50,000$ samples and the pre-computed statistics from the training set, following Heusel et al. (2017). For CelebA $64 \times 64$, we adopt the setting from Song & Ermon (2020), computing the FID between $5,000$ samples and the pre-computed statistics from the test set. For model selection, we follow Song et al. (2021), selecting the checkpoint with the lowest FID score, computed on $2,500$ samples every $10,000$ iterations.

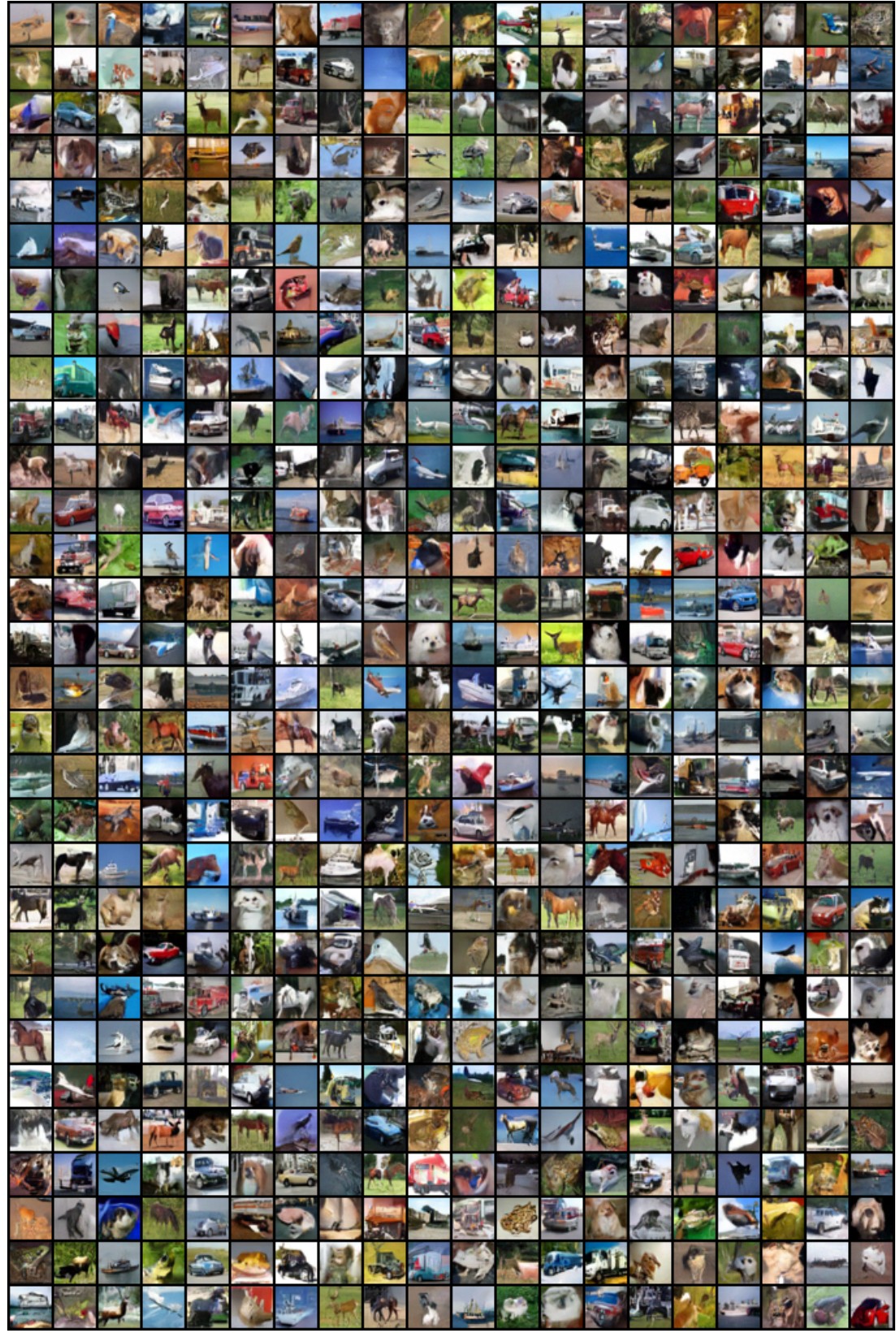

Figure 7: Additional uncurated samples on unconditional CIFAR-10 32 × 32.

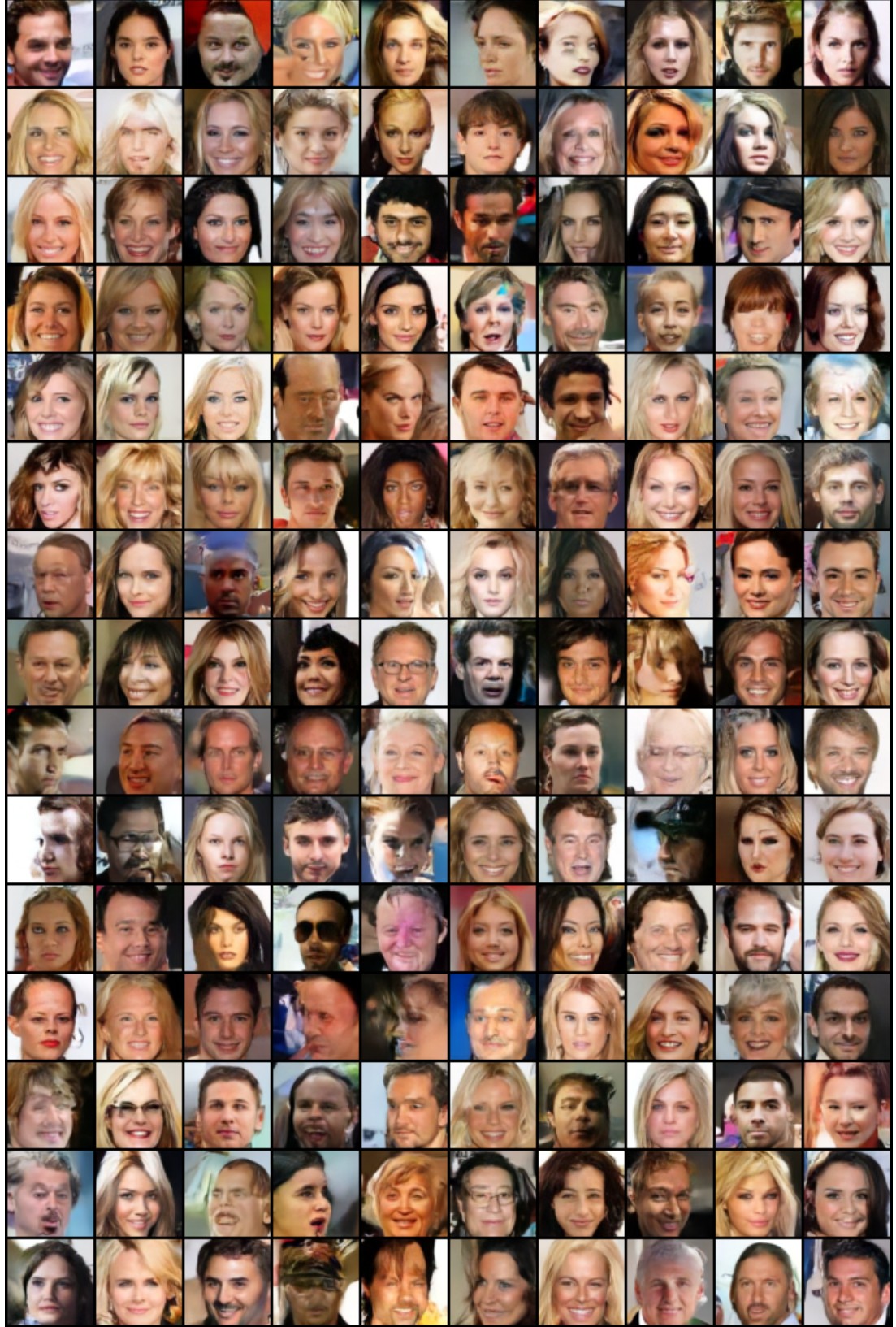

Figure 8: Additional uncurated samples on unconditional CelebA 64 × 64.

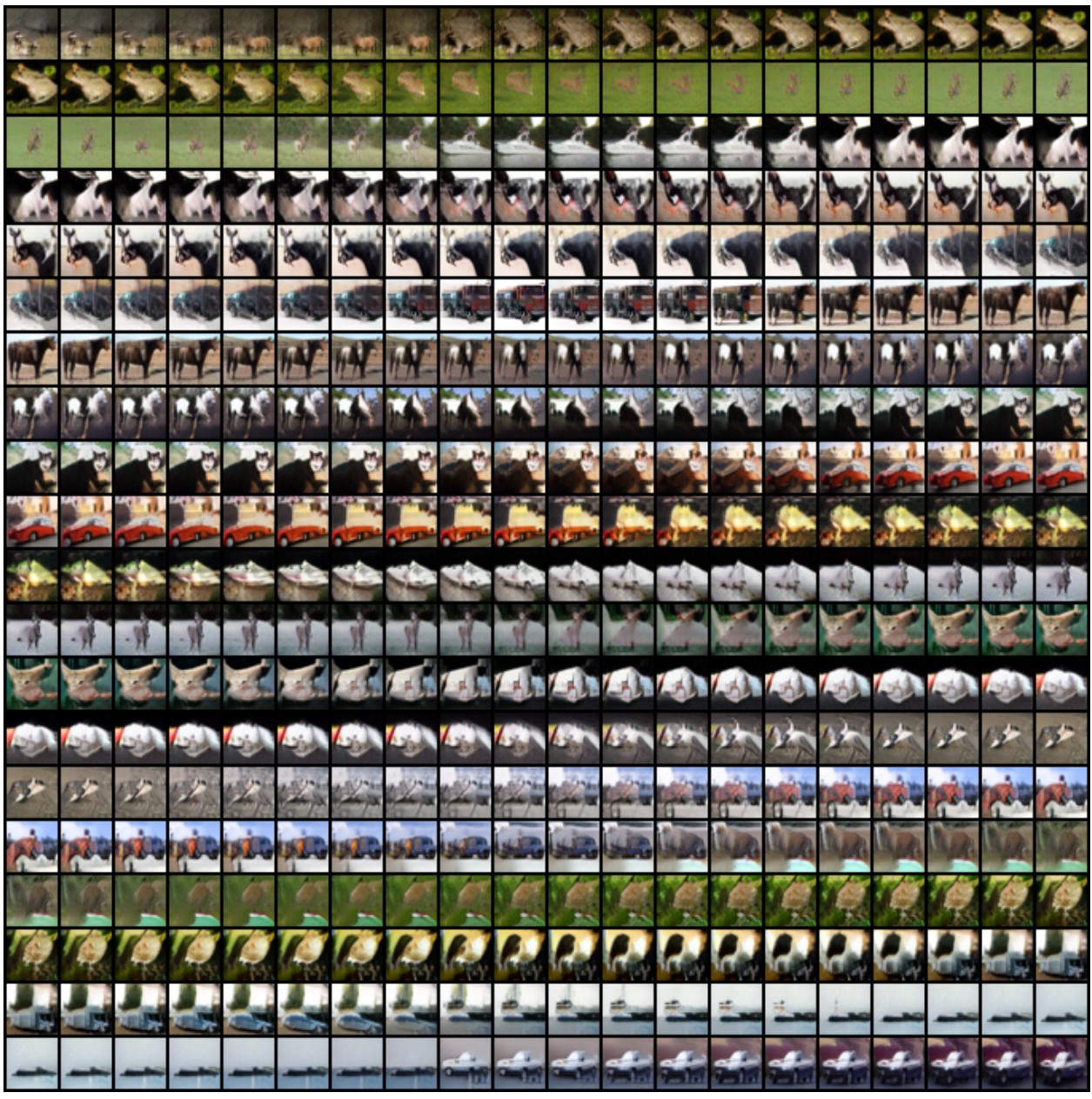

Figure 9: Additional interpolation results on unconditional CIFAR-10 $32 \times 32$.

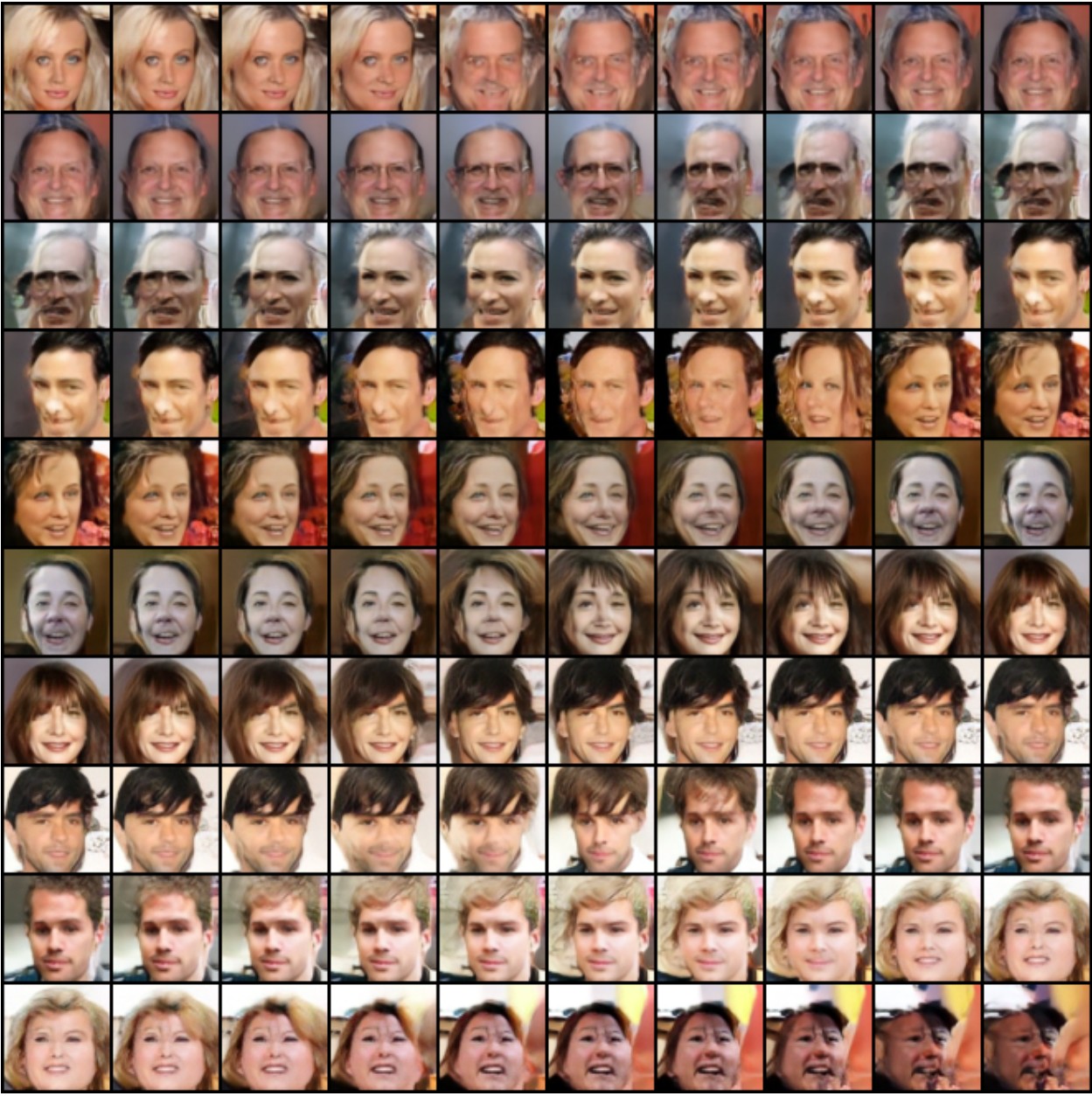

Figure 10: Additional interpolation results on unconditional CelebA $64 \times 64$.

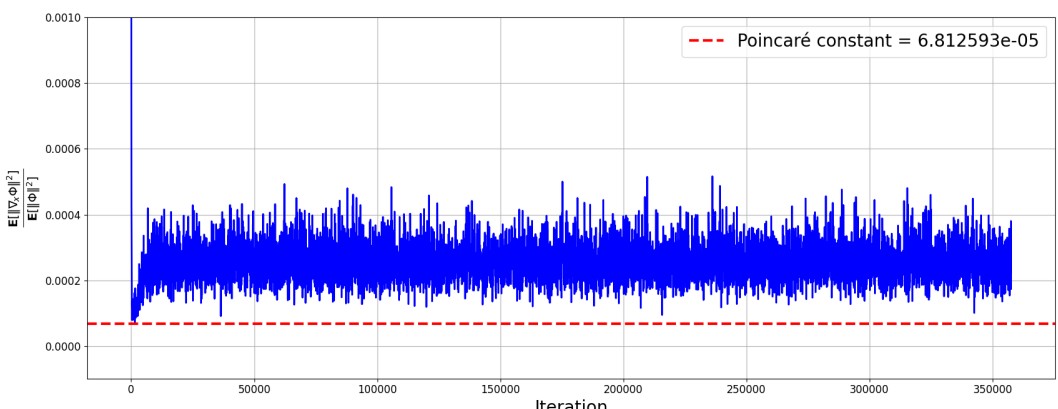

Figure 11: Validation of a Poincaré lower bound using the ratio of the gradient norm to the energy norm on CIFAR-10.

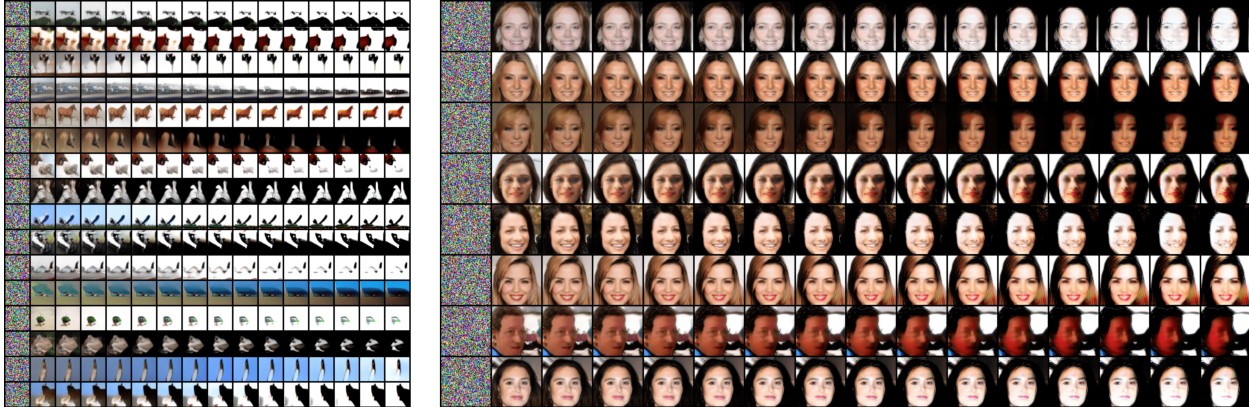

Figure 12: Long-run ODE (RK45) sampling using autonomous potential energy $\Phi(x)$ on CIFAR-10 (left) and CelebA (right).

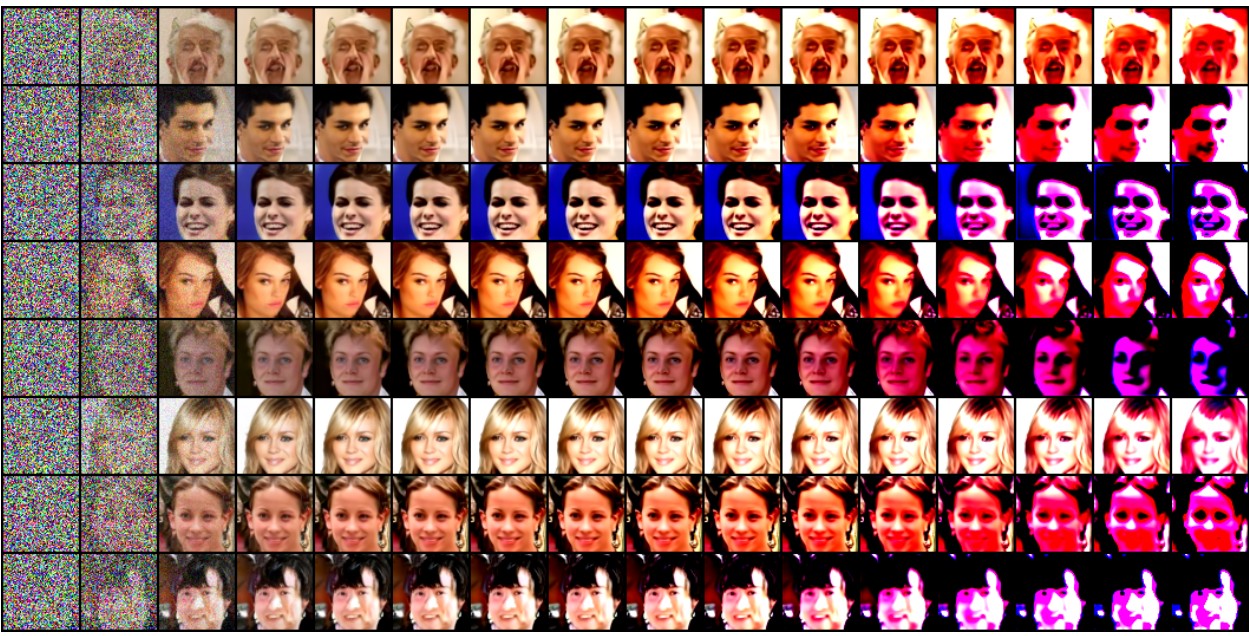

Figure 13: Long-run ODE (RK45) sampling using time-varying potential energy $\Phi(x, t)$ on CelebA.

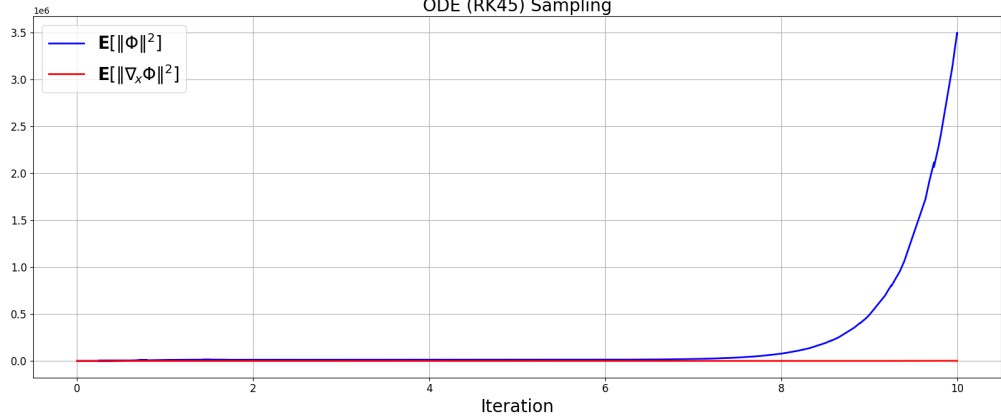

Figure 14: Validation of the convergence of gradient norm and energy norm in long-run ODE (RK45) sampling on CelebA.

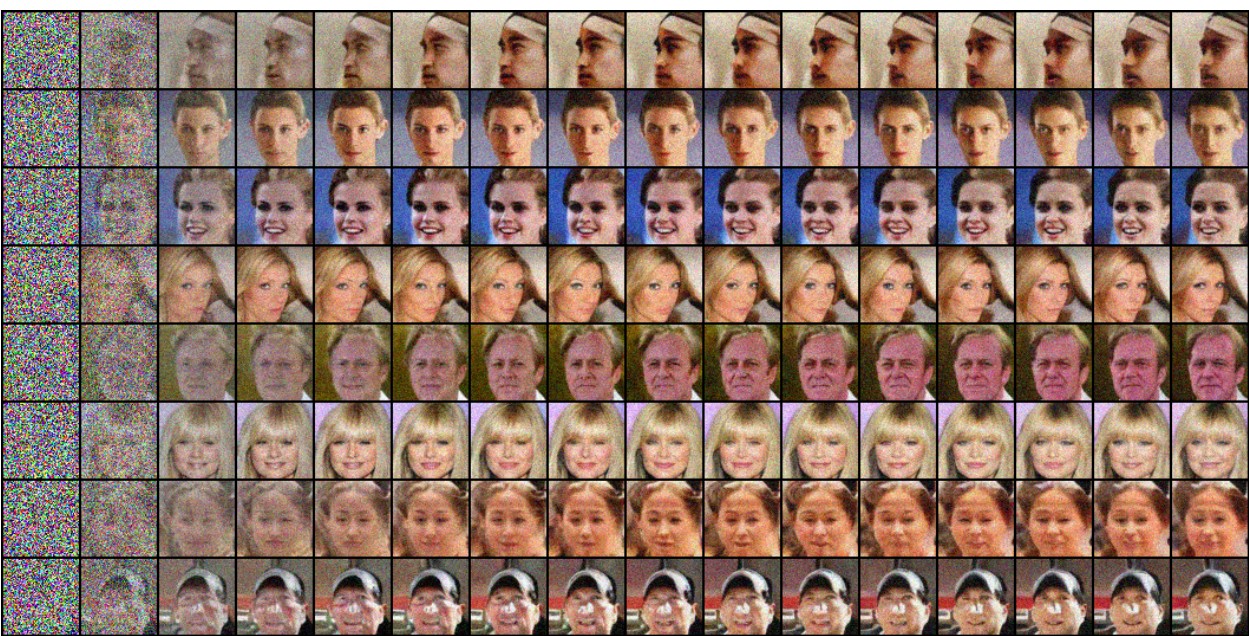

Figure 15: Long-run SGLD sampling using the Boltzmann energy with $\lambda = 0.35$ on CelebA.

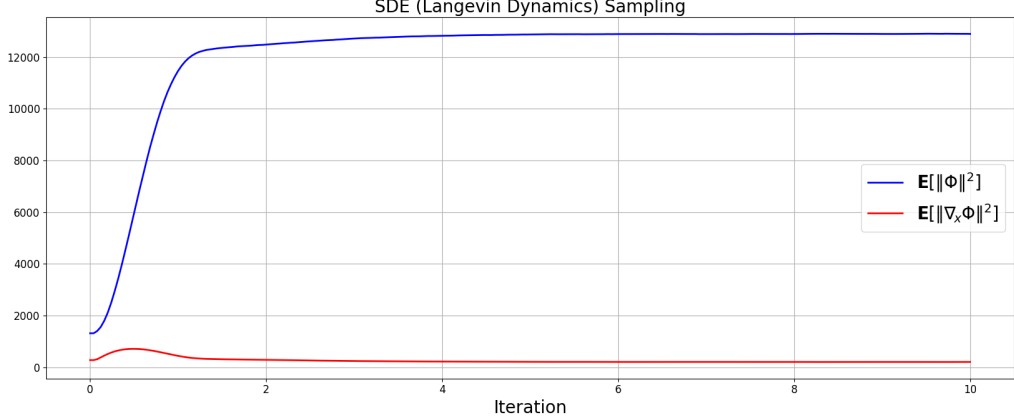

Figure 16: Validation of the convergence of the gradient norm and the energy norm in long-run SGLD sampling on CelebA.

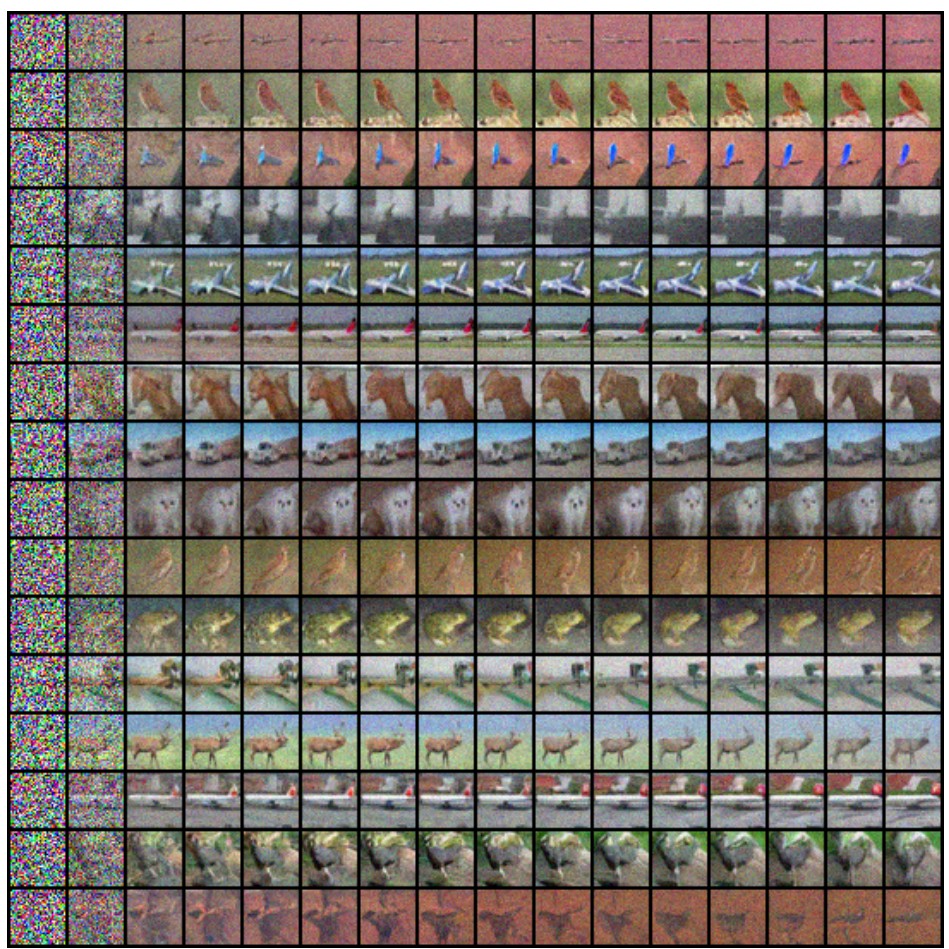

Figure 17: Long-run SGLD sampling using the Boltzmann energy with $\lambda = 0.35$ on CIFAR-10.

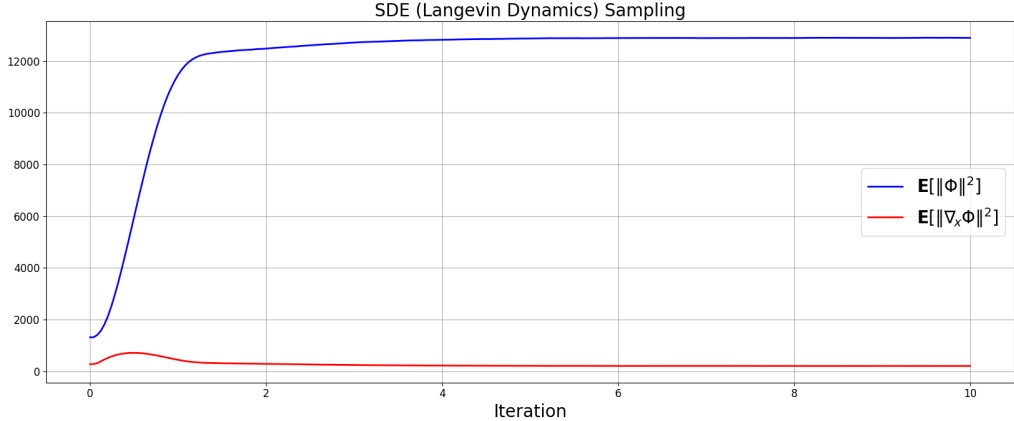

Figure 18: Validation of the convergence of the gradient norm and the energy norm in long-run SGLD sampling on CIFAR-10.

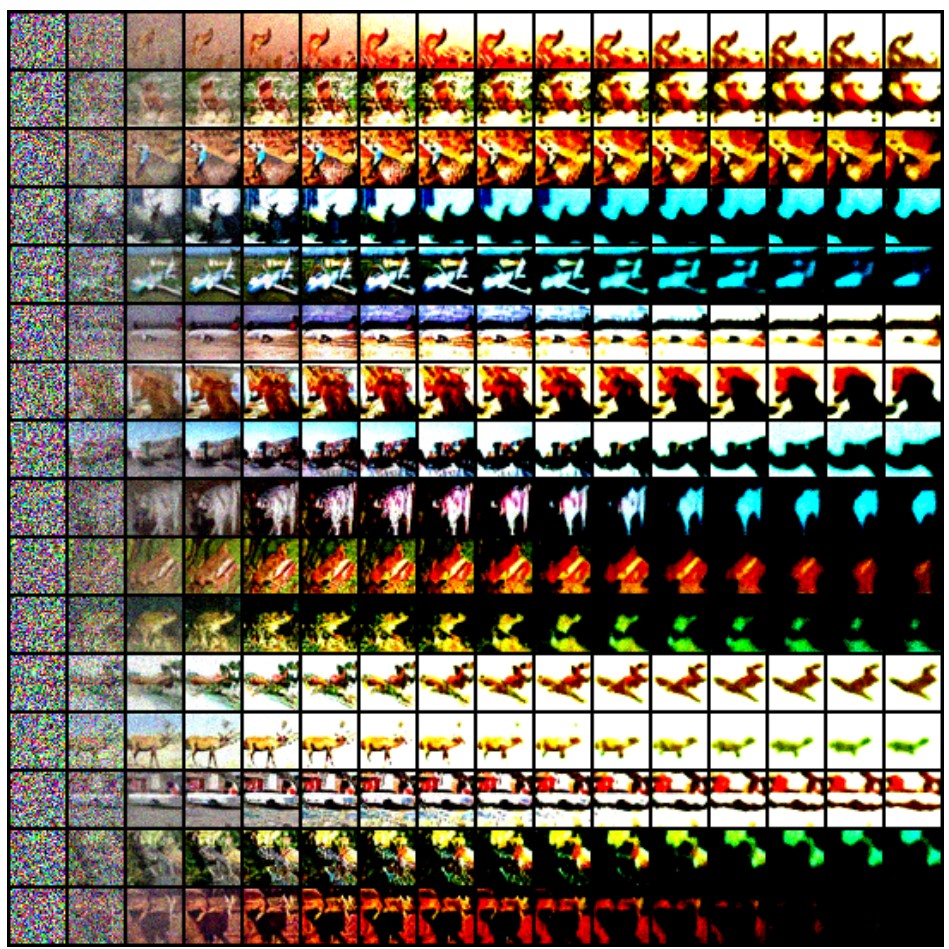

Figure 19: Long-run SGLD sampling using the Boltzmann energy on CIFAR-10 for loss configuration (D).

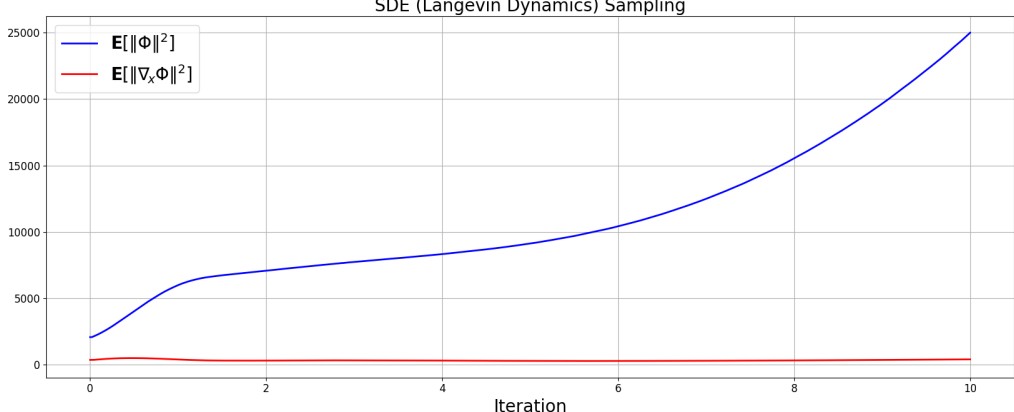

Figure 20: Validation of the convergence of gradient norm and energy norm in long-run SGLD sampling with loss configuration (D) on CIFAR-10.

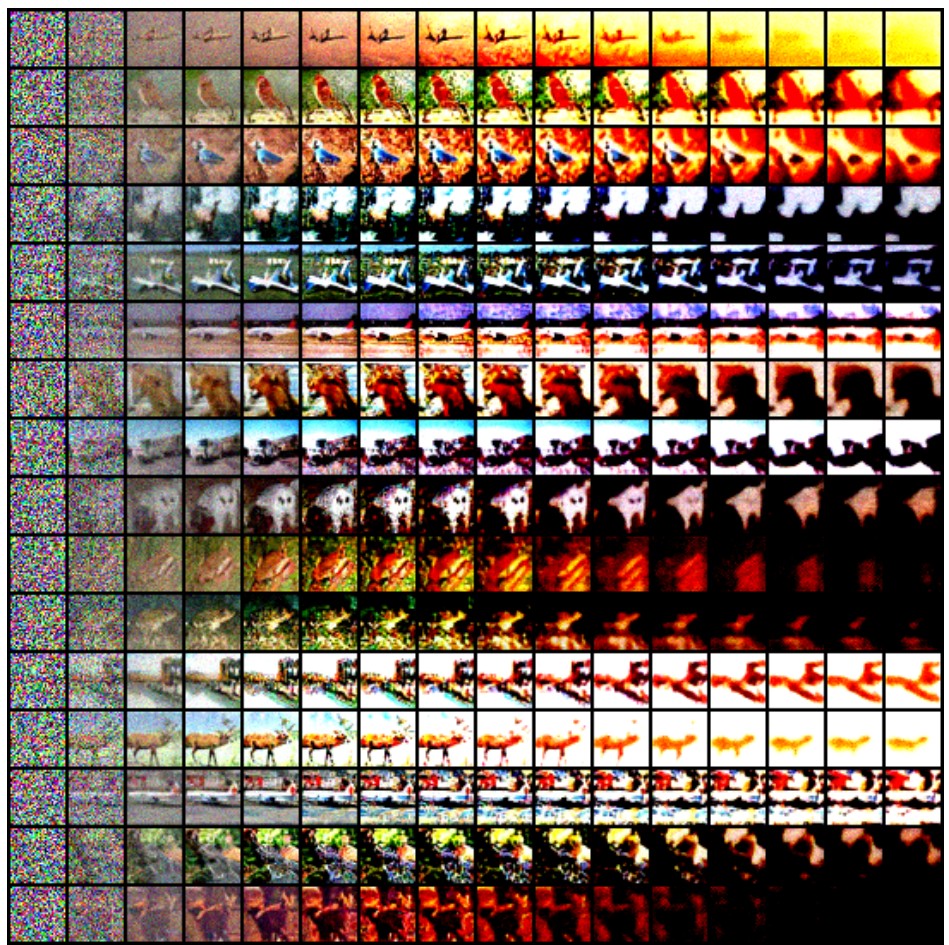

Figure 21: Long-run SGLD sampling using the Boltzmann energy with loss configuration (E) on CIFAR-10.

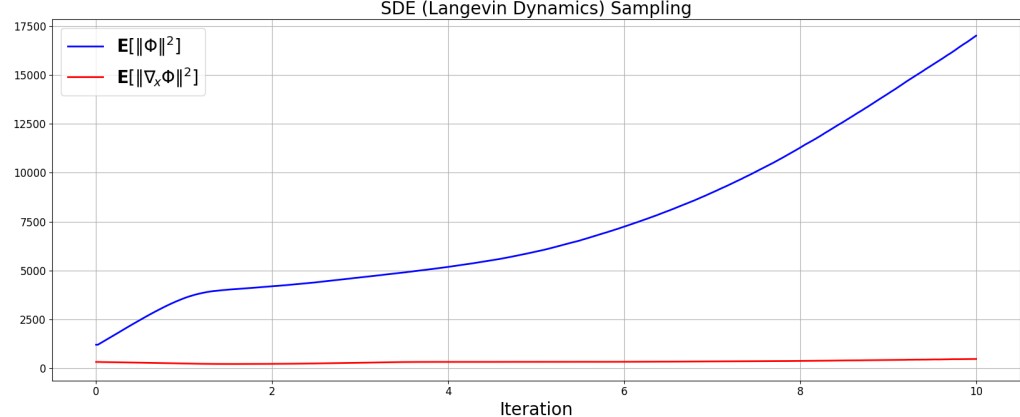

Figure 22: Validation of the convergence of gradient norm and energy norm in long-run SGLD sampling with loss configuration (E) on CIFAR-10.