# OpenReview forum: "Learning Energy-Based Generative Models via Potential Flow: A Variational Principle Approach to Probability Density Homotopy Matching"
_TMLR — Accepted by TMLR_

### Review · Reviewer_baM4 · 2025-02-22

**Summary Of Contributions:**

This paper introduces Variational Potential Flow Bayes (VPFB), a novel framework for training energy-based models (EBMs) that circumvents the computational challenges of traditional contrastive divergence training with implicit MCMC sampling. VPFB constructs a flow-driven density homotopy aligned with the data distribution through a variational loss that minimizes the KL divergence between flow-driven and marginal homotopies. Experiments on image generation, interpolation, out-of-distribution (OOD) detection, and compositional generation demonstrate competitive performance (e.g., FID of 7.08 on CIFAR-10 and strong AUROC scores for OOD detection).

**Audience:**

Yes

**Broader Impact Concerns:**

The paper's potential for misuse in generating harmful content is a concern.

**Claims And Evidence:**

Yes

**Requested Changes:**

- Provide additional theoretical proofs or experimental validation for key assumptions (e.g., boundary behavior of data distributions, validity of Poincaré-type inequalities)
   - Clarify the contradiction between experimental observations in Figure 6 (degraded sampling quality with increasing t) and the theoretical steady-state convergence assumption (when t_max = 0.9)
   - Include comprehensive computational metrics (total parameter count, wall-clock time, inference latency) to quantitatively demonstrate VPFB's computational efficiency advantages over conventional MCMC methods
   - Conduct thorough notation review to ensure consistent symbol usage. Typo: "VAPO" near Equation 24.

**Strengths And Weaknesses:**

Strengths:
   The integration of potential flow with variational principles is interesting, offering a fresh perspective on EBM training, addressing longstanding issues with MCMC sampling. Extensive experiments validate the framework across diverse tasks.
   Weaknesses:
   - In Equation (67), the assumption that $\(\bar{\rho}(x)\)$ vanishes at the boundary $\(\partial\Omega\)$ (i.e., compact support) may not hold for real-world data distributions such as natural images.
   - Proposition 6 postulates the existence of $\(\eta > 0\)$ such that ${E}_{\bar{\rho}}|\nabla \Phi|^2 >=\eta {E}{\bar{\rho}} \Phi^2$, but lacks empirical validation or theoretical justification for this Poincaré-type inequality assumption.
   - The contradiction between the theoretical steady-state convergence (assumed when $\(t_{\text{max}} = 0.9\)$) and the empirical observation in Figure 6 showing degraded sampling quality with increasing $\(t\)$ requires clarification.
   - Ambiguous notation: The manuscript frequently conflates $\(\rho_{\Phi}\)$ (flow-driven density) with $\(\bar{\rho}\)$ (target marginal density), necessitating clearer differentiation.
   - While the introduction positions VPFB as addressing MCMC's computational bottlenecks, the experimental section lacks detailed comparative analysis with baseline MCMC methods to substantiate computational efficiency claims. Since numerical SDE solver might be another computationally costly method.

---

> ### Author Response · Authors · 2025-03-03
>
> We sincerely thank Reviewer baM4 for a thorough and thoughtful review of our paper. We appreciate the positive recognition of our work and the valuable constructive feedback, which has helped us improve our manuscript.
>
> Our rebuttals will be provided in separate comments.
> Additional figures (R.1–R.6) used in rebuttals are provided in the supplementary materials under the *rebuttal* folder.
>
>
> **Broader Impact Concerns**: We appreciate your valuable feedback regarding the broader impact concerns of our approach. In response, we have carefully considered the potential risks associated with generative modeling advancements. To address these concerns, we will add a Broader Impact Statement to the paper, located after the conclusion section. This statement acknowledges the dual-use nature of generative models, including the risks of generating deepfakes and misleading content, and outlines potential safeguards to mitigate these issues.

---

> ### Author Response · Authors · 2025-03-03
> **Boundary Behavior of Data Distribution**
>
> # Requested Change:
>
> **Provide additional theoretical proofs or experimental validation for key assumptions (e.g., boundary behavior of data distributions, validity of Poincaré-type inequalities).**
>
> # Response:
>
> Equation (67) of Proposition 3 and Equation (88) of Proposition 6 applied the integration by parts
> \begin{align}
> \begin{split}
> \int_{\Omega} \nabla_{x} \cdot \big( \bar{\rho}(x) \ \nabla_{x} \Phi \big) \ \Psi \ dx = - \int_{\Omega} \bar{\rho}(x) \ \nabla_{x} \Phi \cdot \nabla_{x} \Psi + \int_{\partial \Omega} \bar{\rho}(x) \ \Psi \ \nabla_{x} \Phi \cdot \hat{n} \ dx
> \end{split}
> \end{align}
> where $\hat{n}$ is the outward unit normal vector to the boundary.
> For the final integral in the integration by parts
> \begin{align}
> \begin{split}
> \int_{\partial \Omega} \bar{\rho}(x) \ \Psi \ \nabla_{x} \Phi \cdot \hat{n} \ dx = 0
> \end{split}
> \end{align}
> to be zero, we require either that $\Psi$ or $\nabla_{x} \Phi \cdot \hat{n}$ vanishes on the boundary $\partial \Omega$ of some open bounded set $\Omega \subset \mathbb{R}^n$.
> A common technique used by Deep Ritz (E & Yu, 2018) to enforce the boundary condition is by introduce additional loss
> \begin{align}
> \begin{split}
> \int_{\partial \Omega} \bar{\rho}(x) \ \nabla_{x} \Phi \cdot \hat{n} \ dx = 0
> \quad \text{or} \quad
> \int_{\partial \Omega} \bar{\rho}(x) \ \Psi \ dx = 0
> \end{split}
> \end{align}
> However, since these losses are evaluated on the boundary $\partial \Omega$, it is necessary to obtain a batch of boundary samples $x \in \partial \Omega$ for neural network training, in addition to the interior samples $x \in \Omega$. This requirement introduces additional computational overhead.
> Alternatively, in the paper, we assume that the marginal density $\bar{\rho}(x)$ vanishes on the boundary $\partial \Omega$, under which $\int_{\partial \Omega} \bar{\rho}(x) \ \Psi \ \nabla_{x} \Phi \cdot \hat{n} \ dx = 0$ also holds.
>
> Although the pixel values of the training images $\bar{x}$ are normalized be within $[-1,1]^n$ during our result implementation, we define $x \in \Omega$ to be on an open bounded subset $\Omega \subset \mathbb{R}^n$. Here, we approximate the real data likelihood ${p_{\text{data}}}(\bar{x})$ via the marginal density
> \begin{align}
> \bar{\rho}(x,t) = \int_{\Omega} {p_{\text{data}}}(\bar{x}) \ {\rho}(x \mid \bar{x},t) \ d\bar{x}
> \end{align}
> using the conditional homotopy ${\rho}(x \mid \bar{x},t)$ (perturbation kernel)
> \begin{align}
> \begin{split}
> {\rho}(x \mid \bar{x},t) = \frac{e^{h(x \mid \bar{x},t)}}{\int_{\Omega} e^{h(x \mid \bar{x},t)} \ dx}  = \mathcal{N}\big(x; \mu(t) \ \bar{x}, \sigma(t)^2 I\big)
> \end{split}
> \end{align}
> where
> \begin{align}
> \begin{split}
> h(x \mid \bar{x}, t) = \alpha(t) \ \log q(x) + \beta(t) \ \log p(\bar{x} \mid x) = -\frac{\alpha(t)}{2 \ \omega^2} \ \|x\|^2 - \frac{\beta(t)}{2 \ \nu^2} \ \|\bar{x} - x\|^2
> \end{split}
> \end{align}
> Therefore, the open set $\Omega$ can be defined sufficiently large so that the conditional homotopy ${\rho}(x \mid \bar{x},t)$ approaches zero at boundary $\partial \Omega$. Since the conditional homotopy is a Gaussian perturbation kernel in real space, this condition is automatically satisfied.  As a result, $\bar{\rho}(x,t)$ would also approaches zero at boundary. We will incorporate this additional justification for the boundary (compact support) assumption in our paper revision. We also apologize for the typo in Equation (5), where $f$ should be $h$.
>
> We acknowledge that generative approaches, such as Rissanen et al. (2023), that define images as a function $u(c)$, where $c \in [0,H]\times[0,W]$ represents the bounded image coordinates, and incorporate Neumann boundary conditions into the design of their heat dissipation perturbation kernel. However, in this work, we adopt the conventional formulation, where $x \in \mathbb{R}^n$ represents the pixel values of an image.
>
> **References**
>
> Weinan E and Bing Yu. The deep ritz method: A deep learning-based numerical algorithm for solving variational problems. Communications in Mathematics and Statistics, 6(1):1–12, 2018.
>
> Severi Rissanen, Markus Heinonen, and Arno Solin. Generative modelling with inverse heat dissipation. In The Eleventh International Conference on Learning Representations, 2023.

---

> > ### Comment · Reviewer_baM4 · 2025-03-12
> >
> > I am satisfied with the authors' response regarding boundary behavior. Their additional justification for boundary behavior assumptions in the rebuttal is mathematically self-consistent and aligns with conventional practices in generative models.

---

> ### Author Response · Authors · 2025-03-03
> **Validity of Poincaré-Type Inequality**
>
> # Requested Change:
>
> **Provide additional theoretical proofs or experimental validation for key assumptions (e.g., boundary behavior of data distributions, validity of Poincaré-type inequalities).**
>
> # Response:
>
> The existence of an unique solution to the variational loss function in Equation (22) require that the Poincaré inequality of Equation (23) holds (Laugesen et al., 2015, Theorem 2.2). However, demonstrating that a model architecture used to parameterize the energy $\Phi$ satisfies Equation (23) from first principles could be challenging due to the complexity of neural networks. On this account, we introduce an additional regularization loss
> $\eta \times {\mathbb{E}}_{\bar{\rho}(x,t)} \big[ \| \Phi(x,t) \|^{2} \big]$
> to the final VPFB loss function in Equation (25) to enforce the Poincaré inequality during training.
>
> To validate the existence of a positive Poincaré constant $\eta$, Figure R.1 (attached in the supplementary materials) plots the ratio of the gradient norm
> ${\mathbb{E}} \big[\| \nabla_x \Phi(x,t)  \|^{2} \big]$
> to the energy norm
> ${\mathbb{E}} \big[\\| \Phi(x,t)  \|^{2} \big]$
> throughout the CIFAR-10 training iterations using a U-Net-parameterized energy function, but without incorporating the additional Poincaré regularization loss. From this plot, we identify the Poincaré constant as the lower bound $\eta = 6.81 \times 10^{-5}$ of the gradient-to-energy ratio. Nonetheless, our experiments indicate that the existence and magnitude of such an unenforced Poincaré constant vary across different neural architectures. For completeness, we incorporate the Poincaré regularization with a small $\eta$ for both the WideResNet and Unet models, which we fine-tune during training for optimal results.

---

> > ### Comment · Reviewer_baM4 · 2025-03-12
> >
> > I am satisfied with the authors' response to the Poincaré inequality. I appreciate their pragmatic solution through the introduction of an additional regularization term. The supplementary experimental results demonstrating the existence of Poincaré constants partially address my request for empirical validation, though the authors appropriately note that the magnitude and existence of unenforced Poincaré constants vary across different neural architectures.

---

> ### Author Response · Authors · 2025-03-03
> **Contradiction between Experimental Observations and Theoretical Steady-State Convergence**
>
> # Requested Change:
>
> **Clarify the contradiction between experimental observations in Figure 6 (degraded sampling quality with increasing t) and the theoretical steady-state convergence assumption (when ${t_\text{max}}$ = 0.9).**
>
> # Response:
>
> To clarify, Figure 6 presents long-run ODE sampling using the autonomous energy model parameterized by WideResNet. We apologize for omitting this detail. Additionally, Figure R.2 illustrates long-run ODE sampling using the time-varying energy model parameterized by Unet. The results indicate a similar deterioration in image quality over extended time periods, albeit to a greater extent compared to Figure 6. Figure R.3 plots the gradient norm $\mathbb{E} \big[ \|\nabla_{x} \Phi(x,t)\|^{2} \big]$ and the energy norm $\mathbb{E} \big[ \| \Phi(x,t)\|^{2} \big]$, neither of which exhibit convergence.
> As reported by Agoritsas et al. (2023) and Nijkamp et al. (2020), these results are consistent with those observed in EBMs trained using non-convergent short-run MCMC. This issue arises from the inherent difficulty neural network models face in learning complex energy landscapes in high-dimensional spaces, such as images, often leading to the formation of sharp local minima. Consequently, ODE sampling may become trapped in these minima, resulting in mode collapse and poor mixing.
>
> To provide a more detailed explanation, conventional EBM training often relies on convergent (long-run) MCMC to thoroughly explore the image space and smooth out local minima by assigning appropriate energy values to these regions. Instead of relying on the expensive and unstable fully thermalized MCMC, we employ a conditional homotopy ${\rho}(x \mid \bar{x},t)$ (perturbation kernel) to perturb the samples through a carefully designed dynamic path during training. Unlike convergent MCMC, this specific dynamic path does not fully explore low-density regions in high-dimensional spaces. As a result, many regions that remain unseen during training may correspond to poorly modeled areas in the energy landscape. This limitation causes the EBM to overfit to high-density regions in the training data, leading to artifacts such as excessive saturation and a loss of background details. Nonetheless, this issue of the curse of dimensionality is less problematic in low-dimensional 2D spaces, where perturbations such as the OT path in flow matching sufficiently cover the space.
>
> To resolve mode collapse in Figure R.2 and non-convergence in Figure R.3, we replace the deterministic ODE sampler with the conventional stochastic Langevin dynamics for image generation, allowing us to sample from the Boltzmann energy $\Phi_{B}(x)$ in Equation (20) via
> \begin{align}
> \begin{split}
> x_{t+1} = x_t + \Delta_t \ \nabla_x \Phi_{B}(x_t) + \sqrt{2 \ \Delta_t} \ \epsilon
> \end{split}
> \end{align}
> where $\epsilon \sim \mathcal{N}(0, \lambda^2 I)$ denotes Gaussian noise with temperature scale (standard deviation) $\lambda$, and $\Delta_t$ denotes the step size. The stochastic disturbance introduced by the diffusive noise in Langevin dynamics helps the sampler to escape from local minima and improve mixing. Figure R.4 shows the long-run Langevin dynamics sampling results using the Unet-parameterized Boltzmann energy with $\lambda=0.35$, where it shows that the samples converge well to the stationary equilibrium without exhibiting mode collapse. Figure R.5 plots the gradient norm $\mathbb{E} \big[ \|\nabla_{x} \Phi(x,t)\|^{2} \big]$ and the energy norm $\mathbb{E} \big[ \| \Phi(x,t)\|^{2} \big]$, where it shows that the gradient norm converges to zero and the energy norm asymptotically reaches a steady state, thus indicating thermalization. These results demonstrate that steady-state convergence is achievable with a stochastic sampler.
> Nonetheless, the temperature scale $\lambda$ needs to be fine-tuned to produce optimal results. Figure R.6 shows that setting $\lambda=0.5$ introduces undesired artifacts, possibly due to poorly modeled regions in the energy landscape. We will include these additional results for the long-run ODE and Langevin dynamics sampling in the revised paper.
>
> **References**
>
> Elisabeth Agoritsas, Giovanni Catania, Aurélien Decelle, and Beatriz Seoane. Explaining the effects of non-convergent MCMC in the training of energy-based models. In Proceedings of the 40th International Conference on Machine Learning, volume 202 of Proceedings of Machine Learning Research, pp. 322–336. PMLR, 2023.
>
> Erik Nijkamp, Mitch Hill, Tian Han, Song-Chun Zhu, and Ying Nian Wu. On the anatomy of mcmc-based maximum likelihood learning of energy-based models. Proceedings of the AAAI Conference on Artificial Intelligence, 34(04):5272–5280, 2020.

---

> > ### Comment · Reviewer_baM4 · 2025-03-12
> > **concerns about the image quality degradation**
> >
> > I still have concerns about the image quality degradation shown in Figure 6. The authors attribute the image quality degradation in Figure 6 and supplementary Figure R.2 to the inherent challenges of neural networks learning complex energy landscapes in high-dimensional spaces, leading to sharp local minima that cause mode collapse and mixing deficiencies in ODE sampling. While this explanation is theoretically plausible and consistent with existing literature, the authors' own observation that replacing deterministic ODE samplers with MCMC enables proper convergence to steady-state equilibrium without mode collapse raises questions about whether this indicates comparative robustness limitations in VPFB.

---

> ### Author Response · Authors · 2025-03-09
> **Quantitative Metrics to Demonstrate Computational Efficiency**
>
> # Requested Change:
>
> **Include comprehensive computational metrics (total parameter count, wall-clock time, inference latency) to quantitatively demonstrate VPFB's computational efficiency advantages over conventional MCMC methods.**
>
> # Response:
>
> The table below compares the inference-time computational efficiency of VPFB against recent EBM baselines:
>
> | Method | Numerical Approach | Number of Model Parameters | Number of Sampling Steps | Wall-Clock Sampling Time (s) | Inference Latency (ms) | FID ↓ |
> | :------------ | :------------ | :------------ | :------------ | :------------ | :------------ | :------------ |
> | VAEBM (Xiao et al., 2021) | SGLD (MCMC) | 135.1M | 16 | 21.3 | 13.31 | 12.16 |
> | CoopFlow (Xie et al., 2022) | SGLD (MCMC) | 45.9M | 30 | 2.5 | 0.833 | 15.80 |
> | DRL (Gao et al., 2021) | SGLD (MCMC) | 34.8M | 180 | 23.9 | 1.328 | 9.58 |
> | CDRL (Zhu et al., 2024) | SGLD (MCMC) | 38.6M | 90 | 12.2 | 1.356 | 4.31 |
> | VPFB-base (Ours) | RK45 (ODE Solver) | 38.3M | 74 | 9.22 | 1.246 | 9.33 |
> | VPFB (Ours) | RK45 (ODE Solver) | 61.5M | 74 | 14.6 | 1.968 | 7.08 |
>
> Here, we included an additional smaller (base) model (VPFB-small) with fewer parameters but a higher FID score. SGLD denotes the Stochastic Gradient Langevin Dynamics.
> Overall, the table highlights the computational efficiency of our proposed method compared to strong EBM baselines while achieving competitive FID scores, albeit falling short of the state-of-the-art CDRL model.
>
> We thank reviewer for this suggestion, and will incorporate this additional table into our revised paper.

---

> > ### Comment · Reviewer_baM4 · 2025-03-12
> >
> > I am satisfied with the additional experimental results provided by the authors. I suggest including this table in the revised paper, along with detailed specifications of the VPFB-small model. I am satisfied with the response to the symbol consistency.

---

> ### Author Response · Authors · 2025-03-09
> **Thorough Notation Review to Ensure Symbol Consistency**
>
> # Requested Change:
>
> **Conduct thorough notation review to ensure consistent symbol usage. Typo: "VAPO" near Equation 24.**
>
> # Response:
>
> We appreciate the reviewer’s feedback regarding notation clarity.
>
> We will revise the paper to further emphasize the distinctions between the densities:
> (a)  $ \rho_{\text{data}}(\bar{x}) $ (**true data likelihood**),
> (b)  $ \rho(x \mid \bar{x},t) $ (**conditional density homotopy**),
> (c)  $ \bar{\rho}(x,t) $ (**marginal density homotopy**), and
> (d)  $ \rho_{\Phi}(x,t) $ (**flow-driven density homotopy**),
> ensuring that they are explicitly defined in the paper. Here, *density homotopy* refers to a density function that evolves over time. In the paper, we used $ \bar{x} $ and $ x $ to distinguish between the training data and the generative samples.
>
> To clarify, $ \bar{\rho}(x,t) $ in Equation (8) represents the intractable, time-varying marginal density that interpolates between the prior $ q(x) $ and the true data likelihood $ \rho_{\text{data}}(x) $. In contrast, $ \rho_{\Phi}(x,t) $ in Equation (12) denotes the time-varying density governed by the potential flow of Equation (11). In our proposed framework, we introduce this flow-driven density $ \rho_{\Phi}(x,t) $ to approximate the intractable marginal density $ \bar{\rho}(x,t) $ through homotopy matching, as described in Equations (13)–(14).
>
> Based on the reviewer’s suggestion, we will conduct a thorough notation review to ensure that all symbols are consistently defined and referenced throughout the text. Additionally, we apologize for the typo near Equation (24), where "VAPO" should be corrected to "VPFB".

---

> ### Author Response · Authors · 2025-03-16
> **Concerns about Image Quality Degradation**
>
> We appreciate the reviewer’s continued insights regarding the image quality degradation in Figure 6 and the necessity of MCMC samplers in achieving proper steady-state convergence. Given that ODE-based samplers lack the stochasticity required to escape sharp local minima, they are inherently more sensitive to poor energy landscape mixing compared to MCMC samplers such as Stochastic Gradient Langevin Dynamics. Nonetheless, the concern raised by the reviewer is valid: while the degradation stems from the inherent difficulty of learning complex high-dimensional energy landscapes, the susceptibility of ODE-based sampling to mode collapse suggests that the Gaussian perturbation used in our work may not sufficiently explore the high-dimensional data space.
>
> We acknowledge this limitation of our current framework, which is inherent to diffusion-based and flow-based models that rely on a time-dependent Gaussian perturbation kernel to generate training samples along a carefully structured dynamic probability path. Agoritsas et al. (2023) highlights that these probability paths behave similarly to short-run MCMC, which limits their ability to fully explore low-density regions in high-dimensional spaces. Consequently, representative energy values are not properly assigned in these regions that remain unseen during training. By replacing the entire VPFB loss with the flow matching loss of Lipman et al. (2023), our CIFAR-10 sampling results in Figures R.13 and R.14 (attached in the supplementary materials) show that the model fails to achieve a Boltzmann equilibrium, even when using stochastic Langevin dynamics sampler. In contrast, the model trained with our proposed VPFB loss successfully achieves steady-state convergence using a Langevin sampler, as shown in Figures R.9 and R.10. Nevertheless, to ensure proper Boltzmann convergence and mitigate mode collapse under a deterministic ODE sampling scheme, it is necessary to smooth out poorly modeled regions through improved data space exploration beyond the Gaussian probability path of diffusion and flow matching. Achieving this would require incorporating long-run (convergent) MCMC during training, albeit at the cost of the computational overhead inherent to traditional EBMs.
>
> As a potential future direction, we aim to address this limitation by integrating Gaussian probability paths with long-run MCMC sampling, following the techniques employed by Gao et al. (2021), Zhu et al. (2024), and Geng et al. (2024). However, these existing approaches primarily model the conditional distribution $p(x_t∣x_{t+1})$ along the reverse diffusion probability path rather than a marginal data distribution $p(x)$, which is the primary focus of our current work. Consequently, these conditional EBMs do not establish a direct connection to the Boltzmann distribution $p(x) \propto e^{\Phi(x)}$, which is fundamental to the theoretical underpinnings of marginal energy-based modeling. Therefore, adapting conditional modeling techniques to fully marginal EBMs would require substantial developments. To mitigate the computational cost of implicit long-run MCMC during training, we could also incorporate pre-sampled replay buffers or leverage model distillation techniques from a pre-trained convergent EBM.
>
> We thank the reviewer for raising this important point and will explore potential solutions to address this limitation in future work.
>
> **References**
>
> Elisabeth Agoritsas, Giovanni Catania, Aurélien Decelle, and Beatriz Seoane. Explaining the effects of non-convergent MCMC in the training of energy-based models. In Proceedings of the 40th International Conference on Machine Learning, volume 202 of Proceedings of Machine Learning Research, pp. 322–336. PMLR, 2023.
>
> Yaron Lipman, Ricky T. Q. Chen, Heli Ben-Hamu, Maximilian Nickel, and Matthew Le. Flow matching for generative modeling. In The Eleventh International Conference on Learning Representations, 2023.
>
> Ruiqi Gao, Yang Song, Ben Poole, Ying Nian Wu, and Diederik P Kingma. Learning energy-based models by diffusion recovery likelihood. In International Conference on Learning Representations, 2021.
>
> Yaxuan Zhu, Jianwen Xie, Ying Nian Wu, and Ruiqi Gao. Learning energy-based models by cooperative diffusion recovery likelihood. In The Twelfth International Conference on Learning Representations, 2024.
>
> Cong Geng, Tian Han, Peng-Tao Jiang, Hao Zhang, Jinwei Chen, Søren Hauberg, and Bo Li. Improving adversarial energy-based model via diffusion process. In Forty-first International Conference on Machine Learning, 2024.

---

### Review · Reviewer_ruyf · 2025-02-24

**Summary Of Contributions:**

The work introduces Variational Potential Flow Bayes (VPFB), which tackles well-known challenges in training energy-based models (EBMs), such as the reliance on MCMC sampling with its inherent issues like poor mode mixing and slow convergence. By employing the Deep Ritz method and aligning density homotopies via a variational loss that minimizes KL divergence, the approach is innovative in that it seeks to remove the need for auxiliary models and implicit MCMC entirely. This direct connection between energy-parameterized flow models and explicit marginal EBMs seems to fill a gap in current research.

**Audience:**

Yes

**Broader Impact Concerns:**

While the paper primarily focuses on a methodological contribution, there are several ethical implications that should be considered:

**Dual-Use Concerns** \
Advances in generative modeling, including improved image generation quality, can be misused to create deepfakes or misleading content. It is important to acknowledge this dual-use nature and discuss potential safeguards.

**Data Bias and Fairness** \
Although the work uses standard datasets, any generative model can inadvertently amplify biases present in the training data. It would be beneficial to discuss how the approach handles such biases and what steps could be taken to mitigate fairness concerns.

Including a Broader Impact Statement that addresses these points would strengthen the submission by proactively considering potential ethical issues associated with generative modeling advancements.

**Claims And Evidence:**

Yes

**Requested Changes:**

**Include Detailed Computational Cost Analysis** \
Proposed Adjustment: Provide a thorough comparison of training efficiency, including metrics such as the number of function evaluations (NFEs), training runtime, and resource usage, against MCMC-based EBMs and state-of-the-art diffusion/flow-based models. \
Impact: Critical – This is essential to substantiate the claim that VPFB avoids the computational overhead associated with MCMC sampling.

**Conduct Comprehensive Ablation Studies** \
Proposed Adjustment: Isolate and quantify the contributions of individual components (e.g., potential flow alignment, variational loss formulation via the Deep Ritz method) to assess their individual impact on performance. \
Impact: Critical – These studies are necessary to clearly demonstrate that each component meaningfully contributes to the overall effectiveness of the approach.

**Analyze and Address the Performance Gap** \
Proposed Adjustment: Provide a detailed analysis of the performance gap compared to state-of-the-art generative models (e.g., DDAEBM), including potential explanations and strategies for improvement. \
Impact: Strengthening – While not critical, this analysis would enhance the clarity of the contributions and provide valuable insights for further improvement.

**Expand the Experimental Scope** \
Proposed Adjustment: Evaluate the method on additional datasets or different data modalities (e.g., higher resolution images or non-image data) to better demonstrate the generality of the approach. \
Impact: Strengthening – Broader experiments would increase the work’s impact and validate its applicability beyond the current scope.

**Strengths And Weaknesses:**

**Strengths**
- Novel Framework \
The paper innovatively tackles longstanding challenges in training energy-based models (EBMs) by eliminating the reliance on computationally intensive MCMC sampling, which typically suffers from issues like poor mode mixing and slow convergence. Instead, it introduces a variational approach that combines the Deep Ritz method with density homotopies to enable efficient and principled training.

- Justification \
The introduction effectively motivates the proposed method by: 1. Clearly outlining the shortcomings of current EBM training techniques. 2. Discussing alternative generative modeling approaches (like flow-based models) and their limitations. 3. Positioning VPFB as a framework that retains the theoretical benefits of EBMs.

- Theoretical rigor \
The paper demonstrates strong theoretical contributions by: 1. Constructs a clear Bayesian marginal homotopy between a simple Gaussian prior and the data likelihood. 2. Formulating a potential flow governed by a conservative vector field, ensuring that the energy landscape is both well-defined and interpretable. 3. Employing the Deep Ritz method to derive a variational loss that minimizes the KL divergence.

**Weakness**
- Computational Cost Analysis Missing \
While the paper claims that VPFB reduces the computational burden by avoiding MCMC sampling, it does not provide a detailed comparison of training costs relative to traditional MCMC-based EBMs or state-of-the-art diffusion/flow-based models. The claim of avoiding computationally heavy MCMC sampling is not fully substantiated without detailed efficiency metrics.

- Performance Gap \
Although the empirical results are competitive, there is a notable performance gap compared to state-of-the-art generative models, including leading EBM methods like DDAEBM. This discrepancy raises concerns about the practical applicability of VPFB. A thorough analysis is needed to explain the performance gap.
Moreover, the number of function evaluations (NFEs) are not compared with baselines, which is critical for fair comparison.

- Limited Ablation Studies \
The paper does not include detailed ablation studies to isolate the contributions of individual components (e.g., potential flow alignment and the variational loss formulation). This omission makes it challenging to evaluate the specific impact and necessity of each component on the overall performance.

---

> ### Author Response · Authors · 2025-03-10
>
> We sincerely thank Reviewer ruyf for a thorough and thoughtful review of our paper. We appreciate the positive recognition of our work and the valuable constructive feedback, which has helped us improve our manuscript.
>
> Our rebuttals will be provided in separate comments. Additional figures (Figures R.9–R.14) used in the rebuttals are provided in the supplementary materials under the rebuttal folder.
>
> **Broader Impact Concerns:** We sincerely appreciate Reviewer ruyf's thoughtful feedback on the ethical implications of our research. We acknowledge the importance of **dual-use concerns** and **data bias and fairness** in generative modeling. In response, we will add a Broader Impact Statement after the conclusion to explicitly address potential risks, such as the misuse of our approach for generating harmful content like deepfakes and the amplification of biases present in training data. Additionally, we will discuss safeguards, including classifier-based guidance, fairness-aware training, and privacy-preserving techniques, to mitigate these risks effectively.

---

> ### Author Response · Authors · 2025-03-15
> **Include Detailed Computational Cost Analysis**
>
> # Requested Change:
>
> **Include Detailed Computational Cost Analysis**
>
> **Proposed Adjustment: Provide a thorough comparison of training efficiency, including metrics such as the number of function evaluations (NFEs), training runtime, and resource usage, against MCMC-based EBMs and state-of-the-art diffusion/flow-based models.**
>
> # Response:
>
> The table below compares the training-time computational efficiency of VPFB against recent EBM baselines:
>
> | Method | Perturbation/Sampling Approach | Number of Model Parameters | Memory Usage (GB) | Number of Training Iterations | Training Time (hours) | FID ↓ |
> | :------------ | :------------ | :------------ | :------------ | :------------ | :------------ | :------------ |
> | VAEBM (Xiao et al., 2021) | SGLD (MCMC) + Encoder Model (Variational Inference) + Replay Buffer | 135.9M | *129* | 25K | *414* | 12.19 |
> | DRL (Gao et al., 2021) | SGLD (MCMC) + Diffusion | 38.6M | 56 | 240K | 172 | 9.58 |
> | CLEL (Lee et al., 2023) | SGLD (MCMC) + Replay Buffer | 30.7M | 10 | 100K | 133 | 8.61 |
> | CDRL (Zhu et al., 2024) | SGLD (MCMC) + Initializer Model (Amortized Inference) + Diffusion | 34.8M | 69 | 400K | 144 | 4.31 |
> | VPFB-base (Ours) | Stationary-Enforced OT-FM | 38.3M | 35 | 300K | 48 | 9.33 |
> | VPFB (Ours) | Stationary-Enforced OT-FM | 61.5M | 60 | 300K | 112 | 7.08 |
>
> Here, we included an additional smaller model (VPFB-base) with fewer parameters but a higher FID score. The training time and GPU memory footprint is measured on a single A100 GPU of 80G memory. Italicized values represent estimates for the VAEBM model, referenced from Lee et al. (2023), based on experiments conducted with a smaller batch size, as the model cannot be trained with the prescribed batch sizes on a single GPU. SGLD denotes the Stochastic Gradient Langevin Dynamics, and OT-FM denotes the Optimal Transport path of Flow Matching (Lipman et al., 2023).
>
> Additionally, the table below compares the inference-time computational efficiency of VPFB against recent EBM baselines:
>
> | Method | Numerical Approach | Number of Model Parameters | Number of Sampling Steps | Inference Time (s) | Inference Latency (ms) | FID ↓ |
> | :------------ | :------------ | :------------ | :------------ | :------------ | :------------ | :------------ |
> | VAEBM (Xiao et al., 2021) | SGLD (MCMC) | 135.1M | 16 | 21.3 | 13.31 | 12.19 |
> | CoopFlow (Xie et al., 2022) | SGLD (MCMC) | 45.9M | 30 | 2.5 | 0.833 | 15.80 |
> | DRL (Gao et al., 2021) | SGLD (MCMC) | 34.8M | 180 | 23.9 | 1.328 | 9.58 |
> | CDRL (Zhu et al., 2024) | SGLD (MCMC) | 38.6M | 90 | 12.2 | 1.356 | 4.31 |
> | VPFB (Ours) | RK45 (ODE Solver) | 61.5M | 74 | 14.6 | 1.968 | 7.08 |
>
> Overall, these results suggest that our method provides improved computational efficiency in both training and inference compared to most strong EBM baselines while maintaining competitive FID scores, particularly for the base model, which has a parameter count similar to the baselines. Nonetheless, there remains a gap in FID performance relative to the state-of-the-art CDRL model.
>
> We thank the reviewer for this valuable comment, and will include these computational cost analyses in the revised paper.
>
> **References**
>
> Zhisheng Xiao, Karsten Kreis, Jan Kautz, and Arash Vahdat. VAEBM: A symbiosis between variational
> autoencoders and energy-based models. In International Conference on Learning Representations, 2021.
>
> Ruiqi Gao, Yang Song, Ben Poole, Ying Nian Wu, and Diederik P. Kingma. Learning energy-based models
> by diffusion recovery likelihood. In International Conference on Learning Representations, 2021.
>
> Hankook Lee, Jongheon Jeong, Sejun Park, and Jinwoo Shin. Guiding energy-based models via contrastive latent variables. In The Eleventh International Conference on Learning Representations, 2023.
>
> Yaron Lipman, Ricky T. Q. Chen, Heli Ben-Hamu, Maximilian Nickel, and Matthew Le. Flow matching for generative modeling. In The Eleventh International Conference on Learning Representations, 2023.
>
> Yaxuan Zhu, Jianwen Xie, Ying Nian Wu, and Ruiqi Gao. Learning energy-based models by cooperative
> diffusion recovery likelihood. In The Twelfth International Conference on Learning Representations, 2024.
>
> Jianwen Xie, Yaxuan Zhu, Jun Li, and Ping Li. A tale of two flows: Cooperative learning of langevin flow
> and normalizing flow toward energy-based models. arXiv preprint arXiv:2205.06924, 2022.

---

> ### Author Response · Authors · 2025-03-16
> **Conduct Comprehensive Ablation Studies**
>
> # Requested Change:
>
> **Conduct Comprehensive Ablation Studies**
>
> **Proposed Adjustment: Isolate and quantify the contributions of individual components (e.g., potential flow alignment, variational loss formulation via the Deep Ritz method) to assess their individual impact on performance.**
>
> # Response:
>
> We appreciate this important comment by the reviewer. In the table below, we perform an additional ablation study to isolate and quantify the impact of individual components in the proposed VPFB loss function. The ablation study is conducted using a smaller (base) model with 38.3M parameters and a reduced training batch size to accelerate training.
>
> | Loss Configurations | Model Configurations | Number of Model Parameters | Training Batch Size | FID ↓ |
> | :------------ | :------------ | :------------ | :------------ | :------------ |
> | (A) VPFB loss | Base | 38.3M | 64 | 9.45 |
> | (B) VPFB loss without covariance | Base | 38.3M | 64 | 13.10 |
> | (C) VPFB loss without cosine distance | Base | 38.3M | 64 | 11.32 |
> | (D) cosine distance $\rightarrow$ inner product | Base | 38.3M | 64 | 9.21 |
> | (E) VPFB loss $\rightarrow$ flow matching loss | Base | 38.3M | 64 | 8.89 |
>
> Notably, the FID scores increase (B) without the covariance loss and (C) the cosine distance gradient alignment, indicating that these loss components are essential to the VPFB training. We note that since (B) learns only the normalized gradient and not the energy magnitude, it requires careful tuning of denormalization during ODE sampling.
>
> Subsequently, we (D) replace the cosine distance with inner product, and (E) replace the entire VPFB loss with the flow matching loss of Lipman et al. (2023). Although these loss configurations yield better FID performances, Figures R.11-R.12 and Figures R.13-R.14 show that they do not achieve Boltzmann equilibrium via stochastic Langevin dynamics sampling. Removing the cosine distance in (D) eliminates the scale invariance of cosine similarity, leading to large variations in gradient magnitudes that disrupt the stability of energy required for steady-state convergence. Similarly, the flow matching loss in (E) does not inherently enforce Boltzmann stationarity. Applying the same weighting $w(t) = (1-t)^2$ to the flow matching loss would halt the learning of the gradient field $v(x,t)=\nabla_x \Phi(x,t)$ near $t=1$. For these reasons, we do not adopt configurations (D) and (E) in our loss framework, despite the lower FID scores.
>
> In contrast, the variational nature of the covariance loss allows it to enforce that the energy values $\Phi(x,t)$ remain stationary near $t=1$ without impeding learning. This covariance loss is fundamental as it corresponds to the Fokker-Planck dynamics (density evolution) $\frac{\partial {\rho_{\Phi}}(x,t)}{\partial t}$. More importantly, its adaptability to weighting ensures a proper establishment of the stationary Boltzmann distribution.
>
> Additionally, the table below presents an ablation study to assess the impact of different training hyperparameter settings.
>
> | Training Hyperparameters | Model Configurations | Number of Model Parameters | Training Batch Size | FID ↓ |
> | :------------ | :------------ | :------------ | :------------ | :------------ |
> | $t_\text{max} = 0.9, t_\text{end} = 2, w(t) = (1-t)^2$ | Base | 38.3M | 64 | 11.03 |
> | cutoff time $t_\text{max}: 0.9 \rightarrow 1 - 10^{-5}$ | Base | 38.3M | 64 | 10.12 |
> | terminal time $t_\text{end}: 2 \rightarrow 1$ | Base | 38.3M | 64 | 9.57 |
> | loss weighting function $w(t): (1-t)^2 \rightarrow (1-t)^{1.5}$ | Base | 38.3M | 64 | 9.45 |
>
> In particular, we varies the cutoff time from $t_\text{max} = 0.9$ to $t_\text{max} = 1 - 10^{-5}$ and the terminal time from $t_\text{end} = 2$ to $t_\text{end} = 1$, bringing both these values closer to 1. Subsequently, we decrease scaling exponent in the loss weighting function from $w(t) = (1-t)^2$ to $w(t) = (1-t)^{1.5}$. Our results above indicate that the FID performances improve for models trained under these training hyperparameter settings. Figures R.9 and R.10 (included in the supplementary materials) show that, under the updated hyperparameters, the model maintains Boltzmann steady-state convergence through stochastic Langevin dynamics sampling.
>
> We thank the reviewer for the valuable suggestion, and will include these additional ablation results in the revised paper.
>
> **References**
>
> Yaron Lipman, Ricky T. Q. Chen, Heli Ben-Hamu, Maximilian Nickel, and Matthew Le. Flow matching for generative modeling. In The Eleventh International Conference on Learning Representations, 2023.

---

> ### Author Response · Authors · 2025-03-18
> **Analyze and Address the Performance Gap**
>
> # Requested Change:
>
> **Analyze and Address the Performance Gap**
>
> **Proposed Adjustment: Provide a detailed analysis of the performance gap compared to state-of-the-art generative models (e.g., DDAEBM), including potential explanations and strategies for improvement.**
>
> # Response:
>
> We appreciate the reviewer’s thoughtful feedback and recognize the importance of clearly addressing the performance gap relative to state-of-the-art models such as DDAEBM. While our proposed Variational Potential Flow Bayes (VPFB) model demonstrates competitive FID performance, we acknowledge a noticeable performance gap compared to leading energy-based models like DDAEBM.
>
> **Difference in Model Architecture**
>
> DDAEBM (Geng et al., 2024) utilizes a multi-model architecture comprising three distinct components: (1) generator model parameterized by the modified U-Net architecture of Xiao et al. (2022), (2) an energy model parameterized by the NCSN++ architecture from Song et al. (2021), and (3) a CNN-based encoder. This multi-component architecture enables specialized modules to collaboratively refine each other's behavior through adversarial training, thereby enhancing generative performance. In contrast, our VPFB model adopts a simpler framework design, employing only a single energy-based model parameterized by the U-Net architecture described in Dhariwal & Nichol (2021). Although this single-component architecture has a comparable parameter count to that of the NCSN++ used in DDAEBM, it may lack the collaborative optimization and mutual refinement advantages that arise from multi-model training setups. Consequently, our VPFB’s FID scores align more closely with those of single-energy or joint-energy EBMs (Salimans & Ho, 2021; Gao et al., 2021; Lee et al., 2023). We hypothesize that incorporating additional auxiliary model components with adversarial training strategies, as utilized by DDAEBM, could enhance the representational capacity and further sharpen the energy landscape of our VPFB model, potentially leading to improved FID performance. Investigating this hybrid approach, which balances computational efficiency with adversarial training strategy, will be reserved for our future work.
>
> **Boltzmann stationarity and FID Trade-off**
>
> In our previous ablation study, we observed a notable trade-off between Boltzmann stationarity and FID performance. This observation aligns with insights from Agoritsas et al. (2023), which show that non-convergent EBMs outperform convergent EBMs trained with long-run MCMC sampling in image generation. Models such as DDAEBM fall within this class of non-convergent EBMs. We hypothesize that this is due to the inherent tension between accurate equilibrium modeling and sharp generative quality. Specifically, imposing strict Boltzmann convergence tends to smooth out the energy landscape, inadvertently flattening local minima that correspond to meaningful data modes. Although such smoothing enhances theoretical interpretability by faithfully approximating the true equilibrium of the Boltzmann distribution, it compromises the sharpness and detail of generated samples, leading to higher FID scores. To mitigate this limitation, we propose to explore advanced sampling techniques such as the Metropolis-Adjusted Langevin Algorithm (MALA) or other adaptive short-run samplers in future work. The gradient-informed proposal and acceptance steps of MALA could potentially enhance local mode exploration efficiency without fully abandoning the Boltzmann equilibrium.
>
> **Conditional vs Marginal EBMs**
>
> Another critical contributing factor is the distinction between conditional EBMs as modeled by DDAEBM, and marginal EBMs considered by VPFB. In particular, Geng et al. (2024) articulate that modeling conditional distributions simplifies the learning task, as these conditional distributions are inherently less multi-modal compared to complex marginal distributions. DDAEBM leverages this insight by decomposing the generation process into discrete diffusion steps, each focusing on simpler, conditional distributions that are easier to model effectively. In contrast, our VPFB explicitly models a global marginal distribution, thus inherently facing greater complexity due to increased modality. Consequently, achieving comparable generative performance poses additional challenges. To address this limitation, future work could explore hierarchical or multi-stage conditional modeling strategies to simplify explicit marginal modeling. Nevertheless, conditional EBMs inherently lack a direct relationship to the marginal Boltzmann distribution, which is fundamental to the theoretical underpinnings of our VPFB framework. Therefore, adapting conditional modeling techniques to fully marginal EBMs would require substantial developments.

---

> ### Author Response · Authors · 2025-03-18
> **Analyze and Address the Performance Gap**
>
> **References**
>
> Cong Geng, Tian Han, Peng-Tao Jiang, Hao Zhang, Jinwei Chen, Søren Hauberg, and Bo Li. Improving adversarial energy-based model via diffusion process. In Forty-first International Conference on Machine Learning, 2024.
>
> Zhisheng Xiao, Karsten Kreis, and Arash Vahdat. Tackling the generative learning trilemma with denoising diffusion GANs. In International Conference on Learning Representations, 2022.
>
> Yang Song, Jascha Sohl-Dickstein, Diederik P Kingma, Abhishek Kumar, Stefano Ermon, and Ben Poole. Score-based generative modeling through stochastic differential equations. In International Conference on Learning Representations, 2021.
>
> Prafulla Dhariwal and Alexander Quinn Nichol. Diffusion models beat GANs on image synthesis. In A. Beygelzimer, Y. Dauphin, P. Liang, and J. Wortman Vaughan (eds.), Advances in Neural Information Processing Systems, 2021.
>
> Tim Salimans and Jonathan Ho. Should EBMs model the energy or the score? In Energy Based Models Workshop - ICLR 2021, 2021.
>
> Ruiqi Gao, Yang Song, Ben Poole, Ying Nian Wu, and Diederik P Kingma. Learning energy-based models by diffusion recovery likelihood. In International Conference on Learning Representations, 2021.
>
> Hankook Lee, Jongheon Jeong, Sejun Park, and Jinwoo Shin. Guiding energy-based models via contrastive latent variables. In The Eleventh International Conference on Learning Representations, 2023.
>
> Elisabeth Agoritsas, Giovanni Catania, Aurélien Decelle, and Beatriz Seoane. Explaining the effects of non-convergent MCMC in the training of energy-based models. In Proceedings of the 40th International Conference on Machine Learning, volume 202 of Proceedings of Machine Learning Research, pp. 322–336. PMLR, 2023.

---

> ### Author Response · Authors · 2025-03-19
> **Expand the Experimental Scope**
>
> # Requested Change:
>
> **Expand the Experimental Scope**
>
> **Proposed Adjustment: Provide a detailed analysis of the performance gap compared to state-of-the-art generative models (e.g., DDAEBM), including potential explanations and strategies for improvement.**
>
> # Response:
> We thank the reviewer for the suggestion to expand the experimental scope. Our current evaluation includes CIFAR-10 (32×32) and CelebA (64×64), where VPFB has demonstrated strong generalization and competitive FID scores. To assess the scalability of VPFB to larger images, we have initiated experiments on the higher-resolution ImageNet (128×128) and LSUN Church (128×128). Given the significantly increased computational demands, we earnestly request additional time to complete the model training and sampling. We will make every effort to include these additional findings in the revised manuscript.

---

### Review · Reviewer_q12m · 2025-03-04

**Summary Of Contributions:**

This work explores the problem of image generation with a time-dependent potential energy function whose gradient dynamics guide samples from a noise prior distribution to realistic image samples. The paper begins by defining a conditional density homotopy and an associated marginal density homotopy, where the marginal density homotopy at t = 0 follows the prior distribtution and the homotopy at t = 1 approximately follows the distribution of observed data. The timesteps 0 < t < 1 provide a path to guide samples from noise to data during the data generation process. The next section finds a formula for the evolution of the density homotopy over time and defines a Poisson equation that gives a condition under which the KL divergence between the homotopy from a learned potential energy flow and the target data homotopy is minimized. The steady-state behavior of such a learned homotopy is shown to follow a Boltzmann distribution with a certain form. Finally, a practical loss function for solving the Poisson equation and learning the potential energy flow is presented. Experiments are conducted for learning densities in low dimensions, generating CIFAR-10 and Celeb-A images, and the OOD performance of the Boltzmann energy.

**Audience:**

Yes

**Broader Impact Concerns:**

Broader impacts were not discussed, although this does not impact my assessment of the paper.

**Claims And Evidence:**

Yes

**Requested Changes:**

It would be helpful to include a section that describes the benefits of the proposed potential energy flow compared to the typical diffusion or flow matching models where the flow vector field is defined directly by the network output rather than the gradient of a scalar valued network.

**Strengths And Weaknesses:**

Strengths:
* The paper presents a novel framework for using potential energy functions to define data-generating gradient flows. The paper brings together a variety of ideas from EBMs, diffusion models, flow matching, and deep Ritz learning into an interesting and novel paradigm.
* Several non-trivial propositions are presented to describe the density homotopy evolution, define a Poisson equation that can minimize KL divergence with a learned model, define the Boltzmann steady-state distribution of the learned model, and to define a loss that can solve the Poisson equation. The appendix provides detailed derivations of each proposition.
* The results show strong generative performance of the proposed method among recent EBMs and strong OOD performance from the Boltzmann density.

Weaknesses:
* To my understanding, the density defined by the Boltzmann energy $\Phi_B$ does not necessarily follow the approximate data density $\bar{\rho} (x)$. Is this correct? In this case, the proposed method does not accomplish the typical EBM goal of learning a Boltzmann energy that can approximate the data density, but rather learns a potential flow that can guide samples from the noise density to the data density in finite time. It might not be entirely accurate to categorize this proposed method as an EBM in the sense used in many prior works.
* Despite the strong performance of the method compared to other EBMs, the results still lag behind diffusion and flow matching models.
* The density estimation application in Figure 1 doesn't seem very convincing, because the learned densities in the top row do not seem to match the expected 2-moons distribution.

---

> ### Author Response · Authors · 2025-03-10
>
> We sincerely thank Reviewer q12m for a thorough and thoughtful review of our paper. We appreciate the positive recognition of our work and the valuable constructive feedback, which has helped us improve our manuscript.
>
> Our rebuttals will be provided in separate comments.
> Additional figures (R.7–R.8) used in rebuttals are provided in the supplementary materials under the *rebuttal* folder.
>
> **Broader Impact Concerns:** We sincerely appreciate Reviewer q12m's feedback. In response, we will add a Broader Impact Statement at the end of the paper, following the conclusion. This section will discuss the ethical implications of generative modeling, including the risks of misuse for harmful content and the potential amplification of biases in training data. Additionally, we will outline safeguards to mitigate these risks and promote the responsible use of this technology.

---

> ### Author Response · Authors · 2025-03-12
> **Clarifying the EBM Perspective and the Alignment of Boltzmann Distribution and Data Likelihood**
>
> # Weakness:
>
> **To my understanding, the density defined by the Boltzmann energy $\Phi_B$ does not necessarily follow the approximate data density $\bar{\rho} (x)$. Is this correct? In this case, the proposed method does not accomplish the typical EBM goal of learning a Boltzmann energy that can approximate the data density, but rather learns a potential flow that can guide samples from the noise density to the data density in finite time. It might not be entirely accurate to categorize this proposed method as an EBM in the sense used in many prior works.**
>
> # Response:
>
> We appreciate the opportunity to clarify that the Boltzmann energy $\Phi_{B}$ is indeed formulated to approximate the true data likelihood $p_{\text{data}}(\bar{x})$. In particular, we define the marginal density homotopy in Equation (8) as
> $\bar{\rho}(x,t) = \int_{\Omega} \ p_{\text{data}}(\bar{x}) \ \rho(x \mid \bar{x}, t) \ d\bar{x}$.
> This marginal homotopy approximates $p_{\text{data}}(\bar{x})$, provided that the conditional density homotopy $\rho(x \mid \bar{x}, t)$ defined in Equation (4) is concentrated around the data point $\bar{x}$ at $t = t_{\text{max}}$, where the cutoff time $t_{\text{max}} < 1$ is set close to 1.
> By defining the conditional density homotopy to be a Gaussian perturbation kernel
> $\rho(x \mid \bar{x}, t) = \mathcal{N}\big(\mu(t) \bar{x}, \sigma(t)^2 I\big)$
> with parameters $\mu(t_{\text{max}}) \to 1$ and $\sigma(t_{\text{max}}) \to 0$, the perturbation kernel approaches a Dirac delta function. As a result, we obtain the approximation
> $\bar{\rho}(x, t=t_{\text{max}}) \approx p_{\text{data}}(\bar{x})$ from Equation (8).
> Here, *density homotopy* refers to a density function that evolves over time.
>
> To establish a Boltzmann distribution, we further require that the time-varying marginal density $\bar{\rho}(x,t)$ converges to a stationary Boltzmann equilibrium, e.g., $\frac{\partial \bar{\rho}(x,t)}{\partial t} = 0$. However, the Gaussian perturbation kernel $\rho(x \mid \bar{x}, t) = \mathcal{N}\big(\mu(t) \bar{x}, \sigma(t)^2 I\big)$ used in diffusion and flow-based models is defined only over a finite interval $t \in [0,t_{\text{max}}]$. Additionally, the resulting marginal homotopy $\bar{\rho}(x,t)$ is not guaranteed to reach equilibrium within this time interval. To resolve these limitations of diffusion or flow-based probability paths, we explicitly enforce stationarity in our training implementation, by imposing a steady-state equilibrium $p_{\infty}(x) = \bar{\rho}(x, t \geq t_{\text{max}}) \approx p_{\text{data}}(\bar{x})$ beyond the cutoff time. This enforces the steady-state equilibrium $p_{\infty}(x)$ aligns with the data likelihood.
>
> Finally, Proposition 5 establishes that when the Fokker-Planck dynamics of $\bar{\rho}(x,t)$ converges to equilibrium $\frac{\partial \bar{\rho}(x,t)}{\partial t} = 0$, the Fokker-Planck equation admits a unique steady-state solution, given by the Boltzmann distribution $p_{\infty}(x) = p_{B}(x) \propto e^{\Phi_{B}(x)}$ in Equation (20). Given that a steady-state equilibrium is enforced via $p_{\infty}(x) = \bar{\rho}(x, t \geq t_{\text{max}}) \approx p_{\text{data}}(\bar{x})$, the stationary Boltzmann distribution approximates the true data likelihood by design. Hence, our proposed framework offers an alternative approach to learning EBMs by enforcing the stabilization of the Fokker-Planck dynamics of the marginal homotopy $\bar{\rho}(x,t)$ when $t \geq t_{\text{max}}$.
>
> Nevertheless, this marginal homotopy $\bar{\rho}(x,t)$ cannot be solved analytically in closed form. Therefore, we approximate $\bar{\rho}(x,t)$ using the flow-driven density homotopy ${\rho_{\Phi}}(x,t)$ introduced in Section 3.2. Proposition 6 further shows that a homotopy matching $\frac{\partial {\rho_{\Phi}}(x,t)}{\partial t} \equiv \frac{\partial \bar{\rho}(x,t)}{\partial t}$ is achieved by minimizing the variational loss function in Equation (22). By doing so, the flow-driven homotopy ${\rho_{\Phi}}(x,t)$ encapsulates the steady-state convergence dynamics of the stationary-enforced marginal $\bar{\rho}(x,t)$.
>
> We thank the reviewer for this insightful comment. We hope that our explanation provides a better clarification of the correspondence between the Boltzmann energy and the data density.

---

> ### Author Response · Authors · 2025-03-12
> **Addressing the Limitation and Overlooked Assumption in 2D Density Estimation**
>
> # Weakness:
>
> **The density estimation application in Figure 1 doesn't seem very convincing, because the learned densities in the top row do not seem to match the expected 2-moons distribution.**
>
> # Response:
>
> We appreciate the reviewer for raising this issue. The derivation of the Boltzmann distribution in Proposition 5 relies on the condition that the deterministic potential flow in Equation (11) satisfies the marginal probability flow ODE in Equation (17).
> Proposition 4 demonstrates that this condition is naturally met when using a Gaussian perturbation kernel
> ${\rho}(x \mid \bar{x},t) = \mathcal{N}\big(x; \mu(t) \bar{x}, \sigma(t)^2 I\big)$
> which characterizes a forward diffusion process governed by the forward-time SDE in Equation (15). In particular, we implemented the optimal transport path of flow matching (Lipman et al., 2023), which corresponds to a forward diffusion process with drift and diffusion functions given by $f(t) = - \frac{1}{t}$ and $g(t) = \sqrt{\frac{2 (1-t)}{t}}$ .
>
> However, we overlooked the fact that the forward-time SDE in Equation (15) (or equivalently, the marginal probability flow ODE in Equation (17)) is valid only when the Gaussian noise perturbation ends at a unimodal Gaussian prior, e.g., $q(x) = {\rho}(x \mid \bar{x},t=0) = \mathcal{N}(0, \omega^2 I)$ (for diffusion models, or starts from it in the case of flow matching), as discussed in Kingma & Gao (2023). We believe this leads to a non-convergent Boltzmann density and a lack of resemblance to the data distribution in the 2D density estimation results shown in Figure 1. We acknowledge this limitation of our current framework and will discuss this in the conclusion of the revised paper. We regret this oversight in our previous density estimation results.
>
> Additionally, we modified the prior distribution in our 2D density estimation experiment, replacing the **8 Gaussians** with a **unimodal Gaussian**, and present the new results in Figure R.7 and Figure R.8 (included in the supplementary materials). Figure R.7 shows the sample trajectories driven by the potential flow $dx(t) = \nabla_{x} \Phi(x, t) \ dt$, sampled via deterministic Euler solver. Figure R.8 presents the sample trajectories and density estimation of the Boltzmann distribution $p_B \propto e^{\Phi_{B}(x)}$, obtained via stochastic Langevin Dynamics sampling. Notably, with a unimodal Gaussian prior, both the potential energy $\Phi(x)$ and Boltzmann energy $\Phi_{B}(x)$ exhibit faster convergence to their steady-state equilibrium. Furthermore, Figure R.8 demonstrates improved steady-state convergence of the Boltzmann energy, resulting in a density estimation that more closely aligns with the 2-moons distribution.
>
> We appreciate this valuable comment and will integrate these updated results into the revised paper.
>
> **References**
>
> Yaron Lipman, Ricky T. Q. Chen, Heli Ben-Hamu, Maximilian Nickel, and Matthew Le. Flow matching for generative modeling. In The Eleventh International Conference on Learning Representations, 2023.
>
> Diederik Kingma and Ruiqi Gao. Understanding diffusion objectives as the elbo with simple data augmentation. In Advances in Neural Information Processing Systems, volume 36, pp. 65484–65516. Curran Associates, Inc., 2023.

---

> ### Author Response · Authors · 2025-03-13
> **Benefits of Energy-Parameterized Potential Flow over Diffusion or Flow Matching Models**
>
> # Requested Change:
>
> **It would be helpful to include a section that describes the benefits of the proposed potential energy flow compared to the typical diffusion or flow matching models where the flow vector field is defined directly by the network output rather than the gradient of a scalar valued network.**
>
> # Response:
>
> We thank the reviewer for this valuable suggestion. Our proposed energy-parameterized potential flow offers several advantages over conventional diffusion and flow matching models, where vector fields are directly parameterized by neural networks rather than derived as the gradients of scalar-valued energy functions. Specifically, these benefits include:
>
> **Interpretable Energy Landscape for Explicit Density Modeling**
>
> By explicitly parameterizing the vector field as the gradient of a scalar potential energy $\Phi(x,t)$, our method provides a natural energy-based representation of data dynamics. Such a formulation supports key energy-based modeling tasks including explicit (marginal) density estimation, composable generation, and out-of-distribution (OOD) detection capabilities not inherently provided by conventional diffusion or flow matching approaches. This is supported in Section 4.4 of our manuscript, where we demonstrate that VPFB achieves strong OOD detection performance due to its energy-based formulation .
>
> **Theoretical Justification as an Energy-based Model**
>
> Our Boltzmann energy formulation in Proposition 5 rigorously connects the deterministic potential flow (Equation 11) to a stationary Boltzmann distribution characterized by the Boltzmann energy $\Phi_B(x)$. This theoretical foundation firmly situates our approach within the energy-based modeling framework, offering theoretical coherence that is lacking in existing diffusion and flow matching models. As a result, it allows our approach to combine the efficiency of diffusion or flow-based generative paths with the interpretability and rigor of the stationary Boltzmann energy representation. We highlight these theoretical insights in Section 3.3 and 3.4, where we derive the Boltzmann formulation for VPFB and demonstrate its ability to model data with a well-defined energy landscape .
>
> **Optimality of the Conservative Vector Field**
>
> Our approach learns a purely gradient-based vector field $v(x,t) = \nabla_x \Phi(x,t)$, in contrast to diffusion and flow matching methods, which may incorporate divergence-free components, as noted in Neklyudov et al. (2023). By enforcing a conservative energy function through Helmholtz decomposition, our method reduces the dynamical cost associated with these divergence-free components, enabling more efficient particle transport and improving training efficiency.
>
> **Improved ODE-Based Sampling Efficiency**
>
> By eliminating the reliance on implicit MCMC sampling, our approach reduces computational overhead and avoids common convergence issues encountered in traditional energy-based models. Furthermore, our flow-based formulation enables deterministic ODE-based sampling that is generally more stable and efficient for generating high-quality samples with fewer steps than stochastic sampling methods.
>
>
> We greatly appreciate the reviewer’s suggestion and will highlight these benefits in a dedicated section of the revised manuscript.
>
> **References**
>
> Kirill Neklyudov, Rob Brekelmans, Daniel Severo, and Alireza Makhzani. Action matching: Learning stochastic dynamics from samples. In Proceedings of the 40th International Conference on Machine Learning, volume 202 of Proceedings of Machine Learning Research, pp. 25858–25889. PMLR, 2023.

---

### Decision · Action_Editor_VVWy · 2025-04-16

**Recommendation:** Accept as is

**Comment:**

The paper offers a novel and theoretically strong framework for training EBMs without MCMC. Despite some empirical limitations, all reviewers agreed that the method demonstrates competitive performance compared to existing EBMs and presents a promising direction for future research.

**Audience:**

Yes

**Claims And Evidence:**

Yes

---

> ### Author Response · Authors · 2025-04-22
> **Expression of Gratitude for Final Recommendation and Review Process**
>
> We would like to express our sincere gratitude to Action Editor VVWy for a thoughtful evaluation and the positive recommendation of our manuscript. We truly appreciate the time and care in handling our submission and chairing the rebuttal.